# Acute RyR1 Ca$^{2+}$ leak enhances NADH-linked mitochondrial respiratory capacity

Nadège Zanou [1✉], Haikel Dridi [2], Steven Reiken[2], Tanes Imamura de Lima[3], Chris Donnelly [1], Umberto De Marchi [4], Manuele Ferrini [1], Jeremy Vidal[1], Leah Sittenfeld[2], Jerome N. Feige [4,5], Pablo M. Garcia-Roves [6], Isabel C. Lopez-Mejia [7], Andrew R. Marks[2,8], Johan Auwerx [3], Bengt Kayser [1] & Nicolas Place [1✉]

Sustained ryanodine receptor (RyR) Ca$^{2+}$ leak is associated with pathological conditions such as heart failure or skeletal muscle weakness. We report that a single session of sprint interval training (SIT), but not of moderate intensity continuous training (MICT), triggers RyR1 protein oxidation and nitrosylation leading to calstabin1 dissociation in healthy human muscle and in in vitro SIT models (simulated SIT or S-SIT). This is accompanied by decreased sarcoplasmic reticulum Ca$^{2+}$ content, increased levels of mitochondrial oxidative phosphorylation proteins, supercomplex formation and enhanced NADH-linked mitochondrial respiratory capacity. Mechanistically, (S-)SIT increases mitochondrial Ca$^{2+}$ uptake in mouse myotubes and muscle fibres, and decreases pyruvate dehydrogenase phosphorylation in human muscle and mouse myotubes. Countering Ca$^{2+}$ leak or preventing mitochondrial Ca$^{2+}$ uptake blunts S-SIT-induced adaptations, a result supported by proteomic analyses. Here we show that triggering acute transient Ca$^{2+}$ leak through RyR1 in healthy muscle may contribute to the multiple health promoting benefits of exercise.

[1] Institute of Sport Sciences and Department of Biomedical Sciences, University of Lausanne, Lausanne, Switzerland. [2] Department of Physiology and Cellular Biophysics, Clyde and Helen Wu Center for Molecular Cardiology, Columbia University Vagelos College of Physicians and Surgeons, New York, NY, USA. [3] Laboratory of Integrative Systems Physiology (LISP), École Polytechnique Fédérale de Lausanne (EPFL), Lausanne, Switzerland. [4] Nestlé Research – École Polytechnique Fédérale de Lausanne (EPFL), Innovation Park, Lausanne, Switzerland. [5] School of Life Sciences, École Polytechnique Fédérale de Lausanne (EPFL), Lausanne, Switzerland. [6] Department of Physiological Sciences, School of Medicine and Health Sciences, University of Barcelona and Bellvitge Biomedical Research Institute (IDIBELL), L'Hospitalet del Llobregat, Barcelona, Spain. [7] Center for Integrative Genomics, University of Lausanne, Lausanne, Switzerland. [8] Department of Medicine, Division of Cardiology, Columbia University Medical Center, New York, NY, USA. ✉email: nad.zanou.mdphd@gmail.com; nicolas.place@unil.ch

Ryanodine receptors (RyRs) encode intracellular $Ca^{2+}$ release channels located on the endo/sarcoplasmic reticulum (ER/SR)[1]. In mammals, there are three isoforms. RyR1 and RyR2 are involved in skeletal and cardiac muscle excitation–contraction coupling, respectively[2], and are also found in non-muscle tissues[3], while RyR3, originally identified in the brain[4], is expressed in many tissues[5]. RyRs are >2 million dalton homotetramers and are the largest known ion channels. RyR1 and RyR2 are stabilized by two forms of *FKBP*-encoded proteins: calstabin (*cal*cium channel *sta*bilizing *bin*ding protein, referring to its role in calcium homeostasis) 1 and calstabin2 (FKBP12 and FKBP12.6, respectively)[6]. The function of RyRs is further modulated by kinases[7], phosphatases[8], phosphodiesterases[7,9] and calmodulin[7]. RyR oxidation, phosphorylation and nitrosylation can lead to dissociation of calstabin from the channel, leading to $Ca^{2+}$ leaking from the ER/SR[7]. Sustained leak can excessively elevate cytosolic and mitochondrial $Ca^{2+}$, inducing increased ROS production, mitochondrial dysfunction and cell damage[10]. This has been linked to pathological conditions such as heart failure[11], muscle dystrophy[12], muscle weakness during cancer[13] and ageing[14]. While sustained leaky RyRs may be involved in pathological conditions, recent findings in healthy people engaging in exercise training suggest that an acute transient SR $Ca^{2+}$ leak via RyR channels in muscle paradoxically might have a beneficial role[15], a contention that remained to be verified.

The benefits of exercise to prevent and treat metabolic diseases are indisputable[16], but the discussion on the best combination of exercise type (volume and intensity needed to obtain a specific outcome) is ongoing. Physical training is defined as "any form of physical exercise that is repeated regularly for a certain period of time"[17]. High-intensity exercise may be more effective than moderate-intensity continuous training (MICT) to counteract metabolic diseases such as overweight or obesity[18,19], a reason why high-intensity interval training (HIIT) is now increasingly used in patient care[20–22]. A particularly time-efficient mode, sprint interval training (SIT), leads to similar increases in maximum aerobic capacity ($VO_2$peak) in healthy populations as compared to classical MICT due to rapid skeletal muscle metabolic adaptations[23,24]. The molecular mechanisms triggering such benefits remain to be determined and RyR1 function/$Ca^{2+}$ may be involved[15,25,26]. We previously reported that one session of SIT modifies the RyR1 channel in human and mouse muscle fibres, which resulted in increased resting cytosolic $Ca^{2+}$ in mouse muscle fibres[26]. In addition, 3 weeks of wheel running in mice induced RyR1 phosphorylation associated with increased fatigue resistance[15]. We report here results describing a molecular mechanism by which an exercise-induced acute transient SR $Ca^{2+}$ leak via RyR1 causes muscle mitochondrial remodeling leading to improved respiratory function.

We studied humans performing a single session of SIT or MICT, and combined data from human muscle biopsies, in vitro simulated SIT and MICT (S-SIT and S-MICT, respectively) on C2C12 myotubes and ex vivo S-SIT and S-MICT on isolated mouse muscle fibres. Our results show that a single session of (S-) SIT, but not of (S-)MICT, induces RyR1 post-translational modifications leading to calstabin1 dissociation from the channel and decreased SR $Ca^{2+}$ content. This is associated with increased mitochondrial $Ca^{2+}$ uptake, reduced pyruvate dehydrogenase (*PDH*)-encoded protein phosphorylation on $Ser^{293}$, increased levels of mitochondrial oxidative phosphorylation (OXPHOS) proteins, increased mitochondrial supercomplex (SC) formation, and improved mitochondrial respiratory capacity. Altering RyR1 $Ca^{2+}$ leak using S107, a 1,4-benzothiazepine derivative known as a Rycal (a new class of drugs that stabilize the closed state of RyR1) thereby reducing intracellular $Ca^{2+}$ leak[27], and knocking down of the mitochondrial $Ca^{2+}$ uniporter channel (*MCU*)-encoded

protein indicate that the positive effects of SIT on mitochondrial adaptations are, at least in part, mediated by a specific RyR1 $Ca^{2+}$-dependent signal and mitochondrial $Ca^{2+}$ uptake that regulates mitochondrial protein abundance and function.

## Results

### In humans, a single session of SIT—but not MICT—triggers muscle RyR1 oxidation and nitrosylation, as well as calstabin1 dissociation. Recreationally active volunteers matched for their aerobic fitness (Supplementary Fig. 1a) performed a single exercise session of MICT (1 h cycling at 65% of power output at $VO_2$peak; total work 680.9 ± 82.2 kJ, Fig. 1a₁) or SIT (six bouts of 30 s "all-out" cycling on a cycle ergometer with 4 min rest; cumulated work of the six bouts 88.8 ± 18.0 kJ, Fig. 1a₂). Knee extensor neuromuscular function was assessed using maximal voluntary contractions (MVC) combined with electrical stimulation before (Pre), immediately after (Post) and 24 h after exercise (24 h post). The molecular changes were investigated in biopsies from *vastus lateralis* muscle, before (Pre), ~10 min after (Post) and 24 h after exercise (24 h post) (Fig. 1a).

In line with our previous studies[26,28], we observed a ~30% decrease in MVC force immediately after one session of SIT or MICT with complete recovery 24 h later (Fig. 1b). We electrically stimulated the femoral nerve and observed a small defect in the maximal voluntary activation level of the knee extensors (i.e. minor central fatigue; Supplementary Fig. 1b) independent of exercise intensity. While no impairment in M-wave amplitude was noticed (i.e. preserved sarcolemmal excitability; Supplementary Fig. 1c), electrically evoked forces at 100 and 10 Hz were reduced (Supplementary Fig. 1d, e) immediately after exercise in both groups, indicating substantial peripheral fatigue (i.e. in the muscle fibres). We also observed greater low-frequency force depression after SIT compared to MICT, as illustrated by the reduced 10/100 Hz ratio (Fig. 1c). As low-frequency force depression is associated with impaired $Ca^{2+}$ handling in skeletal muscle fibres[29], this suggests greater perturbations of $Ca^{2+}$ homeostasis in response to SIT compared to MICT.

We then used a co-immunoprecipitation assay to investigate whether a single session of SIT or MICT may trigger RyR1 post-translational modifications (PTMs; phosphorylation, nitrosylation or oxidation) and calstabin1 dissociation, a leaky RyR signature[27], in human *vastus lateralis* muscle biopsies. After SIT, muscle biopsies showed significant RyR1 oxidation, nitrosylation and phosphorylation, which was accompanied by calstabin1 dissociation (Fig. 1d, e–h), suggesting a potentially leaky RyR1. After MICT, only slight RyR1 nitrosylation was observed (Fig. 1d, g). Although MICT-induced RyR1 nitrosylation was significant immediately post exercise, this was not sufficient to induce calstabin1 dissociation from RyR1 (Fig. 1d, g, h). Our immunoprecipitated (IP) RyR1 preparations were specific as the sarco-endoplasmic $Ca^{2+}$-ATPase (SERCA) protein was detected only in the non-IP sample (Supplementary Fig. 1f). We next investigated the protein levels of the dihydropyridine receptor (DHPR) and of the SERCA, both involved in muscle excitation–contraction coupling. Neither SIT nor MICT significantly modified DHPR, SERCA-2 or SERCA1 protein levels (Supplementary Fig. 1g–j). These findings indicate that a single session of SIT, but not of MICT, triggered RyR1 oxidation, nitrosylation and phosphorylation, accompanied by calstabin1 dissociation, suggesting a leaky RyR1 channel in skeletal muscle of recreationally active healthy young men.

### SIT-induced RyR1 modifications are followed by an increase in levels of mitochondrial OXPHOS proteins in human muscle biopsies. Exercise training increases human skeletal muscle

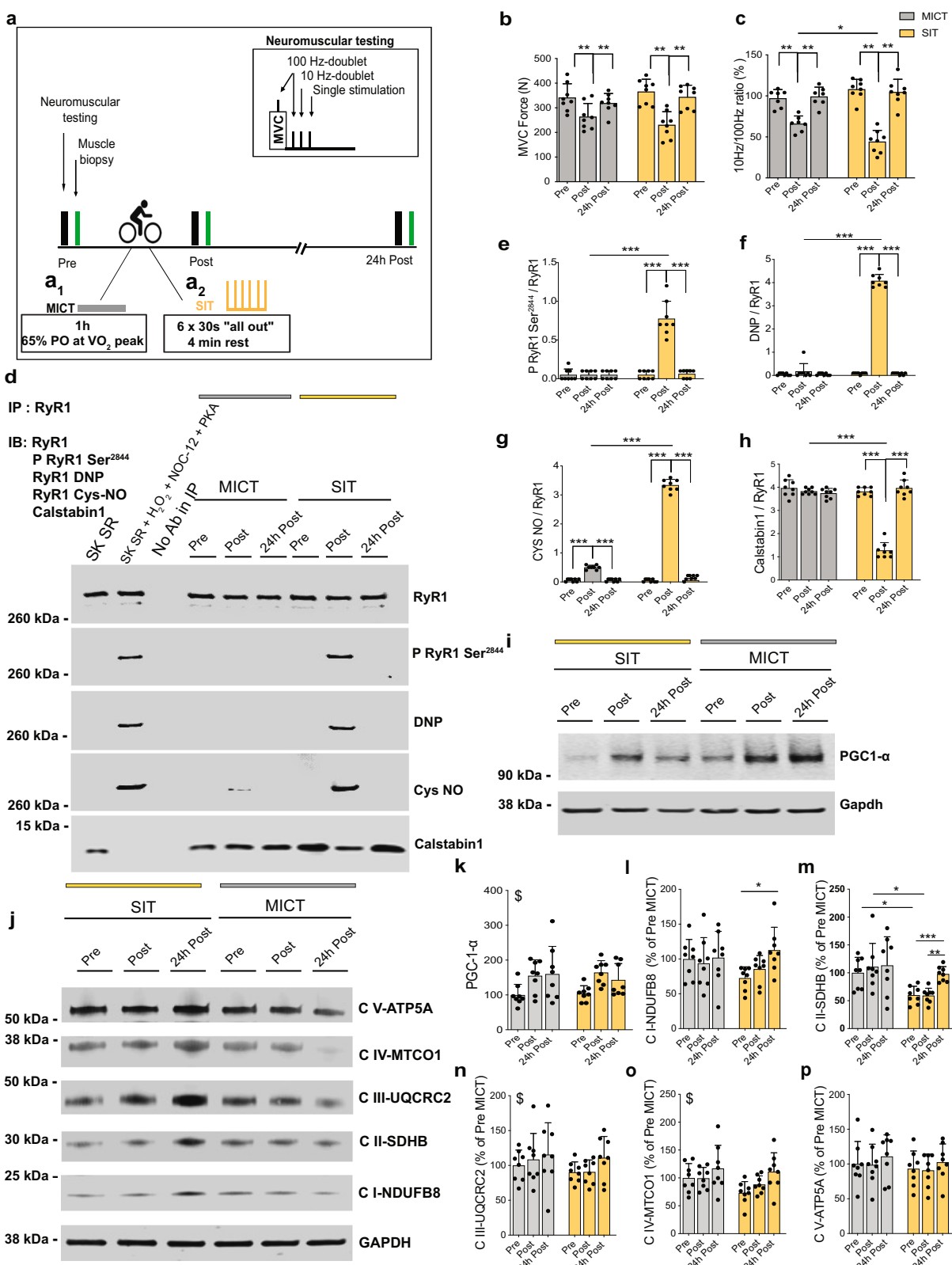

mitochondrial respiratory capacity[30,31]. This results from increased mitochondrial volume, higher OXPHOS protein levels and improved mitochondrial electron transfer system (ETS) capacity and efficiency[32]. The peroxisome proliferator-activated receptor co-activator 1α (PGC-1α) is important in the activation of genes required for mitochondrial biogenesis, in coordination with the nuclear regulatory factor (NRF) and the mitochondrial

transcription factor A (Tfam)[33,34]. Endurance exercise induces a strong transient increase in PGC-1α messenger RNA (mRNA) in human skeletal muscle[35], and even a single bout of endurance exercise increases PGC-1α mRNA and protein levels in human muscle[36]. Using an antibody that recognizes all PGC-1α isoforms[37], we observed that a single session of both MICT and SIT induced increased PGC-1α protein post exercise (Fig. 1i, k).

**Fig. 1 One session of SIT induces RyR1 post-translational modifications, calstabin1 dissociation from the RyR1 complex and leads to increased OXPHOS protein expression in human muscle. a** Models of MICT (**a₁**) and SIT (**a₂**) in humans. **b**, **c** Assessment of **b** knee extensor maximal voluntary contraction (MVC) force; $n = 8$ participants per group and **c** the ratio of electrically evoked forces at 10 and 100 Hz; $n = 7$ and 8 participants for MICT and SIT, respectively. Two-way ANOVA followed by Sidak's multiple comparisons test. **d** Representative immunoblots (IB) of immunoprecipitated (IP) RyR1, RyR1 post-translational modifications and calstabin1 dissociation. DNP (2,4-dinitrophenylhydrazone): RyR1 oxidation. P RyR1 Ser$^{2844}$: RyR1 phosphorylation at serine 2844. Cys NO: RyR1 nitrosylation. SK SR: skeletal muscle sarcoplasmic reticulum vesicle. SK SR treated with 200 μM $H_2O_2$, 250 μM NOC-12 and 5 units PKA per reaction: positive control for RyR1 oxidation and nitrosylation and calstabin1 dissociation. No antibody in IP: negative control. The whole gel and an additional control are shown in Supplementary Fig. 1f. **e–h** Quantification of immunoblots in (**d**); $n = 8$ participants per group. Two-way ANOVA followed by Sidak's multiple comparisons test. **i** Representative immunoblots of PGC-1α. **j** Representative immunoblots of mitochondrial OXPHOS proteins. All the cropped parts of OXPHOS proteins are part of the same blot that is shown in Supplementary Fig. 5. **k** Quantification of PGC-1α proteins in (**i**) related to GAPDH protein and expressed as % of Pre-MICT; $n = 8$ participants per group. Two-way ANOVA. **l–p** Quantification of OXPHOS proteins in (**j**) related to GAPDH protein and expressed as % of Pre-MICT; $n = 8$ participants per group. Two-way ANOVA, followed by Sidak's multiple comparisons test (**l**, **m**). Data are mean ± SD. *$p \leq 0.05$, **$p \leq 0.01$, ***$p \leq 0.001$, and $^{\$}$main effect of time. Source data are provided as a Source Data file.

During muscle contraction, there is a large increase in cytosolic $Ca^{2+}$ due to $Ca^{2+}$ release from the SR. Interestingly, $Ca^{2+}$/calmodulin-dependent protein kinase II (CaMKII) has been shown to trigger PGC-1α expression in response to increased cytosolic $Ca^{2+}$ [38]. CaMKII phosphorylation levels were similarly increased in response to SIT and MICT (Supplementary Fig. 1k, l). No significant changes were observed for NRF1 and Tfam proteins (Supplementary Fig. 1m–o).

We then investigated whether RyR1 modifications in SIT muscle were associated with an increase in OXPHOS proteins. A single session of SIT, but not MICT, was sufficient to significantly increase mitochondrial OXPHOS complexes I (CI) and II (CII) with no difference between groups for the other complexes (Fig. 1j, l–p). In response to exercise, mitochondria can also undergo dynamic remodeling through fission and fusion [39]. The inner mitochondrial membrane fusion process is mediated by optical protein atrophy 1 (OPA1), which has five forms: two long promoting fusion and three short promoting fission [40], while outer mitochondrial membrane fusion is mediated by mitofusin 1 and 2 [41]. The GTPase dynamin-related protein 1 (DRP1) mediates mitochondrial fission [40]. OPA1 total protein has been shown to increase at 3 h post high-intensity, high-volume training, but not 3 h post SIT [42]. However, the effect of SIT on OPA1 forms had yet to be described. Here, we observed no changes in total OPA1 and its different forms after one session of SIT or MICT (Supplementary Fig. 1p–r). DRP1 protein levels were unchanged after SIT or MICT (Supplementary Fig. 1p, s).

**In vitro simulated SIT leads to RyR1 modifications and calstabin1 dissociation accompanied by decreased SR $Ca^{2+}$ content.** In human muscle biopsies, we found RyR1 protein modifications and calstabin1 dissociation accompanied by increased mitochondrial OXPHOS in response to a single session of SIT—but not MICT. To investigate a potential link between those RyR1 modifications and mitochondrial adaptations with techniques unsuitable due to the limited access/availability of human samples, we developed cellular models of SIT (S-SIT) and MICT (S-MICT). We used specific stimulation patterns based on high and low stimulation frequencies, respectively, in mouse C2C12 myotubes [43] that could result in similar changes as observed in our human samples. We chose C2C12 mouse myotubes since they are a useful model to mimic exercise in vitro [44]. From a metabolic point of view, C2C12 myotubes constitute the closest cellular model to adult human muscle available. They are the most suitable for studies of exercise as they share more features with human muscle than rat or primary human skeletal muscle cell cultures [45]. We complemented this model with dissociated muscle fibres from isolated mouse flexor digitorum brevis (FDB) muscle. We chose these models since SIT (i.e. "all-

out" repeated sprints) is not feasible in vivo in mice. We excluded the use of myotubes derived from human primary myoblasts because they do not mimic adult muscle $Ca^{2+}$ handling and lack the capability to contract [46].

To investigate whether a leaky RyR1 status is linked to the observed increased mitochondrial OXPHOS capacity, we first performed S-MICT and S-SIT contractions in C2C12 myotubes using electrical stimulation (Fig. 2a). The $Ca^{2+}$ release patterns differed in response to S-SIT and S-MICT (Supplementary Fig. 2a, b, respectively), as expected from the stimulation patterns. One key marker of exercise-induced metabolic stress is AMP-activated protein kinase (AMPK) phosphorylation [47]. In our cellular models, S-SIT significantly increased AMPK phosphorylation on Thr$^{172}$ (Supplementary Fig. 2c, d), consistent with reports in human skeletal muscle [31]. Metabolomic analyses of whole-cell lysates collected immediately after stimulation showed clear increases in AMP/ATP and ADP/ATP ratios in S-SIT, which support the AMPK phosphorylation in S-SIT myotubes (Fig. 2b). Phosphocreatine/creatine ratios were difficult to interpret and no changes in NAD or NADH levels were observed (Fig. 2b). Both models showed significant increases in vascular endothelial growth factor mRNA levels (Supplementary Fig. 2e) and lactate release (Supplementary Fig. 2f) compared to control, with greater effects in S-SIT than in S-MICT. Taken together, the above results validated our in vitro models mimicking these two exercise modes for mechanistic approaches.

We then investigated RyR1 PTMs and RyR1 association to calstabin1 in response to S-SIT and S-MICT. Just as we observed in human muscle biopsies, a single session of S-SIT, but not S-MICT, induced RyR1 oxidation and nitrosylation, as well as calstabin1 dissociation (Fig. 2c–f), suggesting a leaky RyR1 also in C2C12 myotubes in response to S-SIT. Similar to the human data, our IP RyR1 preparation for the cellular models was free of SERCA contamination (Supplementary Fig. 2g). We investigated in these in vitro models potential alterations in SR $Ca^{2+}$ stores. We first investigated the amount of maximal releasable SR $Ca^{2+}$ in myotubes after S-SIT or S-MICT using the RyR1 agonist caffeine. Immediately after electrical stimulation, caffeine stimulation in the absence of extracellular $Ca^{2+}$ (i.e. with no confounding effects from possibly altered plasma membrane channels) induced less SR $Ca^{2+}$ release in S-SIT myotubes compared to control and S-MICT myotubes, which suggests a reduced SR $Ca^{2+}$ content (Fig. 2g, h). $Ca^{2+}$ can be chelated within the SR through binding to inorganic phosphate, which accumulates during exercise [48] or $Ca^{2+}$ can be sequestered by calsequestrin [49] to limit $Ca^{2+}$ release from the SR. Since these mechanisms could bias our measurements, we also directly measured the SR $Ca^{2+}$ content after S-SIT and S-MICT. We transfected differentiated myotubes with a ratiometric SR-targeted cameleon (D1ER) plasmid to directly assess SR $Ca^{2+}$.

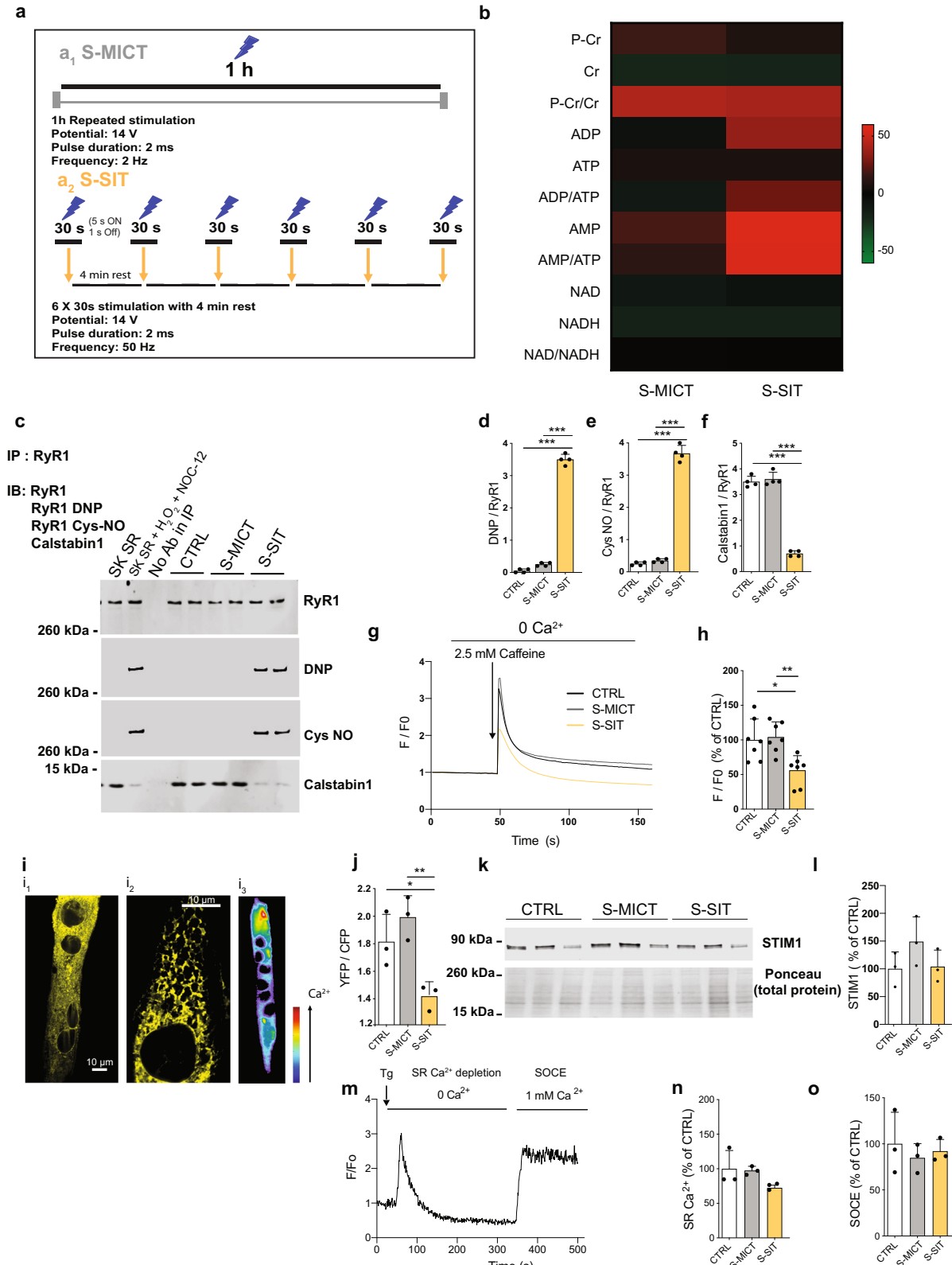

Immediately after S-SIT and S-MICT, resting SR Ca²⁺ stores were quantified using a fluorescence resonance energy transfer system[50]. The amount of D1ER signal, reflecting SR Ca²⁺ levels, was reduced in S-SIT myotubes compared to S-MICT and control myotubes (Fig. 2i, j), indicating decreased SR Ca²⁺ content from a leaky RyR1 after a single S-SIT session. To exclude any alterations in store-operated Ca²⁺ entry (SOCE) in S-SIT and

S-MICT myotubes, we measured the amount of STIM1, the key regulator of SOCE in the skeletal muscle[51] and SOCE Ca²⁺ transients in S-SIT and S-MICT myotubes. We observed no changes in STIM1 expression and SOCE in both conditions as compared to controls (Fig. 2k–o). Altogether, our results in C2C12 myotubes show that S-SIT leads to similar RyR1 modifications to those observed in human muscle after SIT and

**Fig. 2 S-SIT in C2C12 myotubes induces RyR1 post-translational modifications, calstabin1 dissociation and decreased SR Ca$^{2+}$ content. a** S-MICT (**a$_1$**) and S-SIT (**a$_2$**) models in C2C12 myotubes. **b** Metabolomic analysis immediately after stimulation showing changes expressed in % of CTRL values; $n = 2$ independent biological experiments of six technical replicates each. **c–f** Representative immunoblots (**c**) and quantifications (**d–f**) of RyR1 immunoprecipitation and assessment of RyR1 post-translational modifications immediately after stimulation. DNP (2,4-dinitrophenylhydrazone): RyR1 oxidation. Cys NO: RyR1 nitrosylation. SK SR: skeletal muscle sarcoplasmic reticulum vesicle. SK SR treated with 200 μM H$_2$O$_2$ and 250 μM NOC-12: positive control for RyR1 oxidation, nitrosylation and calstabin1 dissociation. No antibody in IP: negative control; $n = 4$ independent biological experiments per group. One-way ANOVA followed by Tukey's multiple comparisons test and Kruskal–Wallis ANOVA, followed by Dunn's multiple comparison test (**e**). The whole gel and an additional control are shown in Supplementary Fig. 2g. **g–h** Original recording (**g**) and quantifications (**h**) of normalized Fluo-4 fluorescence in response to 2.5 mM caffeine immediately after S-MICT or S-SIT; $n = 7$ independent biological experiments per group. One-way ANOVA followed by Tukey's multiple comparisons test. **i, j** Fluorescence resonance energy-based assessment of SR Ca$^{2+}$ content (**i**) and quantifications of D1ER ratio (**j**) immediately after S-MICT and S-SIT (**i**). (**i$_1$**) D1ER plasmid distribution recorded at 535 nm. (**i$_2$**) Detail of the reticular pattern of the D1ER signal. (**i$_3$**) Ca$^{2+}$ distribution pattern is rendered in pseudocolor scale (from dark blue, low Ca$^{2+}$ level to red, high Ca$^{2+}$ level, see arrow); $n = 3$ independent biological experiments per group. One-way ANOVA followed by Tukey's multiple comparisons test. **k–l** Representative immunoblots (**k**) and quantification (**l**) of STIM1 after S-MICT and S-SIT; $n = 3$ independent biological experiments per group. One-way ANOVA. **m–o** Characterization of store-operated Ca$^{2+}$ entry (SOCE) in S-MICT and S-SIT myotubes immediately after stimulation. Original recording of SOCE (**m**); SR Ca$^{2+}$ store quantifications (**n**) and SOCE quantifications (**o**); $n = 3$ independent biological experiments per group. One-way ANOVA. Data are mean ± SD. *$p \leq 0.05$, **$p \leq 0.01$ and ***$p \leq 0.001$. Source data are provided as a Source Data file.

indicate a decreased SR Ca$^{2+}$ content in myotubes after a single S-SIT session, likely the result of Ca$^{2+}$ leak caused by dissociation of calstabin1 from RyR1.

**The leaky RyR1 status is associated with increased mitochondrial OXPHOS proteins expression, SC formation and NADH-linked respiratory capacity.** To investigate the link between RyR1 Ca$^{2+}$ leak and mitochondrial adaptations, we reduced RyR1 Ca$^{2+}$ leak with S107[52] in our S-SIT model and assessed the response of mitochondrial proteins. S107 acts by reinforcing the physical interaction between RyR1 and calstabin1, and thereby prevents/stops Ca$^{2+}$ leak[53]. We showed that 10 μM of S107 treatment in our S-SIT myotubes was sufficient to rebind calstabin1 to RyR1. S107 restored calstabin1 association with RyR1 after 3 h of treatment without affecting the RyR1 PTMs (Fig. 3a–d), in accordance with previous studies[11,52]. S107 treatment also rescued the Ca$^{2+}$ store decrease in S-SIT myotubes (Fig. 3e). Mitochondrial OXPHOS proteins were significantly increased at 72 h post stimulation in S-SIT myotubes, but not in S-MICT myotubes, and S107 treatment (applied immediately after the stimulation for 72 h) blunted these adaptations in S-SIT myotubes (Fig. 3f–k).

The ETS can be organized into super-assembled structures called SCs[54] and these directly improve metabolic capacity and efficiency[55]. Exercise training induces SCs formation in human skeletal muscle as an adaptive mechanism for increased energy demand[56]. We investigated the formation of SCs in S-SIT and S-MICT myotubes at 72 h after a single bout of stimulation. Both S-SIT and S-MICT increased the overall SC content and the formation of specific mitochondrial complexes (Fig. 3l). While both S-SIT and S-MICT induced the formation of complexes V$_n$ + III$_2$ + IV$_1$ and III$_2$ + IV$_2$ without changes in free complex II (Fig. 3n–p), blue native polyacrylamide gel electrophoresis (BN-PAGE) quantification revealed that S-SIT was more effective in assembling SCs primarily composed of complexes I + III + IV in C2C12 cells (Fig. 3l, m)[57].

As in human muscle, we also investigated mitochondrial biogenesis markers in our stimulated myotubes. The mRNA levels of PGC-1α, NRF1 and Tfam were significantly increased in myotubes immediately post S-SIT, but not post S-MICT (Supplementary Fig. 2h–j). Using the same antibodies as for the human samples, we observed significant increases in PGC-1α in S-SIT and S-MICT myotubes at 72 h post S-SIT, but to a greater extent in S-SIT myotubes (Supplementary Fig. 2k, m). This was also supported by similarly increased CaMKII phosphorylation in

both S-SIT and S-MICT cells (Supplementary Fig. 2l, n). NRF1 protein was significantly increased only in S-SIT myotubes (Supplementary Fig. 2o, p). Mitochondria staining with Mito-Tracker red revealed a significant increase in mitochondria area in S-SIT myotubes as compared to S-MICT and control myotubes (Fig. 3q, r). MitoTracker red is known to depend on mitochondrial membrane potential. We, therefore, excluded this potential confounding factor on the mitochondrial area by also using the MitoTracker green probe, which confirmed our results (Supplementary Fig. 2q–s). Importantly, these changes in S-SIT myotubes were blunted with S107 treatment (Fig. 3q, r and Supplementary Fig. 2q–s).

Changes in mitochondrial-associated gene/protein expression and enzyme activity assays provide static, surrogate measures for mitochondrial content and oxidative capacity, but do not reflect the complexity of mitochondrial function[58]. Studies describing mitochondrial protein content and function after intermittent high-intensity training are scarce[32]. We used high-resolution respirometry to investigate whether the above findings were accompanied by improved mitochondrial respiration (Supplementary Fig. 2t)[59]. Leak state (MP$_L$) assessed with malate and pyruvate (at non-phosphorylating state) showed a significant increase in S-SIT compared to S-MICT and control myotubes (Fig. 3s). OXPHOS state (N-linked pathway (N$_P$)), measured by adding saturating concentrations of ADP followed by glutamate, was significantly increased in S-SIT compared to control myotubes, but no effect was observed in S-MICT (Fig. 3t). When oxygen consumption was further increased by the addition of succinate to induce convergent electron transfer into the Q junction (NADH- and succinate-linked pathways, NS$_P$ state) S-SIT myotubes showed greater fluxes than S-MICT and control myotubes (Fig. 3u). When we assessed maximal ETS activity by the addition of the protonophore carbonyl-p-trifluoromethoxyphenylhydrazone (FCCP) (NS$_E$ state), the maximal respiration capacity was significantly increased in S-SIT myotubes only (Fig. 3v). Inhibition of NADH-linked respiration by rotenone highlighted the increase in non-phosphorylated succinate-linked pathway (S$_E$) (Fig. 3w). Importantly, S107 treatment specifically blunted the NADH-linked respiration pathway in S-SIT myotubes (Fig. 3t). These results show that a single session of S-SIT increases mitochondrial proteins and enhances mitochondrial respiratory function.

S-SIT S107 myotubes showed a slight but significant decrease in the total NADH levels compared to S-SIT myotubes at 72 h post stimulation (Fig. 3x), while the NAD and NAD/NADH levels were similar (Fig. 3y, z). Despite the higher ratio

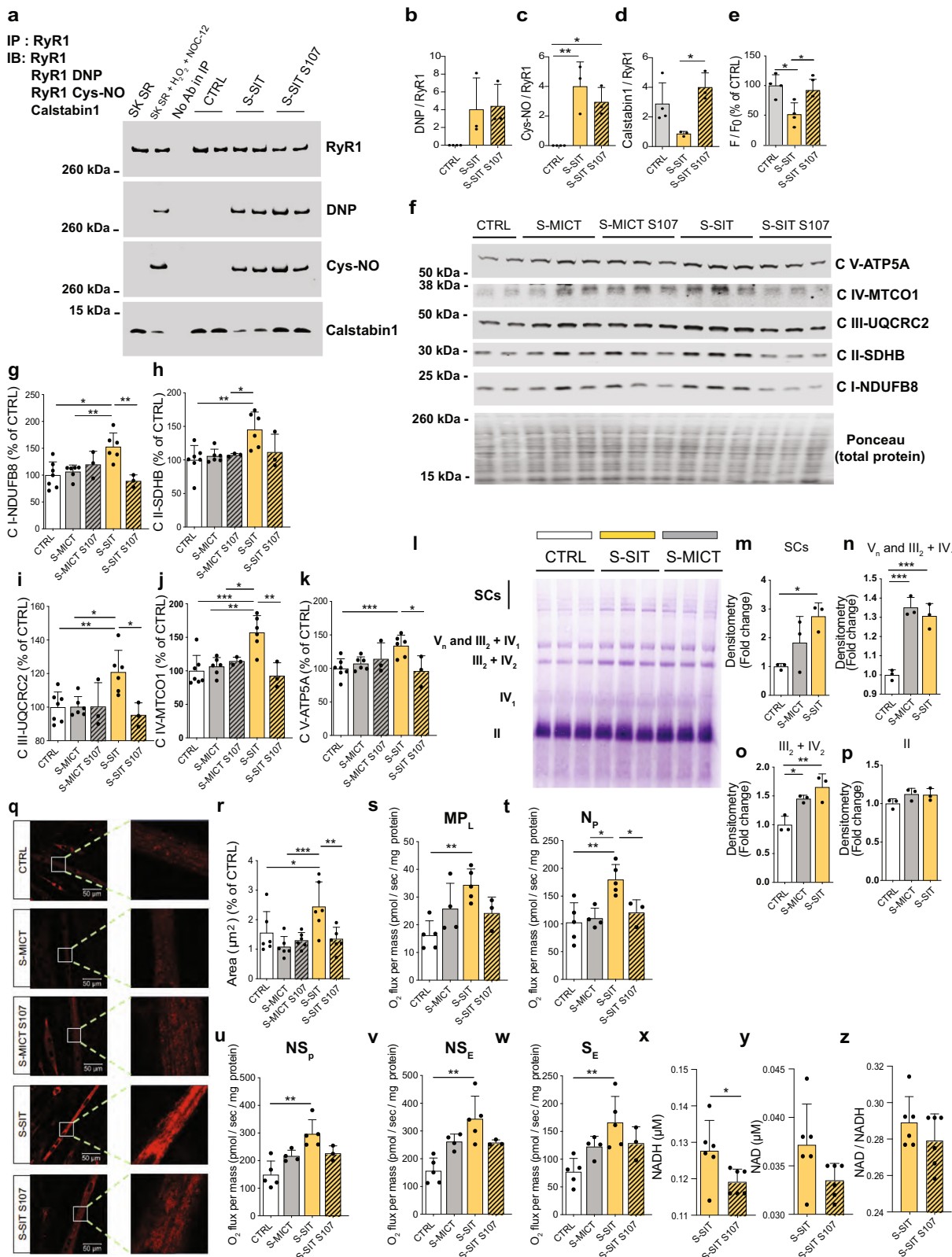

of NADH/NAD+ into the mitochondrial compartment compared to the cytosol, the observed alteration of NADH levels in our conditions should be interpreted as global and not mitochondria-specific.

Taken together, our results suggest a specific difference in the NADH-linked substrates respiratory rate between S-SIT and S-SIT S107 myotubes. This is further supported by the induction of the overall SCs, composed of I + III + IV SCs, in response to S-SIT (Fig. 3l, m). Altogether, our results demonstrate that the positive mitochondrial adaptations towards a more aerobic phenotype in response to S-SIT are driven, at least in part, by acute SR Ca2+ leak through RyR1/calstabin1 dissociation.

**Fig. 3 S-SIT in C2C12 myotubes induces higher mitochondrial protein content, supercomplexes levels and respiratory capacity as compared to MICT, which are blunted by S107-induced RyR1 stabilization. a–d** Representative immunoblots (**a**) and quantifications (**b–d**) of immunoprecipitated RyR1 post-translational modifications 3 h post stimulation without or with S107 treatment for 3 h; $n = 4$ (CTRL) and 3 (S-MICT, S-SIT) independent biological experiments. One-way ANOVA, followed by Tukey's multiple comparisons test (**c**, **d**). **e** Quantification of Fluo-4/AM fluorescence ratio 3 h post stimulation without or with S107; $n = 4$ (CTRL, S-SIT) and 3 (S-SIT S107) independent biological experiments. One-way ANOVA followed by Tukey's multiple comparisons test. **f–k** Representative immunoblots (**f**) and quantifications (**g–k**) of mitochondrial OXPHOS proteins 72 h post stimulation without or with S107 treatment for 72 h. Whole OXPHOS blot shown in Supplementary Fig. 6; $n = 7$ (CTRL), 6 (S-MICT, S-SIT) and 3 (S-MICT S107, S-SIT S107) independent biological experiments. One-way ANOVA followed by Tukey's multiple comparisons test. **l–p** Immunoblots (**l**) and quantification (**m–p**) of supercomplexes after stimulations; $n = 3$ independent biological experiments per group. One-way ANOVA followed by Tukey's multiple comparisons test (**m–o**). **q–r** MitoTracker red fluorescence (**q**) and quantification of mitochondrial area (**r**) 72 h post stimulation without or with S107 treatment; $n = 6$ independent biological experiments per group. One-way ANOVA with Sidak's multiple comparisons test. **s–w** $O_2$ flux per mass (pmol/s/mg of protein) 72 h after stimulation and S107 treatment. **s** $MP_L$: malate pyruvate leak state. **t** $N_P$: N-linked OXPHOS state with ADP-stimulated. **u** $NS_P$: N- and S-OXPHOS pathways. **v** $NS_E$: ET state, noncoupled and **w** $S_E$, S-pathway; $n = 5$ (CTRL, S-SIT), 4 (S-MICT) and 3 (S-SIT S107) independent biological experiments. One-way ANOVA with Sidak's multiple comparisons test. **x–z** NADH (**x**), NAD (**y**) concentration and NADH/NAD ratio (**z**) in S-SIT myotubes without or with S107 treatment for 72 h; $n = 6$ independent biological experiments per group. Unpaired $t$ test. For S107 groups, 10 µM of S107 treatment was applied immediately after stimulation for the indicated time. Data are mean ± SD. *$p \leq 0.05$, **$p \leq 0.01$ and ***$p \leq 0.001$. Source data are provided as a Source Data file.

**Reducing RyR1 Ca$^{2+}$ leak inhibits other physiological adaptations to S-SIT.** To further investigate the positive muscle metabolic adaptations mediated by acute Ca$^{2+}$ leak through RyR1, myotubes submitted to S-SIT were treated with or without S107 for 72 h following electrical stimulation, collected and then used for proteomic analyses. Proteins with significantly altered levels were submitted to pathway analysis. S-SIT S107-treated myotubes showed increased levels of proteins related to DNA and RNA processing (Supplementary Fig. 3a–c), while proteins of the following biological processes showed significant decreases: carbohydrate metabolism including glycolysis, Ca$^{2+}$ and other ion transport, muscle contraction and muscle fibre development, and mitochondrial function including the ETS (Fig. 4a–c). Glucose constitutes the primary source of energy used during SIT[60]. It is converted into pyruvate, which can be converted into lactate or into acetyl-CoA to enter the tricarboxylic acid (TCA) cycle in order to be fully metabolized aerobically. Given that S107 treatment was applied to the S-SIT myotubes after the electrical stimulation, the overall decrease of proteins involved in specific pathways in S-SIT S107 myotubes suggests that, in response to S-SIT, Ca$^{2+}$ leak through RyR1 triggers multiple physiological adaptations in the skeletal muscle, among which carbohydrate metabolism particularly stands out.

Inhibition of RyR1 Ca$^{2+}$ leak may also decrease SERCA-mediated Ca$^{2+}$ uptake and ATP utilization in the cytosol, and that could be a confounding factor for the observed mitochondrial changes in our S-SIT and S-SIT S107 myotubes. Sarcolipin (SLN) is a small molecule that is known to bind to SERCA inhibiting its activity. SERCA uses the energy derived from the hydrolysis of ATP to transport Ca$^{2+}$ ions across the SR membrane[61]. SLN binding to SERCA promotes the uncoupling of the SERCA pump and slippage of Ca$^{2+}$ into the cytoplasm instead of the SR lumen[62]. Expression of SERCA1 (the main isoform of SERCA expressed in C2C12 myotubes[63]) and its main regulator in the skeletal muscle, SLN[64], was not different between SIT and S-SIT S107 myotubes at 72 h post stimulation (Supplementary Fig. 4b, c). SERCA/SLN co-immunoprecipitation assay showed no significant difference between S-SIT and S-SIT S107 myotubes compared to controls (Supplementary Fig. 4a, d). However, others have shown that SLN decreases SERCA Ca$^{2+}$ uptake but does not alter ATP hydrolysis[65], thus implicating SLN as an uncoupler of SERCA (it continues to hydrolyse ATP but less Ca$^{2+}$ is transported to the SR lumen)[66]. We, therefore, prepared microsomes (SR-enriched fractions) (Supplementary Fig. 4e, f) to directly measure SERCA1 ATPase activity. Our results showed increased ATPase activity in S-SIT myotubes compared to control

and S-SIT S107 myotubes at 72 h post stimulation (Supplementary Fig. 4g, h), suggesting increased ATP utilization at S-SIT myotube SR. Interestingly, ATP levels in whole-cell lysates showed a similar increase in S-SIT and S-SIT S107 myotubes compared to controls (Supplementary Fig. 4i). The high levels of ATP in S-SIT myotubes, despite the increased ATP utilization at the SR, suggest a higher metabolic state in S-SIT myotubes compared to control and S-SIT S107 myotubes.

**S-SIT increases mitochondrial Ca$^{2+}$ content.** Ca$^{2+}$ is a well-known second messenger involved in many physiological processes in numerous tissues including the skeletal muscle[67,68]. Ca$^{2+}$ released through the SR can be taken up by mitochondria through the outer mitochondrial membrane via the voltage-dependent anion channel[69,70] to cross the inner mitochondrial membrane via the MCU[71]. Such mitochondrial Ca$^{2+}$ flux has been linked to improved mitochondrial bioenergetics[72]. We, therefore, investigated whether the Ca$^{2+}$ leak through RyR1 in response to S-SIT induced mitochondrial Ca$^{2+}$ uptake to trigger positive mitochondrial adaptations. We used the Rhod-2/AM probe and time-lapse confocal live imaging to investigate mitochondrial Ca$^{2+}$ uptake[14] in response to a single bout of S-SIT or S-MICT. Control myotubes loaded with Rhod-2/AM probe did not show any mitochondrial Ca$^{2+}$ increase in response to light exposure (Fig. 5a), while mitochondrial Ca$^{2+}$ significantly increased in both S-MICT and S-SIT myotubes (Fig. 5b, c, f). Mitochondrial Ca$^{2+}$ during S-MICT returned rapidly towards basal levels, while it remained elevated after S-SIT (Fig. 5b, c, g), suggesting Ca$^{2+}$ accumulation in the mitochondria after S-SIT. S-SIT myotubes pre-treated with 10 µM S107 did not show mitochondrial Ca$^{2+}$ accumulation (Fig. 5d), indicating that Ca$^{2+}$ leak through RyR1 is the main source of the observed mitochondrial Ca$^{2+}$ uptake after S-SIT. A similar result was obtained when S-SIT myotubes were pre-treated with 20 µM of the MCU inhibitor mitoxantrone (MTX)[73] (Fig. 5e, g). These results suggest that part of the Ca$^{2+}$ leaking through RyR1 is directed to the mitochondria through MCU, which supports the decreased SR Ca$^{2+}$ observed immediately after a SIT session (Fig. 2g–j).

To confirm S-SIT-induced mitochondrial Ca$^{2+}$ accumulation in mature skeletal muscle, we performed mitochondrial Ca$^{2+}$ uptake measurements in dissociated mouse FDB muscle fibres. We applied S-MICT and S-SIT to FDB muscle fibres and followed mitochondrial Ca$^{2+}$ accumulation after the end of the stimulation as in the C2C12 myotubes. Control muscle fibres showed no increase in mitochondrial Ca$^{2+}$ with light exposure (Fig. 5h). Like in C2C12 myotubes,

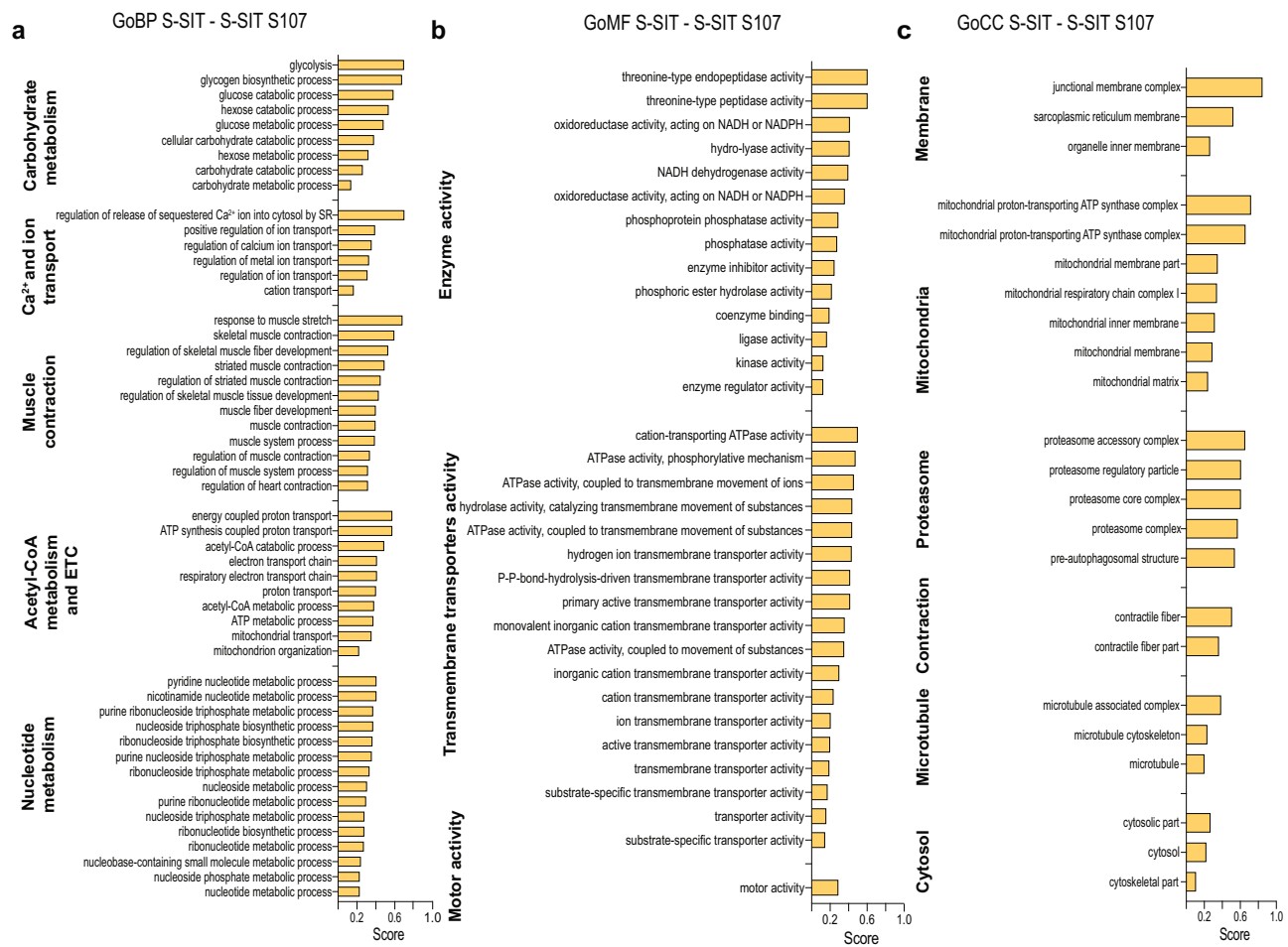

**Fig. 4 Proteomic analysis showing the global mapping of RyR1 Ca²⁺ leak on muscle adaptations to S-SIT. a–c** Proteomic analysis of protein groups related to Gene Ontology Biological Processes (GoBP), Molecular Function (GoMF) and Cellular Component (GoCC) that are significantly decreased S-SIT after 10 μM S107 treatment for 72 h (the treatment was applied immediately after stimulation). Protein groups exceeding 400 proteins were excluded. The median values of S-SIT − S-SIT S107 difference were calculated, and a score affected to the amplitude of the difference. The positive scores display the pathways significantly inhibited by S107 treatment. $n = 5$ per group. Benjamini–Hochberg corrected $t$ test. Source data are provided as a Source Data file.

S-MICT and S-SIT resulted in altered mitochondrial Ca²⁺ transients in FDB muscle fibres (Fig. 5i, j, k). Again, the amount of Ca²⁺ taken up by the mitochondria at the end of the S-SIT session was higher than at the end of the S-MICT session (Fig. 5l).

We then examined whether RyR1 Ca²⁺ leak elicits enough Ca²⁺ domains to trigger mitochondrial Ca²⁺ uptake through the MCU in our in vitro models. Mitochondrial Ca²⁺ uptake is known to occur in specific conditions, which require local microdomains of elevated Ca²⁺ between the SR and the mitochondria[74]. Despite the consensus that MCU needs a Ca²⁺ concentration in the micromolar range, there is evidence that mitochondria also can take up Ca²⁺ at nanomolar concentrations[75,76]. We hypothesized that such Ca²⁺ domains are formed in response to leaky RyR1 and can trigger mitochondrial Ca²⁺ uptake. We first checked the pattern of RyR1 opening-induced mitochondrial Ca²⁺ accumulation in our myotubes by treating them with 2.5 mM caffeine while monitoring mitochondrial Ca²⁺ with the Rhod-2/AM probe. Caffeine induced a rapid, large mitochondrial Ca²⁺ increase, which progressively decreased, upon which the addition of 10 μM of rapamycin (which dissociates calstabin1 from RyR1[14]) induced a small additional mitochondrial Ca²⁺ uptake (Supplementary Fig. 4j). When the myotubes were first treated with 10 μM rapamycin, they also showed a rapid mitochondrial Ca²⁺ increase, but with a lower amplitude than that of caffeine

treatment (Supplementary Fig. 4k). This mitochondrial Ca²⁺ uptake showed a progressive increase that reached a plateau, and the addition of caffeine elicited no further increase in mitochondrial Ca²⁺ (Supplementary Fig. 4k). These results suggest that rapamycin-induced leaky RyR1 is sufficient to elicit a progressive and sustained mitochondrial Ca²⁺ uptake. Overall, our observations in C2C12 myotubes and mouse FDB fibres indicate that, in response to S-SIT, a leaky RyR1 channel leads to Ca²⁺ uptake by the mitochondria. While electrical stimulation is known to elicit mitochondrial Ca²⁺ uptake[72], we here show a specific mitochondrial Ca²⁺ accumulation following S-SIT, after the electrical stimulation ended, which suggests a role of RyR1 Ca²⁺ leak rather than Ca²⁺ release in response to electrical stimulation.

**Mitochondrial Ca²⁺ uptake induced by RyR1 Ca²⁺ leak dephosphorylates PDH and underpins OXPHOS complex I increase in response to S-SIT.** Several mitochondrial enzymes are Ca²⁺ sensitive. Among them, PDH is a gateway enzyme for carbohydrate-derived pyruvate entry into the TCA cycle for complete oxidation[77]. PDH catalyses pyruvate decarboxylation to acetyl-CoA, and the reaction leads to a reduction of NAD⁺ to NADH. PDH activity is covalently regulated by phosphorylation on four residues of the PDH E1 subunit: Ser²⁹³, Ser²⁹⁵, Ser³⁰⁰ and Ser²³². PDH kinases inactivate PDH by phosphorylation, whereas

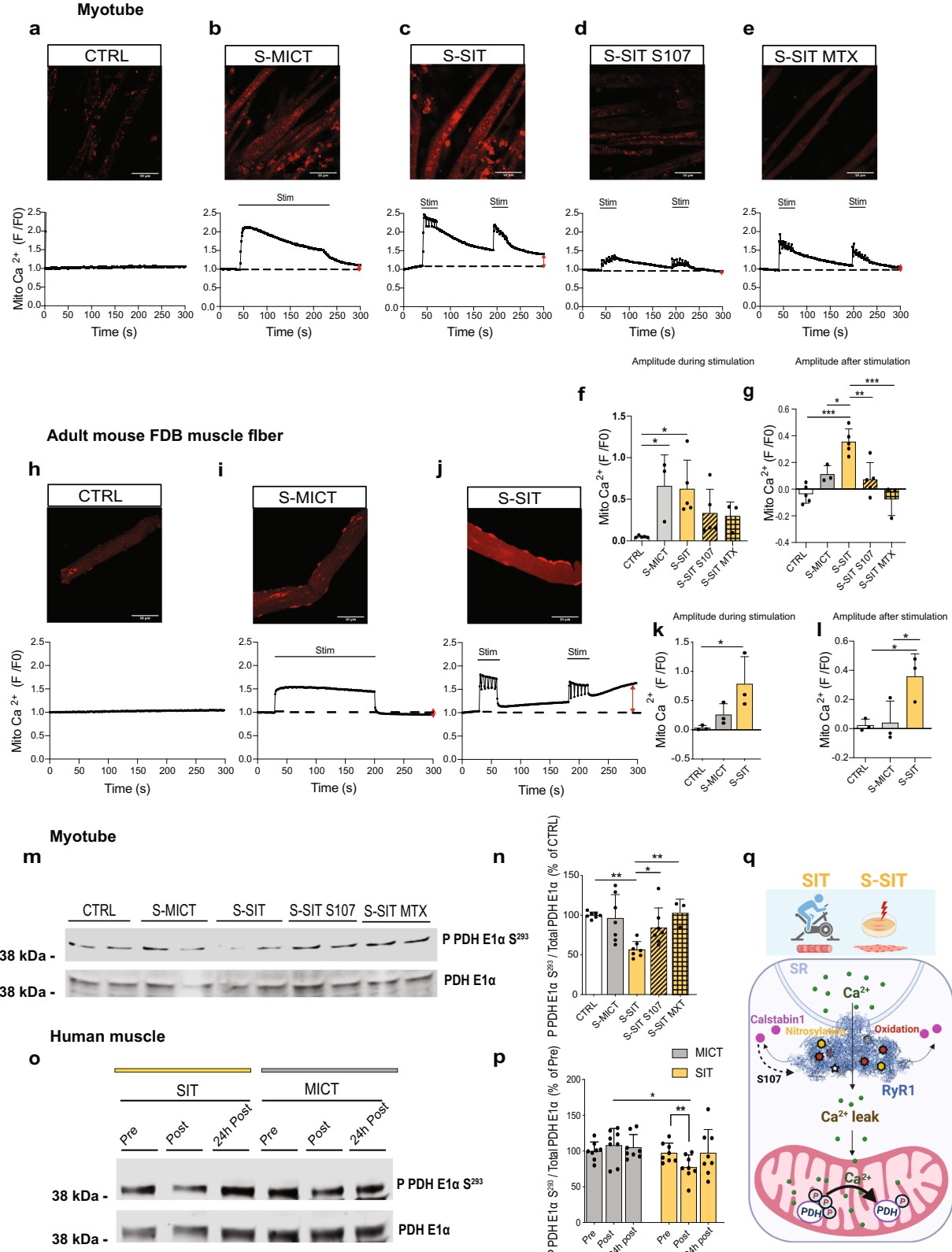

PDH phosphatases activate PDH by dephosphorylation[78]. PDH activity is regulated in human muscle in response to exercise[79]. The following arguments prompted us to investigate PDH phosphorylation in response to SIT: (i) skeletal muscle PDH activity is dependent on exercise intensity[80] and MCU$^{-/-}$ mice showed marked impairment in their ability to perform strenuous work[81]; (ii) MCU knockout induced an increase in PDH phosphorylation similarly to PDH phosphatases knockout linking mitochondrial Ca$^{2+}$ uptake to PDH phosphorylation levels[82].

We, therefore, measured PDH phosphorylation levels as a readout of the increased mitochondrial Ca$^{2+}$ uptake in S-SIT myotubes. PDH phosphorylation level on Ser$^{293}$ (P PDH E1α Ser$^{293}$) was significantly decreased in S-SIT myotubes at 1 h post stimulation, but not in S-MICT myotubes (Fig. 5m, n), reflecting

**Fig. 5 (S-)SIT-induced leaky RyR1 increases mitochondrial $Ca^{2+}$ uptake and decreases PDH phosphorylation levels in muscle cells. a–e** Normalized Rhod-2 fluorescence imaging in C2C12 myotubes. A 300 s time-lapse confocal recording of a bout of slightly modified S-SIT and S-MICT protocols: **b** S-MICT and **c** S-SIT. **d**, **e** Mitochondrial $Ca^{2+}$ uptake in S-SIT myotubes pre-treated with 10 µM S107 for 1 h (**d**) or 20 µM mitoxantrone (MTX) (**e**). Horizontal black lines indicate the periods of stimulation; vertical red arrows indicate the amplitude of mitochondrial $Ca^{2+}$ at the end of the recordings. **f–g** Maximal amplitude of normalized Rhod-2 fluorescence during stimulations (**f**) and 80 s after the end of S-MICT and S-SIT stimulations (**g**) in (**a–e**); $n = 5$ (CTRL, S-SIT and S-SIT S107) and 3 (S-MICT, S-SIT MTX) independent biological experiments. One-way ANOVA followed by Tukey's multiple comparisons test. **h–j** Normalized Rhod-2 fluorescence imaging in mouse FDB intact single muscle fibres. Same protocols as described for (**a–c**), except that the voltage was 40 V. **k–l** Maximal amplitude of normalized Rhod-2 fluorescence in FDB muscle fibres in (**h–j**) during (**k**) and 80 s after the end of (**l**) of S-MICT and S-SIT stimulations; $n = 3$ mice per group. One-way ANOVA followed by Tukey's multiple comparisons test. **m–n** Representative immunoblots (**m**) and quantification (**n**) of phosphorylated PDH E1α at serine 293 related to total PDH E1α in myotubes 1 h after stimulation; 10 µM S107 or 20 µM MTX were applied after the stimulation for 1 h when indicated; $n = 7$ (CTRL, S-MICT, S-SIT and S-SIT S107) and 3 (S-SIT MTX) independent biological experiments. One-way ANOVA followed by Sidak's multiple comparisons test. **o**, **p** Representative immunoblots (**o**) and quantification (**p**) of phosphorylated PDH E1α at serine 293 related to total PDH E1α in human muscles; $n = 8$ participants per group. Two-way ANOVA followed by Sidak's multiple comparisons test. **q** Proposed schematic of RyR1 $Ca^{2+}$ leak-activated mitochondrial PDH E1α dephosphorylation in response to S-SIT. Data are mean ± SD. $^{*}p \leq 0.05$, $^{**}p \leq 0.01$ and $^{***}p \leq 0.001$. Source data are provided as a Source Data file.

SIT-induced mitochondrial $Ca^{2+}$ uptake and PDH activation. PDH dephosphorylation was also observed in our human muscle biopsies collected post SIT—but not MICT (Fig. 5o, p), indicating again that our S-SIT model mirrors the effects of a single SIT session in human muscle. To investigate the potential role of RyR1 $Ca^{2+}$ leak in the process, we treated S-SIT myotubes with 10 µM S107 immediately after the stimulation and for 1 h before measurement of PDH phosphorylation levels. The decreased PDH phosphorylation was blunted (Fig. 5m, n), pointing to a causal role of RyR1 $Ca^{2+}$ leak in PDH dephosphorylation in response to S-SIT. S-SIT myotube treatment with 20 µM MTX immediately after the stimulation and for 1 h also restored PDH phosphorylation levels, confirming the role of mitochondrial $Ca^{2+}$ uptake in the process (Fig. 5m, n). It follows that SIT induces RyR1 PTMs and calstabin1 dissociation from the RyR1, which leads to a leaky RyR1, mitochondrial $Ca^{2+}$ uptake and PDH dephosphorylation (Fig. 5q).

We finally investigated whether RyR1 $Ca^{2+}$ leak-induced mitochondrial $Ca^{2+}$ uptake plays a role in the late (72 h post) mitochondrial adaptations in response to S-SIT. For this purpose, we transfected S-SIT myotubes with small interfering RNAs (siRNAs) directed against MCU immediately after the stimulation and measured OXPHOS proteins 72 h later. We first confirmed that the si-MCU was effective to decrease MCU expression and mitochondrial $Ca^{2+}$ uptake (Fig. 6a, b) in differentiated myotubes without altering the expression of the main proteins involved in SR $Ca^{2+}$ transients (RyR1, SERCA, and calstabin1) (Fig. 6a). S-SIT myotubes transfected with the si-MCU immediately after the stimulation showed decreased MCU protein levels 72 h after the stimulation as compared to scrambled siRNA transfection (si-CTRL and S-SIT si-CTRL) (Fig. 6d). S-SIT si-CTRL myotubes showed significantly increased OXPHOS CI and IV proteins, whereas S-SIT si-MCU specifically prevented OXPHOS CI modifications (Fig. 6c, e–i). Since we downregulated MCU after S-SIT, our results suggest that mitochondrial $Ca^{2+}$ uptake after completion of S-SIT (which we linked to the leaky RyR1) is responsible for the increase in mitochondrial OXPHOS CI expression. These results are in agreement with recent work showing an association between decreased MCU levels and a specifically decreased OXPHOS CI protein in cardiac tissues obtained from Barth syndrome patients[83]. Altogether, these observations point to a new role of MCU and mitochondrial $Ca^{2+}$ in the regulation of OXPHOS CI that opens an area for further investigations.

## Discussion

We report RyR1 PTMs followed by calstabin1 dissociation in human muscle in response to a single session of SIT—but not MICT, and in C2C12 myotubes submitted to S-SIT and not S-MICT. This was associated with greater mitochondrial adaptations in human SIT muscle and in S-SIT myotubes. Our in vitro models allowed us to causally link RyR1 $Ca^{2+}$ leak to mitochondrial remodeling and improved mitochondrial function, since the latter were blunted by the RyR stabilizer S107. In healthy individuals, intense exercise-induced acute RyR1 $Ca^{2+}$ leak in muscle can therefore be considered beneficial since it triggers mitochondrial proteins increase. This places acute RyR $Ca^{2+}$ leak in conditions of health apart from sustained RyR $Ca^{2+}$ leak observed in pathological conditions.

In a previous study, we reported RyR1 fragmentation in response to a single session of SIT in humans, which presumably led to a leaky RyR1[26]. Follow-up research suggested that RyR1 fragmentation depends on subject susceptibility and training status, bringing the notion of responders and non-responders[28]. Here, we focused on calstabin1 dissociation from the RyR1 as it is a strong and consistent signature of leaky RyR1 channels[53]. Our working hypothesis was that SIT-induced acute RyR1 $Ca^{2+}$ leak caused muscle mitochondrial remodeling leading to improved respiratory function. Our results mechanistically suggest that $Ca^{2+}$ leak through RyR1 in response to S-SIT is taken up by the mitochondria and triggers the dephosphorylation of PDH as (i) S-SIT myotubes accumulate more mitochondrial $Ca^{2+}$ as compared to S-MICT myotubes, a phenotype abolished by S107 treatment. In addition, (ii) S-SIT myotubes and human SIT muscle showed less PDH phosphorylation as compared to S-MICT and MICT, respectively, and (iii) S107 treatment and MCU inhibition with MTX restored PDH phosphorylation levels in S-SIT. In the skeletal muscle, the SR and mitochondria are spatially coupled[69] and $Ca^{2+}$ released from the SR can be taken up by the mitochondria[72,84,85]. Muscle cell depolarization enhances mitochondrial $Ca^{2+}$ uptake. This is linked to mitochondrial bioenergetics since RyR1 inhibition during muscle cell depolarization decreases ATP-linked $O_2$ consumption[72]. In both our S-MICT and S-SIT models, mitochondrial $Ca^{2+}$ amplitude increased during electrical stimulation and then decreased after the stimulation. While the decrease was almost complete in S-MICT, S-SIT myotubes showed a small but sustained mitochondrial $Ca^{2+}$ plateau that could not be linked to muscle cell depolarization, and which likely activated the mitochondrial $Ca^{2+}$-dependent enzyme PDH. The early PDH dephosphorylation observed after S-SIT could not be directly linked to the late mitochondrial adaptations (increased OXPHOS proteins and respiration after 72 h), especially as our respiration data were collected using saturating substrate conditions. However, a role of RyR1 $Ca^{2+}$ leak in mitochondrial NADH-linked adaptations is supported by (i) our proteomic data showing alteration of protein groups linked to PDH activity upon treatment with S107

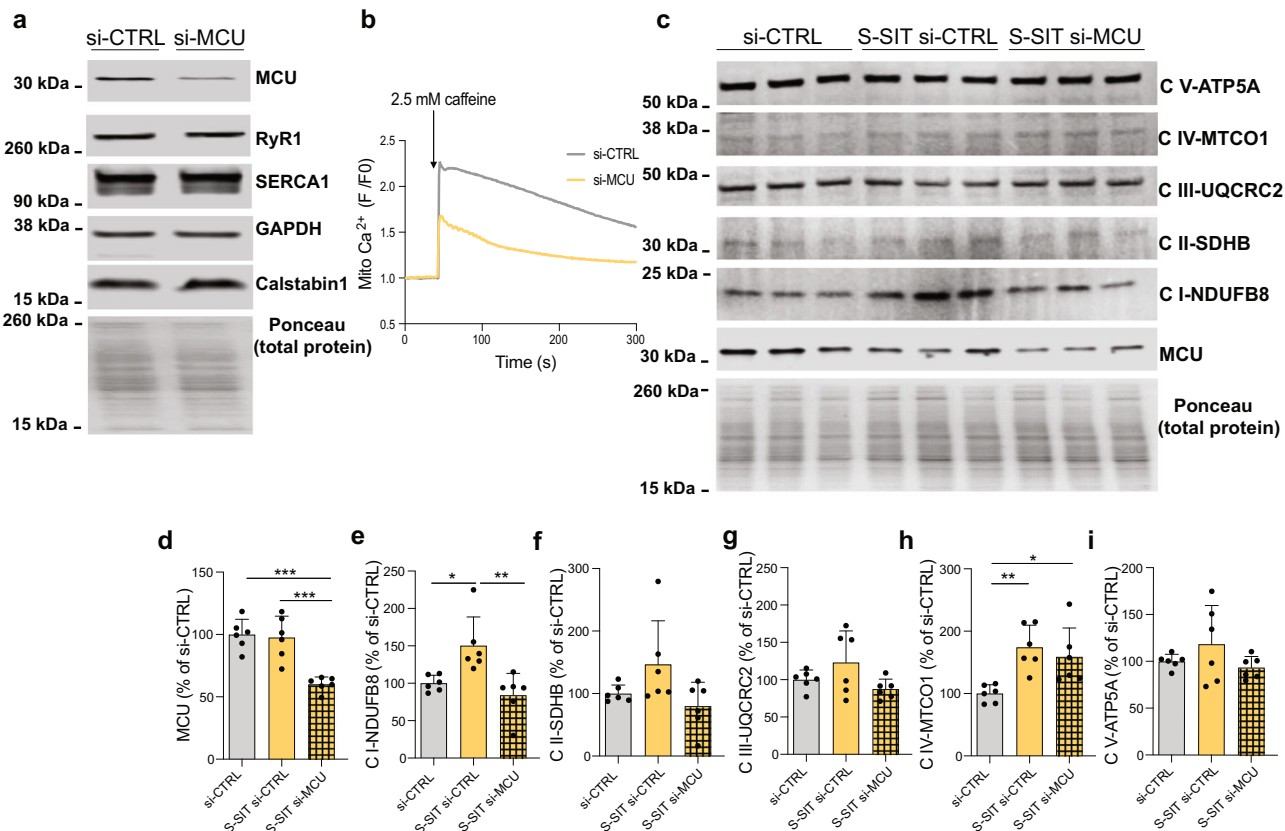

**Fig. 6 Mitochondrial Ca$^{2+}$ uptake contributes to mitochondrial adaptations to S-SIT. a** Immunoblots of MCU, RyR1, SERCA1, GAPDH and calstabin1 in si-MCU (siRNAs directed against MCU) compared to si-CTRL (negative control siRNAs) myotubes. **b** Original recordings of normalized Rhod-2 fluorescence imaging in si-MCU compared to si-CTRL myotubes at 72 h post transfection. **c** Representative immunoblots of mitochondrial OXPHOS and MCU proteins expression in S-SIT si-MCU myotubes at 72 h post stimulation (cells were transfected immediately after the stimulation with the siRNAs against MCU) compared to S-SIT si-CTRL myotubes (cells were transfected immediately after the stimulation with the negative control siRNAs) and si-CTRL (non-stimulated myotubes transfected with the negative control siRNAs). All the cropped parts of OXPHOS proteins are part of the same blot that is shown in Supplementary Fig. 6. **d–i** Quantifications of the immunoblots in (**c**) related to total protein and expressed as % of si-CTRL; $n = 6$ independent biological experiments per group. One-way ANOVA followed by Tukey's multiple comparisons test (**d**, **e**, **h**). Data are mean ± SD. *$p \leq 0.05$, **$p \leq 0.01$ and ***$p \leq 0.001$. Source data are provided as a Source Data file.

(Supplementary Data 1 and 2) and (ii) decreased NADH levels in S-SIT S107 myotubes. MCU downregulation in myotubes blunted the increase in OXPHOS CI proteins (NDUFB8) in response to S-SIT, also pointing to a specific role of RyR1 Ca$^{2+}$ leak in S-SIT-induced adaptations. These observations combined with the S-SIT S107 data strongly support a beneficial role of RyR1 Ca$^{2+}$ leak and mitochondrial Ca$^{2+}$ uptake in the mitochondrial adaptations in response to S-SIT.

Jain et al.[86] previously reported that high-fat feeding increased mitochondrial ROS production triggering RyR1 S-nitrosylation and CaMK phosphorylation in the muscle cell, which was associated with increased mitochondrial biogenesis. We found different patterns of RyR1 PTMs but similar CaMK phosphorylation levels in response to SIT and MICT. During muscle contraction, the large Ca$^{2+}$ release from the SR strongly and quickly induces CaMK phosphorylation and activity[87], which supports the similar increase in CaMKII phosphorylation levels in SIT and MICT. Moreover, Wright et al.[38] previously reported a CaMK-dependent p38 activation and PGC-1α expression in response to rising cytosolic Ca$^{2+}$. This also supports our data showing similar increases in CaMKII phosphorylation levels and PGC-1α expression in response to SIT and MICT. Despite these similarities between SIT and MICT, our study points to a specific causality between RyR1 Ca$^{2+}$ leak-induced mitochondrial Ca$^{2+}$ uptake and activation of NADH-linked mitochondrial adaptations in response to a single session of SIT.

It has been advanced that HIIT-induced mitochondrial adaptations would require several days/bouts of exercise to induce beneficial adaptions[88]. In a recent study, Skelly et al. reported acute induction of mitochondrial biogenesis markers in SIT and MICT human muscles fibres irrespective to muscle fibre type, while a long-term training revealed differential muscle fibre-dependent adaptations[89]. An independent study by Trewin et al.[31] reported changes in mitochondrial protein levels and mitochondrial respiration in response to a single session of HIIT and MICT, despite a lower workload in HIIT. Moreover, a rapid PGC-1α protein nuclear localization has been reported in response to SIT compared to MICT[90]. In our study, although both SIT and MICT showed increases in PGC-1α protein in human muscle and cells, only SIT showed significant increases in OXPHOS CI and CII proteins. As PGC-1 nuclear localization is important for its function, this may be differently regulated in SIT and MICT. Furthermore, whether the acute mitochondrial inductions we observed would translate into long-term adaptations remains unknown. Several studies have compared metabolic adaptations to several weeks of SIT vs. MICT and have reported similarly increased aerobic capacity (VO$_2$max) and mitochondrial content[23,91,92]. However, the cellular stress and the resulting metabolic signals for mitochondrial biogenesis were shown to depend on exercise intensity, with limited work suggesting that the increases of mitochondrial content are superior after SIT

compared to MICT, at least when matched-work comparisons are made within the same individuals[91].

Regular exercise leads to widespread changes in numerous cells, tissues and organs (e.g. skeletal and cardiac muscle, brain, liver, fat) conferring multiple health-promoting benefits[17]. Physical fitness level in healthy middle-aged men is a strong predictor of mortality and small improvements in physical fitness are associated with a significantly lowered risk of mortality[16].

Even though today's recommendations suggest that any episode of any physical activity conveys benefits[17], it is important to determine which time-efficient combinations of exercise (with different type, volume and intensity) are optimal to obtain a given outcome as the response might differ according to the fitness status. For maximal health benefits, especially for healthy fit people, even if the means towards that goal imply regular strenuous physical activity, it should perhaps be not too hard[93]. A recent human study compared cardio-metabolic health parameters and muscle adaptations upon 12 weeks of HIIT, resistance training or a combination of both. Only HIIT improved cardio-metabolic health parameters, while both HIIT and combined training increased aerobic capacity and skeletal muscle mitochondrial respiration[22]. Proteomics showed increased proteins groups of the electron transport chain, OXPHOS, TCA cycle and glycolysis after HIIT training[22].

Using a maximum intensity variant of interval training, we found that inhibiting RyR1 $Ca^{2+}$ leak with S107 after a single session of S-SIT decreases protein groups involved in glycolysis, acetyl-CoA metabolism and electron transport chain, muscle contraction and ion (including $Ca^{2+}$) transport in S-SIT myotubes, thus revealing the importance of RyR1 $Ca^{2+}$ leak in the mechanisms triggering improved muscle mitochondrial function in response to exercise. This suggests that triggering acute RyR1 $Ca^{2+}$ leak causes mitochondrial remodeling, but may also play a key role in the myriad of other adaptations to exercise training in the muscle.

This study also has some limitations. In order to mechanistically discover the causal link between RyR1 $Ca^{2+}$ leak and muscle beneficial adaptations in response to SIT, we developed in vitro SIT and MICT models to generate data that were combined with observations made on humans. Since the focus of this study is mitochondrial adaptations linked to $Ca^{2+}$ leak through the RyR1, the in vitro models were designed to appropriately reflect the main observed changes in human muscle. While the combined models helped to decipher the mechanisms of the mitochondrial adaptations triggered by RyR1 $Ca^{2+}$ leak, our in vitro models do not properly represent actual SIT and MICT models in humans. Moreover, our results do not rule out that other mechanisms contribute to the observed adaptations as $Ca^{2+}$ may not be the only mechanism involved in the process.

In conclusion, SIT is a time-efficient exercise intervention that can counteract the effects of physical inactivity, one of the top health risk factors worldwide. In the present study, we contribute to the understanding of the molecular mechanisms underlying its beneficial effects in muscle. We draw two main conclusions: (i) high-intensity exercise triggers RyR1 post-translational changes leading to an acute leaky state and (ii) this is causally linked to improved mitochondrial remodeling and function of the activated muscle. On a conceptual level, we conclude that acute RyR1 $Ca^{2+}$ leak in muscle can be considered beneficial and shows clear differences with the sustained RyR $Ca^{2+}$ leaks observed in pathological conditions. These original mechanistic observations pave the way for new research to further investigate the role of acute RyR $Ca^{2+}$ leak in other physiological conditions. Whether such acute RyR $Ca^{2+}$ leak may also play a beneficial role in pathological conditions remains to be investigated.

## Methods

**Human experiments: general procedures.** The human general procedures were followed as previously described[26] with slight modifications. The study was validated by the Commission d'éthique de la recherche sur l'être humain du Canton de Vaud (protocol 2017-00303) and performed in accordance with the Declaration of Helsinki. Sixteen male recreationally active subjects gave written informed consent before participation. The participants refrained from physical activity and caffeine consumption for 24 and 12 h before the experimental session, respectively. Participants were familiarized with electrical stimulation and voluntary contraction procedures at least 48 h before the first experimental session. In the familiarization session, they performed an incremental test to exhaustion on a cycle ergometer (Lode Excalibur Sport, Lode, Groningen, Netherlands). The test started at a power of 1 W and was increased by 1 W every 2 s. The participants were instructed to maintain a cadence between 60 and 80 r.p.m. and when they were unable to maintain 60 r.p.m. the test was stopped. The mean of the $VO_2$ values in the last 30 s of the test was used to determine $VO_2peak$. Gas exchange was measured with a stationary gas analyser (Quark CPET, COSMED, Rome, Italy). The participants ($n = 8$ per group) were allocated into two groups (SIT and MICT) based on their $VO_2peak$ so that mean $VO_2peak$ was similar for each group (Supplementary Fig. 1a).

MICT consisted of 1 h of cycling on a cycle ergometer at 65% of the maximal aerobic power reached during the incremental $VO_2max$ test. Each experiment was preceded by a standard warm-up on the cycle ergometer (5 min at 100 W). SIT was comprised of 30 s all-out cycling bouts at 0.7 N.m/kg body mass on a cycle ergometer (Lode Excalibur Sport, Lode, Groningen, Netherlands), with 4 min recovery periods (rest) between bouts.

Knee extensor neuromuscular function of the right (dominant) leg was tested before (Pre), immediately post (Post) and 24 h after (24 h Post) exercise. The tests consisted of a 5 s MVC with a superimposed 100 Hz doublet (paired stimuli) evoked via supramaximal electrical stimulation of the femoral nerve (twitch interpolation technique), followed by supramaximal stimulations of a relaxed muscle evoked at 2 s intervals: a doublet at 100 Hz, 10 Hz, and a single stimulus to obtain the compound muscle action potential (M-wave).

Our study complies in full with the STROBE statement. The schematic of Figs. 1a, 5q are made using the BioRender application: https://app.biorender.com.

**Muscle biopsies.** Needle biopsies were taken from the left (non-dominant leg) *vastus lateralis* muscle before, ~10 min and 24 h after exercise, using previously described and validated procedures[94]. Briefly, after skin sterilization and local anaesthesia, a 1–2-mm-long skin cut was made with the tip of a scalpel. Biopsies were collected using an automatic biopsy device (Bard Biopsy Instrument, Bard Radiology, Covington, GA, USA). A 14-gauge disposable trocar mounted in the device was inserted through the cut, perpendicular to the muscle fibres, until the fascia was pierced. Three samples (~15 mg each) were collected from one puncture site at each time point. Muscle samples were immediately frozen in liquid nitrogen and stored at −80 °C until analysis.

**C2C12 cell culture.** C2C12 mouse skeletal myoblasts were obtained from the American Type Culture Collection and grown in a proliferation medium (PM) composed of Dulbecco's modified Eagle's medium (DMEM; Thermo Fisher Scientific, Basel, Switzerland) supplemented with 10% foetal bovine serum (FBS) (Thermo Fisher Scientific, Basel, Switzerland), 100 IU/ml penicillin, 100 µg/ml streptomycin (Thermo Fisher Scientific, Basel, Switzerland) and 1% non-essential amino acids (Thermo Fisher Scientific, Basel, Switzerland), and maintained at 37 °C in a humidified atmosphere with 5% $CO_2$. To induce differentiation, myoblasts were grown to 80–90% confluence, the proliferation medium was then replaced with a differentiation medium (DM), consisting of DMEM supplemented with 2% horse serum (Thermo Fisher Scientific, Basel, Switzerland).

**Electrical stimulation of C2C12 myotubes.** Well-differentiated C2C12 myotubes (day 6 or 7 post differentiation) in 6-well plates (Corning, NY, USA) containing 4 ml of DM were electrically stimulated (C-Pace EM stimulator, IONOPTIX LLC, MA, USA) as follows: 1 h repeated stimulation at 14 V, 2 Hz and 2 ms pulse duration (S-MICT) or 6 × 30 s pulse (5 s on, 1 s off) at 14 V, 50 Hz and 2 ms pulse duration with 4 min rest (S-SIT) (Fig. 2a).

The DM was replaced before and after the electrical stimulation and the cells were harvested at the appropriate time point. Different treatment conditions from the same plate per independent culture were used and the different wells from the same condition per culture were considered as technical replicates.

**Myotube siRNA transfection.** C2C12 myotubes at day 6 of differentiation were electrically stimulated using the S-SIT protocol. Stimulated myotubes were immediately transfected with 50 nM final concentration of silencer select negative control siRNAs (si-CTRL) (Thermo Fisher Scientific, Basel, Switzerland) or silencer select pre-designed siRNAs directed against MCU (si-MCU) (Thermo Fisher Scientific, Basel, Switzerland) using Lipofectamine RNAimax protocol according to the manufacturer (Thermo Fisher Scientific, Basel, Switzerland). The myotubes were collected at day 3 post transfection for western blot analysis.

**Animals.** *C57BL/6J* (B6) mice obtained from Janvier were provided by the animal facility from the Department of Biomedical Sciences, University of Lausanne. The animals were maintained in a temperature-controlled animal facility with a 12-h light/12-h dark cycle and had access to food and water according to the Swiss Animal Protection Ordinance (OPAn). The 8-week female mice were used in this study. Our protocol was approved by the Animal Ethics Committee of Lausanne (commission cantonale pour l'expérimentation animale) with the number VD3489.

**Mouse FDB muscle fibre dissociation.** The FDB muscles were isolated from 8-week female C57BL/6J (B6) mice and incubated for 38 min at 37 °C in an oxygenated Krebs-HEPES solution (in mM: NaCl 135.5, $MgCl_2$ 1.2, KCl 5.9, glucose 11.5, HEPES 11.5, $CaCl_2$ 1.8, final pH 7.3) containing 0.2% collagenase type IV (Sigma-Aldrich Corp., St. Louis, MO, USA). Muscles were then washed twice in DMEM/Ham's F12 (Sigma-Aldrich Corp., St. Louis, MO, USA) supplemented with 2% FBS (Sigma-Aldrich, St. Louis, MS, USA) and mechanically dissociated by repeated passages through fire-polished Pasteur pipettes of progressively decreasing diameter. For $Ca^{2+}$ measurements, dissociated fibres were plated on 35-mm-diameter glass-bottom MatTek dishes (MatTek, Ashland, MA, USA) coated with Matrigel (BD Bioscience, San Jose, CA, USA). Culture dishes were kept in an incubator, with 5% $CO_2$ at 37 °C for 2 h to allow the fibres to attach[95].

**Cytosolic $Ca^{2+}$ imaging using Fluo-4 AM in C2C12 myotubes.** C2C12 myoblasts were plated on poly-D-lysine-coated 35-mm-diameter glass-bottom MatTek dishes (MatTek, Ashland, MA, USA). At 80% of confluence, the myoblasts were differentiated by replacing the PM with DM and myotubes were used at day 5–7 of differentiation. Myotubes were stimulated using simulated MICT or SIT protocols (S-MICT and S-SIT); then, at the indicated time of measurement, they were loaded with the cytosolic $Ca^{2+}$ indicator Fluo-4 AM (5 μM, Invitrogen, Basel, Switzerland) solubilized in a Krebs solution (in mM: NaCl 135.5, $MgCl_2$ 1.2, KCl 5.9, glucose 11.5, HEPES 11.5, $CaCl_2$ 1.8, final pH 7.3) for 20 min in the incubator. Cells were then rinsed twice with a $Ca^{2+}$-free Krebs solution (in mM: NaCl 135.5, $MgCl_2$ 1.2, KCl 5.9, glucose 11.5, HEPES 11.5, 200 μM Na-EGTA, final pH 7.3). Fluo-4 fluorescence was monitored using a confocal microscope system (Zeiss LSM 5 Live, Oberkochen, Germany; ×40 oil immersion lens; the excitation wavelength was 488 nm and the emitted fluorescence was recorded between 495 and 525 nm). After recording the basal fluorescence, myotubes were stimulated with 2.5 mM (final concentration) caffeine to trigger $Ca^{2+}$ release from the SR. Zen software 2012 version (Zeiss, Oberkochen, Germany) was used for the acquisition and data were exported to excel files for analysis. The use of the single excitation/emission Fluo-4 dye necessitated normalizing to pre-stimulation values to account for possible differences in dye loading and excitation strength.

**SR $Ca^{2+}$ measurements with D1ER sensor.** SR $Ca^{2+}$ was measured with the ratiometric genetically encoded $Ca^{2+}$ sensor D1ER as previously described[96]. Briefly, C2C12-derived myotubes cultured on poly-D-lysine-treated 35-mm-diameter glass-bottom dishes (MatTek, Ashland, MA, USA) were transfected with 2 μg of D1ER plasmid, using Lipofectamine 3000 protocol according to the manufacturer (Thermo Fisher Scientific, Basel, Switzerland). Two days after transfection myotubes were washed four times, electrically stimulated and SR $Ca^{2+}$ measurements were performed at 37 °C in $Ca^{2+}$-free Krebs solution (in mM: NaCl 135.5, $MgCl_2$ 1.2, KCl 5.9, glucose 11.5, HEPES 11.5, 200 μm Na-EGTA, final pH 7.3). Glass coverslips were inserted in a thermostatic chamber (Life Imaging Services, Basel, Switzerland). Cells were imaged on a DMI6000 B inverted fluorescence microscope, using an HCX PL APO ×40/1.30 numerical aperture oil immersion objective (Leica Microsystems, Wetzlar, Germany) and an Evolve 512 back-illuminated CCD with 16 × 16-pixel camera (Photometrics, Tucson, AZ, USA). Cells were excited at 430 nm through a BP436/20 filter. The two emission images were acquired with BP480/40 and BP535/30 emission filters. Fluorescence ratios were calculated in MetaFluor 7.0 (Meta Imaging Series, Molecular Devices, San Jose, CA, USA) and analysed in Excel (Microsoft, Seattle, WA, USA) and GraphPad Prism 8.3.1 (GraphPad, San Diego, CA, USA). Images were taken every 2 s. Resting D1ER fluorescence (535/480) ratio values fluorescence intensity data were measured before (CTRL) and after the simulated MICT or SIT protocols (S-MICT and S-SIT).

**SOCE assessment.** C2C12 S-MICT and S-SIT myotubes in MatTek glass-bottom dishes (MatTek, Ashland, MA, USA) were loaded with the cytosolic $Ca^{2+}$ indicator Fluo-4 AM (5 μM, Invitrogen, Basel, Switzerland) solubilized in a Krebs solution (in mM: NaCl 135.5, $MgCl_2$ 1.2, KCl 5.9, glucose 11.5, HEPES 11.5, $CaCl_2$ 1.8, final pH 7.3) immediately after the stimulation for 20 min in the incubator. Cells were then rinsed twice with a $Ca^{2+}$-free Krebs solution (in mM: NaCl 135.5, $MgCl_2$ 1.2, KCl 5.9, glucose 11.5, HEPES 11.5, 200 μM, Na-EGTA, final pH 7.3) and mounted to the confocal microscope system (Zeiss LSM 5 Live, Oberkochen, Germany; ×40 oil immersion lens; the excitation wavelength was 488 nm and the emitted fluorescence was recorded between 495 and 525 nm). The cells were stimulated with 1 μM thapsigargin (Sigma, St. Louis, USA) to deplete the SR $Ca^{2+}$ stores in Krebs $Ca^{2+}$-free solution. SOCE was investigated by adding the Krebs containing $Ca^{2+}$ solution to the dishes to reach a final concentration of 1 mM $Ca^{2+}$.

**RyR1 immunoprecipition and immunoblotting.** RyR1 immunoprecipition was performed as previously described[14] with slight modifications. Briefly, muscle biopsies or C2C12 myotubes were isotonically lysed in an ice-cold lysis buffer composed of: 50 mM Tris-HCl (pH 7.4), 150 mM NaCl, 20 mM NaF and 1 mM $Na_3VO_4$, and protease inhibitors (100 μl per 5 mg muscle tissue or per myotube well of a 6-well plate). The lysates were sonicated and then centrifuged at 9300 × g at 4 °C for 10 min. The supernatant was collected, and protein concentration was quantified using a BCA Assay Kit (Thermo Fisher Scientific, Basel, Switzerland). An anti-RyR1 antibody (4 μg 5029 Ab, Prof. Marks' lab, Columbia University, NY, USA) was used to immunoprecipitate RyR1 from 250 μg homogenate of human muscle or cells. The samples were incubated with the antibody in 0.5 ml of a modified RIPA buffer (50 mM Tris-HCl, pH 7.4, 0.9% NaCl, 5 mM NaF, 1 mM $Na_3VO_4$, 1% Triton X-100 and protease inhibitors) for 1 h at 4 °C. The immune complexes were incubated with protein A Sepharose beads (Sigma-Aldrich, St. Louis, MS, USA) at 4 °C for 1 h and the beads were washed three times with buffer. Proteins were separated on sodium dodecyl sulfate (SDS)–polyacrylamide electrophoresis gels (4–15% SDS-precast gradient gels to visualize both RyR1 and calstabin1 or 6% homemade gels for RyR1 alone and its post-translational modifications) and transferred onto nitrocellulose membranes for 1 h at 400 mA. After incubation with blocking solution (LI-COR Biosciences, Lincoln NE, phosphate-buffered saline (PBS) v/v) to prevent non-specific antibody binding, immunoblots were developed with rabbit anti-RyR1 (5029, Prof. Marks' lab, 1:5000), rabbit anti-phospho-RyR1-pSer2844 (Prof. Marks' lab, 1:5000), rabbit anti-Cys-NO antibody (Sigma-Aldrich, St. Louis, MS, USA, 1:2000) or rabbit anti-FKBP12 (Abcam, Cambridge, UK, 1:2500) antibodies. To determine channel oxidation, the carbonyl groups on the protein side chains were derivatized to 2,4-dinitrophenylhydrazone (DNP-hydrazone) by reaction with 2,4-dinitrophenylhydrazine. The DNP signal on RyR1 was determined by immunoblotting with an anti-DNP antibody (Sigma-Aldrich, St. Louis, MS, USA, 1:2000). Immunoreactive bands were visualized using infrared fluorescence (IR-Odyssey scanner, LI-COR, Lincoln, NE, USA). Band densities were quantified using Image Studio v.5.2.5 (LI-COR, Lincoln, NE, USA). Skeletal muscle cells were treated with 200 μM $H_2O_2$ (positive control for RyR oxidation), 250 μM NOC-12 (positive control of RyR nitrosylation) and 5 units PKA/reaction (positive control for RyR phosphorylation). Beads coupled with proteins without antibody were used as an IP negative control. IP purity was further tested by investigating SERCA protein contamination (using the rabbit SERCA-2a antibody, Abcam ab137020) as compared to non-IP samples (results shown in Supplementary Figs. 1 and 2).

**Western blot analysis.** A lysis buffer containing the following: 20 mM Tris/HCl (pH 6.8), 2 mM EDTA (pH 8), 137 mM NaCl, 10% glycerol, 10% Triton X-100, 10 mM glycerophosphate, 1 mM $KH_2PO_4$, 1 mM PMSF, 1 mM $NaVO_3$, 50 mM NaF, 10 mM NaPPi, and a protease inhibitor mixture (Roche, Complete Mini, Basel, Switzerland) was used to resuspend myotube pellets (100 μl/well of a 6-well plate) or human muscle samples (100 μl/5 mg of tissue). The preparation was homogenized with pipette tips for cells or potter for muscles, incubated for 1 h at 4 °C and gently sonicated. Then, nuclei and debris were removed by centrifugation at 9300 × g at 4 °C for 10 min. Protein quantification was assessed using the BCA Kit (Thermo Fisher Scientific, Ecublens, Switzerland). Fifteen to 20 μg of protein were incubated with 2× Laemmli sample buffer containing SDS and 2-mercapto-ethanol (Bio-Rad, Hercules, CA, USA) for 3 min at 95 °C, electrophoresed 1 h on 4–15% SDS-precast gradient gels (Bio-Rad, Hercules, CA, USA) and wet transferred 1 h onto PVDF membranes. Membranes were stained with Red Ponceau (homemade) and total protein bands were quantified using the Image Studio software v.5.2.5 (LI-COR, Lincoln, NE, USA). Then, the Red Ponceau was washed out with PBS and the membranes were saturated 1 h at room temperature with PBS-LI-COR blocking buffer (LI-COR, Lincoln, NE, USA). Blots were incubated overnight with rabbit anti-FKBP12 (Abcam, Cambridge, UK, 1:2500), rabbit anti-SERCA1 (Abcam, Cambridge, UK, 1:1000), mouse anti-total OXPHOS (Abcam, Cambridge, UK, 1:1000), mouse anti-PGC-1-α (Sigma, St. Louis, USA, 1:1000), rabbit anti-GAPDH (Sigma, St. Louis, USA, 1:5000), rabbit anti-PDH E1α phospho-serine 293 (Abcam, Cambridge, UK, 1:1000), mouse anti-PDHα (Abcam, Cambridge, UK, 1:1000), rabbit anti-STIM1 (Sigma, St. Louis, USA, 1:1000), mouse anti-MCU (Sigma, St. Louis, USA, 1:1000), rabbit CaMKII phospho-threonine 286 (Cell Signaling, Leiden, Netherlands, 1:1000), rabbit CaMKII (Cell Signaling, Leiden, Netherlands, 1:1000). Membranes were washed in PBS-buffered saline-Tween-20 and incubated for 1 h at room temperature with IRDye 680-conjugated donkey anti-mouse or anti-rabbit IgG (LI-COR, Lincoln, NE, USA, 1:10,000) and IRDye 800-conjugated donkey anti-mouse or anti-rabbit IgG (LI-COR, Lincoln, NE, USA, 1: 5000) in blocking buffer. Immunoreactive bands were visualized using infrared fluorescence (IR-Odyssey scanner, LI-COR, Lincoln, NE, USA). Band densities were quantified using Image Studio v 5.2.5 (LI-COR, Lincoln, NE, USA). Protein intensity signal was normalized to that of GAPDH (which was stable across samples and conditions) in human samples, while total protein staining (found as a more representative loading control for cells) was used to normalize protein content quantified in cells. All loading controls (GAPDH or total proteins) were investigated on the same gels of the protein of interest. Whole representative gels for OXPHOS proteins are provided in Supplementary Fig. 5 (for humans) and Supplementary Fig. 6 (for the cells).

The LI-COR system allows the detection of different proteins at the same time (revealed in different channels). When needed, the membranes were stripped using the appropriate LI-COR stripping solution (LI-COR, Lincoln, NE, USA). For human samples, all the protein quantifications were expressed as a percentage of the Pre-MICT values (kept as the reference for analysis of variance (ANOVA) testing). The protein quantifications for the cell samples were reported to that of CTRL or si-CTRL cells. Detailed information on the antibodies used in this study (supplier, catalogue number, clone name and number, lot number and method of validation) is provided in Supplementary data 3.

**ATP, NAD and NADH measurements**. ATP levels were measured in cell lysates using a luciferase-based assay (Sigma, St. Louis, USA) according to the manufacturer's instructions. Luminescence was measured using a plate reader (VICTOR Multilabel Plate Reader, Perkin-Elmer) in the presence of the substrate D-luciferin. The light intensity as relative light units was considered as a direct measurement of intracellular ATP concentration.

NAD and NADH levels were determined using the NAD/NADH Assay Kit (Abcam, Cambridge, UK) according to the manufacturer's instructions. Briefly, cell lysates were used for each enzyme recycling reaction. The fluorescence was measured in a fluorescence microplate (VICTOR Multilabel Plate Reader, Perkin-Elmer) at 540/590 nm. NAD and total NAD/NADH levels were calculated from a standard curve (μM).

**Measurement of mitochondrial morphology in C2C12 cells**. Myotubes were incubated in 200 nM MitoTracker red (Invitrogen, Basel, Switzerland) in Krebs solution (mM: NaCl 135.5, MgCl$_2$ 1.2, KCl 5.9, glucose 11.5, HEPES 11.5, CaCl$_2$ 1.8, final pH 7.3) and protected from light for 15 min at 37 °C. Rhod-2 fluorescence was detected by using a confocal laser scanning microscopy (inverted Zeiss LSM 710 confocal microscope, Oberkochen, Germany) technique to define single layers of cells at ×40 magnification. Mitochondrial shape descriptors and size measurement were determined using Fiji[97], an enhanced version of ImageJ software (http://fiji.sc/) as previously described[98,99].

**Measurement of mitochondrial Ca$^{2+}$ uptake**. To investigate mitochondrial Ca$^{2+}$ uptake, cultured C2C12 myotubes or dissociated FDB muscle fibres in MatTek glass-bottom 6-well plates with a bottom coverslip (MatTek, Ashland, MA, USA) were loaded with 1 ml of Krebs solution (mM: NaCl 135.5, MgCl$_2$ 1.2, KCl 5.9, glucose 11.5, HEPES 11.5, CaCl$_2$ 1.8, final pH 7.3) containing 1 μM of the mitochondrial fluorescent indicator Rhod-2 AM (Invitrogen, Basel, Switzerland) for 1 h at room temperature. Cells were then washed twice with Krebs solution and Rhod-2 fluorescence was measured using a confocal microscope (inverted Zeiss LSM 710 confocal microscope, Oberkochen, Germany) and ×40 oil immersion lens, with excitation at 532 nm and the emitted signal collected through a bandpass filter (540–625 nm). In some conditions, 10 μM of S107 or 20 μM of MTX was mixed into the Rhod-2/AM incubation solution (for 1 h) and maintained during the acquisition. A 180 s bout of S-MICT stimulation pattern (180 s, 14 V, 2 ms, 2 Hz continuous stimulation) or 2 × 30 s bouts separated by 2 min of no stimulation of the S-SIT stimulation pattern (5 s on, 1 s off, 14 V, 2 ms, 50 Hz) protocols were applied to the myotubes during a 300-s time-series live acquisition. These slightly modified S-MICT and S-SIT protocols allowed us to follow the Ca$^{2+}$ fluxes continuously and to avoid photobleaching. FDB fibres were excited as described for myotubes, with the exception that the voltage was set to 40 V. CTRL myotubes were only exposed to the laser light. Change in Rhod-2 fluorescence was calculated by reporting the peak of fluorescence to the baseline (normalized fluorescence). The amplitude of Rhod-2 normalized fluorescence measured 80 s after the end of the stimulation in each condition was used as an indicator of resting mitochondrial Ca$^{2+}$ uptake in response to exercise.

**Mitochondrial O$_2$ flux measurements using the Oroboros O2k**. High-resolution respirometry measurements were made as previously reported[59] and (Gnaiger_2019_MitoFit_Preprint_Arch_doi_10.26124_mitofit_190001). Fully differentiated C2C12 myotubes were stimulated with one session of the S-MICT or S-SIT protocols, and then washed and provided with warm fresh DM. The respiration measurements were performed 3 days after stimulation. In some conditions, cells were incubated in a DM containing 10 μM of S107 applied immediately after electrical stimulation for 72 h (S107 was also added to the MIRO5 buffer during data acquisition). The Oroboros O2k chambers were equilibrated using MIRO5 buffer (mM: EGTA 0.5, MgCl$_2$ 3, lactobionic acid 60, taurine 20, KH$_2$PO$_4$ 10, HEPES 20, D-sucrose 110, bovine serum albumin fatty acid free 1 g/l, pH 7.3). Then, one well of a 6-well plate (3–4 mg of protein/ml) was trypsinized 3 min with Trypsin/EDTA 0.05% (Invitrogen, Basel, Switzerland), centrifuged for 5 min at room temperature and 212 × g. The pellet was gently washed with a MiR05 buffer and resuspended in 2.5 ml of MiR05 solution and 2 mL of cell suspension inserted in each chamber. Mitochondrial respiration rates were assessed at 37 °C using a slightly modified SUIT-008 protocol of the DatLab 7.3 software (Oroboros, Innsbruck, Austria) (Supplementary Fig. 2p). After stabilization of respiration, the cells were permeabilized using 1 μl of digitonin (50 μg/ml), then successively incubated with 5 μl of malate (2 mM), 10 μl of pyruvate (10 mM) to check leak respiration (MP$_L$). MgCl$_2$ (0.5 mol/mol ADP) associated with 20 μl ADP (500 mM)

was then added to assess OXPHOS capacity supported by pyruvate and malate completed with 5 μl glutamate (10 mM), N$_P$. The integrity of the mitochondria was checked after the addition of ADP by using 5 μl of cytochrome C (4 mM). To investigate the activity of NS$_P$, 20 μl of succinate (10 mM) was added to the chamber. The ETS (NS$_E$) was assessed by using a 1 μl titration protocol with FCCP (1 mM); then, the activity of complex I was inhibited by 1 μl of rotenone (1 mM) (S$_E$) and residual oxygen consumption (ROX) was determined by addition of 1 μl of antimycin A (5 mM). An air calibration was performed on each experimental day. Protein quantification was performed in each condition on a representative well using the BCA assay (Thermo Fisher Scientific, Basel, Switzerland). The specific O$_2$ flux per mass was obtained after normalization of ROX-corrected O$_2$ fluxes by mg of protein in the chamber.

**Mitochondrial SC quantification**. Mitochondrial isolation and BN-PAGE were performed as described in detail elsewhere[100]. Fully differentiated myotubes were homogenized in 2 ml of cold sucrose isolation buffer with 40 strokes in a Wheaton glass tube at maximum speed. The mitochondrial fraction was isolated by collecting the supernatant after two rounds of centrifugation at 600 × g for 10 min and pelleted at 7000 × g for 10 min. The final pellet was resuspended in 200 μl of cold isolation buffer, and mitochondrial protein levels were quantified using the Lowry method. Fifty micrograms of mitochondrial extracts were solubilized with digitonin (8 g/g—digitonin/protein ratio) and centrifuged at 20,000 × g. The supernatant was collected and loaded into a NativePAGE 3–12% Bis-Tris-Gel (Invitrogen, Basel, Switzerland). After separation of the bands, the mitochondrial proteins were transferred onto a PVDF membrane using an iBlot gel Transfer Device (Invitrogen, Basel, Switzerland), followed by protein fixation with 8% acetic acid. The membrane was blocked, incubated with the OXPHOS primary antibodies (Abcam, Cambridge, UK) and anti-mitochondrial cytochrome C oxidase (Abcam, Cambridge, UK) and further incubated with the secondary mouse antibody (Novex, Thermo Fisher Scientific, Basel, Switzerland). Chromogenic substrate solution (Novex, Thermo Fisher Scientific, Basel, Switzerland) was used for the detection of the bands. After air-drying, the stained membrane was scanned.

**Metabolite quantification**

*Sample preparation*. Cells were scraped and extracted by the addition of 500 μl of MeOH:H$_2$O (4:1) per dish. This solution containing lysed cells was further homogenized in the Cryolys Precellys 24 sample Homogenizer (2 × 20 s at 9300 × g, Bertin Technologies, Rockville, MD, USA) with ceramic beads. The bead beater was air-cooled down at a flow rate of 110 L/min at 6 bar. Homogenized extracts were centrifuged for 15 min at 4000 × g at 4 °C (Hermle, Gosheim, Germany) and the resulting supernatant was collected and evaporated to dryness in a vacuum concentrator (LabConco, Missouri, USA). Dried sample extracts were resuspended in MeOH:H$_2$O (4:1, v/v) prior to liquid chromatography coupled to tandem mass spectrometry (LC-MS/MS) analysis according to the total protein content.

*Protein quantification*. The protein pellets were evaporated and lysed in 20 mM Tris-HCl (pH 7.5), 4 M guanidine hydrochloride, 150 mM NaCl, 1 mM Na$_2$EDTA, 1 mM EGTA, 1% Triton, 2.5 mM sodium pyrophosphate, 1 mM beta-glycerophosphate, 1 mM Na$_3$VO$_4$ and 1 μg/ml leupeptin using the Cryolys Precellys 24 sample homogenizer (2 × 20 s at 10,000 r.p.m., Bertin Technologies, Rockville, MD, USA) with ceramic beads. BCA Protein Assay Kit (Thermo Scientific, Massachusetts, USA) was used to measure (A562 nm) total protein concentration (Hidex, Turku, Finland).

*LC-MS/MS analysis*. Extracted samples were analysed by hydrophilic interaction liquid chromatography coupled to tandem mass spectrometry (HILIC-MS/MS) in negative ionization mode using a 6495 triple quadrupole system interfaced with 1290 UHPLC System (Agilent Technologies)[101].

Chromatographic separation was carried out using a SeQuant ZIC-pHILIC (100 mm, 2.1 mm ID and 5 μm particle size, Merck, Darmstadt, Germany) column. The mobile phase was composed of A = 20 mM ammonium acetate and 20 mM NH$_4$OH in water at pH 9.7 and B = 100% acetonitrile. The linear gradient elution from 90% (0–1.5 min) to 50% B (8–11 min) down to 45% B (12–15 min). Finally, the initial chromatographic conditions were established as a post-run during 9 min for column re-equilibration. The flow rate was 300 μl/min, column temperature 30 °C and sample injection volume 2 μl. Electrospray ionization source conditions were set as follows: dry gas temperature 290 °C and flow 14 L/min, sheath gas temperature 350 °C, nebulizer 45 psi, and flow 12 L/min, nozzle voltage 0 V and capillary voltage −2000 V. Dynamic multiple reaction monitoring was used as acquisition mode with a total cycle time of 600 ms. Optimized collision energies for each metabolite were applied.

Raw LC-MS/MS data were processed using the Agilent Quantitative analysis software (version B.07.00, MassHunter Agilent Technologies). For absolute quantification, calibration curves and the stable isotope-labelled internal standards were used to determine the response factor. Linearity of the standard curves was evaluated for each metabolite using a 7-point range. In addition, peak area integration was manually curated and corrected when necessary.

A cut-off was set by the metabolomics platform at 20% change compared to CTRL, over which data variations are considered as relevant.

The raw data related to the metabolite quantification are presented in Supplementary Data 4.

## Proteomics data acquisition

*Protein digestion.* Differentiated C2C12 myotubes were exposed to the SIT stimulation protocol and then divided into two groups with one immediately treated after the SIT session with 10 μM S107 for 72 h. After 72 h, cell pellets were collected and stored at −80 °C for later analysis. Replicate samples of SIT and SIT S107 (5 per group) were digested with the miST method (modified version of the in-StageTip method, ref. [102]). Briefly, frozen cell pellets were resuspended in 100 μl miST lysis buffer (1% sodium deoxycholate, 100 mM Tris pH 8.6, 10 mM dithiothreitol) by vigorous vortexing. Resuspended samples were heated at 95 °C for 5 min, and 100 μg of protein were transferred into new tubes, based on tryptophan quantification[103]. Samples were then diluted 1:1 (v:v) with water containing 4 mM MgCl$_2$ and benzonase (Merck #70746, 100× dilution of stock = 250 U/μl) and incubated for 15 min at room temperature to digest nucleic acids. Reduced disulfides were alkylated by adding 1/4 vol (25 μl) of 160 mM chloroacetamide (final 32 mM) and incubating at 25 °C for 45 min in the dark. Samples were adjusted to 3 mM EDTA and digested with 1 μg Trypsin/LysC mix (Promega #V5073) for 1 h at 37 °C, followed by a second 1 h digestion with a second and identical aliquot of proteases. To remove sodium deoxycholate, two sample volumes of isopropanol containing 1% trifluoroacetic acid (TFA) were added to the digests, and the samples were desalted on a strong cation exchange (SCX) plate (Oasis MCX; Waters Corp., Milford, MA, USA) by centrifugation. After washing with isopropanol/1%TFA, peptides were eluted in 250 μl of 80% MeCN, 19% water and 1% (v/v) ammonia.

*Tandem mass tag (TMT) labelling.* Eluates after SCX desalting were dried and resuspended in 100 μl water. Thirty microliters of digests were then aliquoted and dried again, before resumption in 25 μl of 50 mM TEAB buffer, pH 8.0. For labelling, 0.2 mg of TMT reagent in 20 μl acetonitrile were added to the samples for 1 h at room temperature, after which excess reagent was quenched with 1 μl of 5% hydroxylamine for 15 min at room temperature.

An aliquot (0.8 μl) was injected before mixing to assess labelling completion (>98% peptide spectrum matches) by database search with TMT as variable modification (MASCOT software, www.matrixscience.com). After mixing, the TMT multiplex sample was dried and desalted on a SepPak micro C18 96-well plate (Waters Corp., Milford, MA, USA).

*Peptide fractionation.* The dried desalted eluate was dissolved in 4 M urea containing 0.1% ampholytes pH 3–10 (GE Healthcare, Ecublens, Switzerland). Then, 75% of the sample (225 μg) was fractionated by off-gel focusing as described[104]. The 24 peptide fractions obtained were desalted on a SepPak micro C18 96-well plate, dried and dissolved in 25 μl of 0.05% trifluoroacetic acid, 2% (v/v) acetonitrile for liquid chromatography-mass spectrometry/mass spectrometry (LC-MS/MS) analysis.

*MS analysis.* Data-dependent LC-MS/MS analysis of TMT samples was carried out on a Fusion Tribrid Orbitrap mass spectrometer (Thermo Fisher Scientific, Basel, Switzerland) interfaced through a nano-electrospray ion source to an Ultimate 3000 RSLCnano HPLC System (Dionex, Thermo Fisher Scientific, Basel, Switzerland). Peptides were separated on a reversed-phase custom packed 40 cm C18 column (75 μm ID, 100 Å, Reprosil Pur 1.9 μm particles, Dr. Maisch HPLC Gmbh, Ammerbuch-Entringen, Germany) with a 4–76% acetonitrile gradient in 0.1% formic acid (total time = 140 min). Full MS survey scans were performed at 120,000 resolution. A data-dependent acquisition method controlled by the Xcalibur 4.2 software (Thermo Fisher Scientific, Basel, Switzerland) was used to optimize the number of precursors selected (top speed) of charge $2^+$ to $5^+$ while maintaining a fixed scan cycle of 1.5 s. The precursor isolation window used was 0.7 Th.

Peptides were fragmented by higher energy collision dissociation with a normalized energy of 37%. MS2 scans were performed at a resolution of 50,000 in the Orbitrap cell, to resolve 10-plex TMT reporter ions. The $m/z$ of fragmented precursors was then dynamically excluded from selection during 60 s.

*MS data analysis.* Data files were analysed with MaxQuant 1.6.3.4[105,106], incorporating the Andromeda search engine[106]. Cysteine carbamidomethylation and TMT labelling (peptide N termini and lysine side chains) were selected as fixed modifications while methionine oxidation and protein N-terminal acetylation were specified as variable modifications. The sequence databases used for searching were the mouse (*Mus musculus*) Reference Proteome based on the UniProt database (www.uniprot.org, version of January 31, 2019, containing 54,211 sequences), and a "contaminant" database containing the most usual environmental contaminants and the enzymes used for digestion (keratins, trypsin, etc). Mass tolerance was 4.5 p.p.m. on precursors (after recalibration) and 20 p.p.m. on higher energy collision dissociation fragments. Both peptide and protein identifications were filtered at a 1% false discovery rate (FDR) relative to hits against a decoy database built by reversing protein sequences. For TMT analysis, the raw reporter ion intensities generated by MaxQuant (with a mass tolerance of 0.003 Da) and summed for each protein group were used in all following steps to derive quantitation.

*Processing of quantitative data and statistical tests.* The MaxQuant output table "proteinGroups.txt" was processed with the Perseus software[107] to remove proteins matched to the contaminants database as well as proteins identified only by modified peptides or reverse database hits, and those without any quantitative values, yielding a first unfiltered list of 7229 identified proteins. Next, the table was filtered to retain only proteins identified by a minimum of three peptides (5870 protein groups left).

After log 2 transformation of all intensity values and normalization by median subtraction, a two-sample *t* test with Benjamini–Hochberg FDR correction (threshold at 0.05 on the adjusted *p* value;[108] was performed between the SIT and SIT S107 groups and showed 365 significant proteins. Gene ontology annotation enrichment test was carried out with Perseus, using an average difference in log 2 scale between both groups, and applying a Benjamini–Hochberg FDR of 2%. The resulting score used in Fig. 3 and Supplementary Fig. S3 indicates how far is the centre of the distribution of values for the protein category considered relatively to the overall distribution of values[109]. The interval of this positional score is between −1 and 1.

*Raw data deposition.* All the mass spectrometry proteomics raw data together with MaxQuant output tables are available via the Proteomexchange Consortium via the PRIDE[110] partner repository with the dataset identifier PXD018409.

Submission details:
Project Name: "Triggering acute RyR1 Ca$^{2+}$ leak leads to improved mitochondrial remodeling and function".
Project accession: PXD018409.
Project DOI: Not applicable.
The proteomics data generated in this study have been deposited with PRIDE under accession code PXD018409.
Analysed proteomics data are provided in Supplementary Data 1 and 2. SIE (sprint interval exercise) refers to SIT.
The Supplementary Method is provided in the Supplementary information file.

**Chemicals**. Fluo-4 AM, DMEM, CM-H2DCFDA, Rhod-2 AM, FBS, HS, NEAA, streptomycin–penicillin Lipofectamine 3000, OptiMEM solutions were purchased from Life Technologies (Thermo Fisher, Basel, Switzerland). Details on chemicals are provided in Supplementary Data 3.

**Statistical analysis**. Except for proteomic and metabolomic analyses, data are presented as mean ± SD. Statistical significance was determined using unpaired *t* tests to compare two groups or paired *t* tests to compare two time points. When data were normally distributed, analysis of variance (ANOVA) was performed to compare more than two groups. Two-way ANOVA was followed by Tukey's or Sidak's multiple comparisons post hoc tests to compare many groups or multiple time points when the interaction was significant. One-way ANOVA was followed by Tukey's or Sidak's multiple comparisons post hoc tests to compare different groups when the main effect was significant. When data were not normally distributed, Kruskal–Wallis ANOVA test followed by Dunn's multiple comparisons were used to compare different groups. Data were analysed using GraphPad Prism version 8.4.2 and SigmaPlot version 11.0. The level of significance was fixed at $p \leq 0.05$. All detailed *p* values are presented in Supplementary Data 5. The figures were mounted using Adobe Illustrator 23.0.3.

**Reporting summary**. Further information on research design is available in the Nature Research Reporting Summary linked to this article.

## Data availability

All data generated and analysed during this study are included in this published article (and its Supplementary information files). The proteomics data generated in this study have been deposited with PRIDE under accession code PXD018409. Analysed proteomics data are provided in Supplementary Data 1 and 2. The data related to the metabolite quantification of this study are presented in Supplementary Data 4. All detailed *p* values are presented in Supplementary Data 5. Source data are provided with this paper.

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

## Acknowledgements

We thank Prof. Lluis Fajas, Prof. Luc Pellerin and Prof. Romano Regazzi for infrastructure support and discussions; Mr. Gilles Dubuis, Mrs. Cendrine Repond, Dr. Sarah Geller and Dr. Pau Gama Perez for technical support. We also thank the animal facility

of the Department of Biomedical Sciences, the proteomics, metabolomics and the cellular imaging facility platforms at the University of Lausanne. We thank Drs R.Y. Tsien and A. Palmer (University of California, San Diego) for providing the D1ER construct. This work was supported by the SNF Grant (no. IZK0Z3_173941 to N.Z., no. 194964 to C.D., Ambizione PZ00P3_168077 and PRIMA PR00P3_193166 to I.C.L.-M.) and the Subside "Tremplin/Relève Académique" of the UNIL to N.Z. T.I.d.L. received financial support from FAPESP (2019/11171-7). P.M.G-R received financial support from the Instituto de Salud Carlos III (ISCIII) Grant PI15/00701, co-financed by the European Regional Development Fund "A way to build Europe". This work was also supported by grants from the NIH to A.R.M. (T32HL120826, R01HL145473, R01DK118240, R01HL142903, R01HL061503, R01HL140934, R01AR070194, R25NS076445). J.A. received financial support from the Ecole Polytechnique Fédérale de Lausanne, the Fondation Marcel Levaillant and the Swiss Foundation for Research on Muscle Diseases (FSRMM). B.K. and N.P. were supported by institutional funds.

## Author contributions

Conceptualization: N.Z., B.K. and N.P. Methodology: N.Z., S.R., H.D., C.D., T.I.d.L., U.D.M., P.M.G-R., J.N.F., A.R.M., B.K. and N.P. Investigation: N.Z., S.R., H.D., T.I.d.L., U.D.M, C.D., M.F., L.S., J.V., B.K. and N.P. performed the experiments. Formal analysis: N.Z., H.D., S.R., C.D., T.I.d.L., U.D.M., M.F., I.C.L-M., P.M.G-R. and N.P. Writing: N.Z. wrote the manuscript, all authors contributed to editing. Project administration and management: N.Z. Funding acquisition and resources: N.Z., P.M.G-R., I.C.L.-M., B.K., A.R.M., J.A. and N.P. Project supervision: B.K. and N.P.

## Competing interests

U.D.M. and J.N.F. are employees of Nestlé Research, which is part of the Société des Produits Nestlé SA. A.R.M. is chair of the SAB and holds stock in ARMGO Pharma Inc., a biotech company developing drugs targeting RyR channel leak. The remaining authors declare no competing interests.
