## [Peer Review File · Nature Communications]

Acute RyR1 Ca²⁺ leak enhances NADH-linked mitochondrial respiratory capacityReviewers' comments:

Reviewer #1 (Remarks to the Author):

High intensity interval training (HIIT) including sprint interval training (SIT) is known to be an efficient type of exercise that can lead to even larger increases in mitochondrial biogenesis and VO₂max than classical moderate intensity exercise. Moreover, HIIT has gained interest both for athletes as well as a promising alternative for patient care. Here the authors show that one single bout of SIT leads to increased OXPHOS proteins and improved respiratory capacity. These findings are explained to be mediated by RyR1 Ca²⁺ leak, a well described mechanism that some of the co-authors previously have shown to contribute to muscle weakness as well as normal ageing. As the authors claim, there is strong support that Ca²⁺ is involved in HIIT-induced musculoskeletal beneficial effects, however, to solely link it to a short-term and small RyR1 Ca²⁺ leak (since effects impeded by S107) appears too simplified.

Comments:

SIT is a shorter and more intensive muscle stimulation, ie. the Ca²⁺ transients during this period, as well as the intracellular Ca²⁺ and stress on the OXPHOS system will also be higher during the stimulation period than for MICT. Thus, SIT will trigger reactive oxygen species (ROS) formation. The authors have detected ROS modifications on RyR1, which raises the question if an antioxidant would treatment have the same effect as S107?

Is S107 treatment causing decreased amount of post-translational modifications (DNP, SNO) of RyR1 after SIT?

The authors show that one of the effects of the SIT-induced RyR1-mediated Ca²⁺ leak is enhanced mitochondrial Ca²⁺ uptake via MCU and PDH activation, which is contributing to enhanced OXPHOS. A concern is that the MCU complex is thought to have a very low affinity for Ca²⁺ (KD of 20–30 μM under physiological conditions). Thus, the intracellular Ca²⁺ concentration should be approximately 5–10 μM for considerable mitochondrial Ca²⁺ influx, which is high and unlikely unless that mitochondria are juxtaposed with the SR (e.g Rizzuto et al. 1998, Patergnani et al. 2011). I.e. microdomains with high Ca²⁺ concentrations ([Ca²⁺]_i > 10 μM) is thought to form transiently in regions of close apposition between the mitochondria and the RyR1, ensuring a prompt accumulation of Ca²⁺ inside the mitochondria. Thus, the SIT stimulation could possibly lead to the required magnitude of intracellular [Ca²⁺]_i, however, how can a small and transient RyR1-mediated Ca²⁺ leak lead to the required magnitude of intracellular [Ca²⁺]_i? Furthermore, the C2C12 cells are appropriate for genetical manipulation and thus would SIT-induced mitochondrial adaptation be blunted in MCU knockdown cells?

Page 3, reference 15 is a study on rodents not “healthy people”.

Calstabin1 and calstabin 2 are for a large majority of scientist more known as FKBP12 and FKBP12.6, respectively, thus for an unfamiliar reader it must be clear that they are the same proteins.

Many antibodies used in the manuscript are listed, but the antibodies used to detect DNP, CySNO and P-RyR1 are not listed in the table, please add.

Serca2a is a slow-twitch isoform and the muscle biopsy was taken from vastus lateralis which is a mixed muscle type. Is the expression of fast-twitch type Serca1 also altered by SIT (or MICT)?

The immunoblots for OXPHOS protein levels are a cocktail of antibodies so it would be more suitable to present it all on one gel instead of cropped into several blots.

Page 5, third paragraph “Both were unchanged...” but the authors refer to three different proteins.

The authors could not see any SIT-induced differences in PGC1a expression from human biopsies (Fig S1I) but in the C2C12 cells (Fig. S2M). The molecular weight marker is different between the two blots, please specify the molecular weight for the band analyzed. The predicted molecular weight for PGC1a1 is 90.4 kDa but it's a protein known to have many post-translational modifications and apparent weight is commonly around 120 kDa. PGC1a antibodies are notoriously unreliable and thus running it together with a negative control is recommended. Moreover, PGC1a has four different isoforms a1-a4. Which isoforms were detected with the antibody? Calbiochem/Merck (ST1202) has an antibody that binds to the N-terminus that has been tested against KO tissue and able to detect several of the PGC1 isoforms.

Moreover, with an antibody which is unreliable and given that PGC1a is an unstable protein with a short half-life, the mRNA levels are commonly used and a tested readout to examine effects of exercise. Were any SIT or MICT-induced alterations in mRNA levels observed?

Page 6, first paragraph appears very speculative. Intermitochondrial communication is present in skeletal muscle but unlike mitochondria involved in fusion events in cultured cells, muscle mitochondria do not appear to move out of position during presumed fusions (see review Lavorato et al 2020). With this in mind, what is the suggested role of increased OPA1 (long-isoform) after SIT?

Page 7, Fig 2B, RyR1-phosphorylation is not mentioned but was listed in Fig 1E. Were there no increase in P-RyR1 detected? If so, are only ROS mediated post-translational modifications involved in the downstream signaling observed after SIT?

Page 7, caffeine stimulation in the absence of extracellular Ca²⁺ was used to show that SIT influences the SR Ca²⁺ release. However, this does not necessarily need to exclude a compensatory SIT-induced SOCE mechanism. E.g. Ivarsson et al. 2019 showed that voluntary running influences SOCE and STIM expression. Were there any SIT or MICT-induced SOCE observed?

Page 8, Fig 2J-O, difficult that by eye detect any changes in expression.

Page 8, Fig 2P-R, “observed increased mitochondrial perimeter....” Please clarify what this observation reflects or what the physiological relevance is.

General for all immunoblots, it is troubling that the different groups Ctrl, MICT, SIT etc are cropped and separated and hence not compared on the same blot. Especially for proteins where there are small alterations in expression levels (e.g 1D, 1L, 2B, 2J, 4I, S1F, S1I, S1M, S2M) the comparisons between groups should be made from the same membrane/gel.

Page 9, second paragraph “S107 acts by reinforcing the physical interaction between RyR1 and calstabin1, preventing or stopping Ca²⁺ leak”. Please add a blot that verifies that S107 treatment prevents SIT-induced calstabin (FKBP12) dissociation.

Reviewer #2 (Remarks to the Author):

The present manuscript reports a thorough investigation of molecular and cellular responses to intense exercise (human muscle) or simulated exercise (myotubes, mouse muscle fibres) that aimed to elucidate the role of RyR1 calcium leak in exercise-induced skeletal muscle remodeling. In a series of experiments, the authors demonstrated that intense, but not “moderate,” exercise caused acute modifications to the RyR1 protein that reduce SR calcium content (i.e., elicit leak), increase indices of mitochondrial content, increase mitochondrial calcium uptake, and activate PDH. The authors demonstrated that an RyR1 stabilizer blunted responses to SIT in vitro. Overall, the authors suggest that their results provide evidence for a beneficial role of acute RyR1 calcium leak in response to exercise and that this response is exercise-intensity dependent.

Clearly, the authors have done a tremendous amount of work to support their conclusions. The choice of experiments and use of various models was appropriate to test the authors' hypothesis, and I think the conclusions are supported by the results presented in their manuscript; however, I think the authors can improve their manuscript by clarifying aspects of the methods, particularly some details regarding the Western blotting, the electrical stimulation protocols, and how they determined their sample sizes. Adding some missing information will be helpful, but I recommend that the authors also be more careful when referring to exercise, indicating whether real exercise or electrical stimulation elicited specific results. Furthermore— and I apologize for making this general statement—I do not think the manuscript is particularly well written. I think a thorough check for small grammar mistakes (e.g., missing commas, odd phrasing, etc.) will help the authors convey their results better to the audience. Examples from the introduction include (i) the final sentence of paragraph two; (ii) the first sentence of paragraph three; (iii) the sentence about S107 in the third paragraph. Some additional commas to set off certain phrases and shorter sentences may help. Similar issues are present throughout the manuscript. Personally, I had to re-read multiple sentences to discern the point the authors were trying to make, and I could not always figure it out.

After re-reading my own comments, I wish to clarify that I see a lot of value in this manuscript, and I think the results are novel and likely to be of interest to several fields of research. The authors have done a commendable job. Yet, I still have major concerns about particular aspects of the manuscript that require clarification.

General

Throughout the results, I am unclear of how the authors decided on their reported sample sizes. For human muscle, the sample size is obvious, as it is the number of human participants. For other experiments, my impression is that replicates are counted as independent samples. For example, in Figure 2 (panel S), the authors indicated the sample size as "4-5 wells per group from 3 independent stimulated dishes." Here, it reads as if the sample size is 3 (i.e., 1 per group), not 12-15 (i.e., 4-5 per group). Similarly, when 6 myotubes are examined from each of 3 independent stimulated dishes (Figure 2, panel I), I interpret this as 6 replicates of 3 samples (so an n of 3). My interpretation here is further supported by the authors language later in the manuscript: In Figure 4, the authors specify that 5-9 independent dishes were used per group and 3-6 myotubes were examined per dish. In this example, it seems that the independent dishes were the sample size and that myotubes were averaged to produce one value per dish. Thus, the authors seem to determine sample size differently depending on the experiment. Please clarify the sample size throughout, distinguishing between samples and replicates.

I appreciate that the authors have included a lot of data and many figures, but there seem to be some important mistakes. Specifically, some Western blot images appear to be duplicated within the

manuscript. This is apparent in Figure 4, panel I (Ctrl vs. MICT are identical for PDH E1alpha) and Figure S1 panel F (MICT and SIT are identical for DHPR). Please correct these images.

Why were only human males included in this experiment? I don't expect the authors to re-do their study, but was there any reason to expect females not to respond to SIT?

The need to clarify actual exercise from electrical stimulation in vitro may reflect my own bias for human research; however, I think it's important that the authors use specific language throughout the manuscript to indicate what results occurred in response to exercise in humans and what results occurred in response to electrical stimulation in cells derived from mice. The use of multiple models is a strength of the experiment; however, others may misinterpret the findings without greater attention to detail on the part of the authors. As an example, in the first sentence of the discussion, the authors choice of wording implies that mice performed SIT, which they did not. Given that this paragraph does not mention electrical stimulation, the reader may expect that these results were the result of actual exercise. The model to which specific results belong could be made more explicit, even if the overall conclusions remain the same.

The other reason to make this distinction is that the electrical stimulation protocols seem to be somewhat arbitrary. Other than representing low and high intensity stimulations, I don't think it's possible to determine how well either represents SIT and MICT performed by humans. How did the authors choose their "MICT" protocol? At 1/25th the frequency, the difference between the "SIT" and "MICT" protocols seems much larger than would be apparent for human exercise, where power outputs are probably 2-3 times greater for SIT relative to MICT. This is important—as it is with human exercise trials—because if the authors choose an intensity that is too low, the "MICT" protocol would not be expected to elicit a response. Similarly, how was the "SIT" protocol chosen? While I see less issues here, it's possible that a high-intensity electrical stimulus that was lower than what the authors used (e.g., 40 Hz) may not have elicited the results presented herein. Please explain how the protocols were chosen. Unless the authors have a strong reason for disagreeing, I think it should be made explicit that these two protocols do not necessarily represent MICT vs. SIT in humans and that the results are not generalizable to all comparisons of continuous and interval exercise. As an example, if 4 Hz was used for the "MICT" protocol, could similar results have been obtained for both intensities? Without more information from the authors, I do not think this possibility can be ruled out. Expressing some brief limitations to the generalizability of these results would be helpful.

The authors do not comment on their previous finding of RyR1 fragmentation. Given the prominence of this finding in the previous PNAS paper, it seems odd not to mention that result here. Indeed, on page 3, the authors only refer to their mouse data from this manuscript. Similarly, the authors do not refer to another of their studies in human muscle from Schlittler et al. Both findings in human muscle seem relevant. Please explain why full-length RyR1 abundance was unchanged in the present human/cell experiments. What is the relevance of "fragmentation" as a post-translational modification?

Place N, Ivarsson N, Venckunas T, Neyroud D, Brazaitis M, Cheng AJ, Ochala J, Kamandulis S, Girard S, Volungevičius G, Paužas H. Ryanodine receptor fragmentation and sarcoplasmic reticulum Ca²⁺ leak after one session of high-intensity interval exercise. *Proceedings of the National Academy of Sciences*. 2015 Dec 15;112(50):15492-7.

Schlittler M, Neyroud D, Tanga C, Zanou N, Kamandulis S, Skurvydas A, Kayser B, Westerblad H, Place N, Andersson DC. Three weeks of sprint interval training improved high-intensity cycling performance and limited ryanodine receptor modifications in recreationally active human subjects. *European journal of applied physiology*. 2019 Sep 1;119(9):1951-8.

I do not fully understand the importance of the PDH dataset. The final sentence of the results section seems to implicate PDH as being necessary in the process of adaptation rather than as a biomarker of calcium uptake into the mitochondria (and perhaps an elevated capacity to oxidize carbohydrate in vivo). Are the authors making the point that the PDH response is somehow necessary for the mitochondrial adaptations, or am I confused? An entire paragraph of the discussion discusses PDH data, and the finding is included in the title; but the panel presented in figure 4 doesn't seem to extend beyond PDH being dephosphorylated: what happens next that is relevant for stimulating an increase in the aerobic capacity of muscle cells? I understand that the PDH data allows the authors to suggest increased mitochondrial calcium uptake in human muscle samples, but the myotube data seems to be a more direct measure of mitochondrial calcium uptake than PDH dephosphorylation. Please clarify.

Introduction

Page 3: The juxtaposition of HIIT being used in patient care and SIT being as effective as MICT gives the impression that SIT is being used in patient care. I do not think that SIT is commonly used in patient care. Please correct or provide evidence for this point.

Page 3: Are SIT and MICT exercise “models”? This phrasing is a bit odd, particularly in a sentence that uses the word “models” in other contexts. Furthermore, how are human muscle biopsies a “model”?

Page 3: Here, the authors state “Ser393;” however, this seems to be a typo (i.e., it should be 293).

Results/Methods

Page 4: Reporting the total work performed as kJ/kg is uncommon for exercise physiology. I suggest reporting these numbers as kJ and also including the average power output (here or in the supplementary information).

Page 4 (and methods): The MICT program for human participants needs to be clarified. At what intensity was this exercise performed? 65% of VO₂peak or the mechanical associated with 65% of VO₂peak during the ramp incremental test? The mechanical power associated with 65% VO₂peak during the ramp incremental test would elicit a much higher VO₂ during constant load exercise (due to transit time delay and kinetics of VO₂). Did the authors measure VO₂ during the MICT effort to ensure that the chosen power output elicited 65% of VO₂peak? If so, please report this value. If 65% of the peak power output was used, as indicated in the methods, please correct the results section to reflect this point. Similarly, the methods section indicates 4 min of rest; however, the results reports a workload. Please clarify that the “rests” were active recovery (and provide power output). If I’ve misunderstood the authors here, please let me know.

Page 4 (and methods): Please indicate the timing of post-exercise neuromuscular assessment: What was the delay from the cessation of exercise to the MVC? If exact values are not available, an approximation would still be helpful to the reader, as the timing is relevant to the central fatigue measurement.

Methods: What is the effect of centrifuging muscle samples at 10,000 rpm on the proteins of interest? Previously, Murphy and Lamb (2013) commented that many proteins, specifically the calcium-handling protein, calsequestrin, are lost in the cellular debris that is discarded with this procedure. Would the use of centrifugation result in fractionation of the samples in the present manuscript?

Murphy RM, Lamb GD. Important considerations for protein analyses using antibody based techniques: down-sizing Western blotting up-sizes outcomes. *The Journal of physiology*. 2013 Dec 1;591(23):5823-31.

Figures: The authors report ponceau stains in their figures and indicate in the methods that these stains were quantified, but it is unclear how these images were used for normalization, if at all. Please clarify whether these images were visually inspected or quantified to correct for differences in loading. In general, the normalization of western blots is unclear. In the figure caption for Figure 1, it states that GAPDH was used to normalize OXPHOS proteins, but this isn’t described in the methods. In Figure 2, OXPHOS western blots were not normalized to GAPDH? Why not, and how were they normalized?

Figures: For all Ponceau images, it’s unclear what part of the membrane is being presented. Can the authors indicate molecular weights or name the proteins if obvious (e.g., myosin or actin)? Is the same region of each membrane shown in all Ponceau images? If so, this could be stated in the methods.

Page 4: Were there positive/negative controls for the Co-IP procedure?

Methods: For Western blots, were replicates performed?

Methods: What procedure was used to probe multiple proteins? Were proteins measured on separate membranes? Were membranes cut to allow for multiple proteins to be probed simultaneously? Were membranes stripped and probed sequentially? Does the LICOR system allow for multiple primary/secondary antibodies to be used simultaneously? A brief explanation in the methods would be helpful, given the reliance on western blotting results.

Figure 1: The use of bars and asterisks in the figures needs some clarification. For example, Panels I and J of Figure 1 seems to show 6 asterisks above the SIT bars, which I think should be two groups of three asterisks (as is shown for panel K in this figure). Regardless, the meaning of three asterisks has been omitted from the figure 1 caption (I found it elsewhere).

Methods: I apologize if I missed it, but it doesn't seem that electrical stimulation methods for "SIT" and "MICT" in mouse muscle fibres is reported in the text.

Page 5: "NRF1" should be in parentheses

Page 5: I do not understand this sentence as written: "PGC-1a total protein, nuclear localization or post-translational modifications, including acetylation, regulate PGC-1a activity 30." Specifically, how does the total abundance "regulate" activity of this protein? Perhaps combining this sentence with the sentence that follows would help clarify the point here.

Page 7: I do not think the term "tended" is appropriate for a p-value that was greater than 0.05.

Figure 1 (and page 12): The authors point to the rapid increases in mitochondrial content from a single session of SIT, which from Figure 1 appear to be ~150-175% relative to pre. Such changes are comparable to (or greater) than what would be expected with months of exercise training. Do these changes in complex I, II, and IV protein content actually reflect changes in mitochondrial density when used in this context? Given that 2/5 OXPHOS proteins did not respond to the SIT stimulus, how did the authors decide that the overall result was "rapid beneficial mitochondrial adaptations"? Was any attempt made to measure respiration in these samples? I see that similar (and maybe larger) responses

were observed in cell models, particularly for mitochondrial respiration. While the statistical difference between groups seems relevant, do the authors think that SIT is actually capable of increasing mitochondrial content in human muscle this dramatically?

Page 8: Some references to figures are a little confusing. This sentence refers to multiple figures with different types of data and proteins/genes other than PGC1a: “GC-1a protein was increased 72 h post SIT, but not MICT (Figures S2K-O).”

Page 9: In this sentence, it reads to me as if S107 was not effective in blocking the effects of MICT, rather than MICT not being effective, compared to SIT. Please clarify: “S107 treatment inhibited the increase in mitochondrial OXPHOS proteins (Figures 3 B-G), mitochondrial fusion protein Mfn2 (Figures 3B and 3H) and mitochondrial perimeter and area (Figures 3 I-K) observed in response to SIT – but not MICT (Figures 3 B-H).”

Figure 4: Images and graphs are missing for SIT S107. Furthermore, if measurements were taken where arrows indicate (i.e., 300s), it seems inappropriate to compare conditions after different amounts of time have passed since stimulation (e.g., MICT vs. SIT). In Figure 4C (SIT), the trace appears likely to decrease sharply if more time were provided. Here, the difference in stimulation duration (60s vs. 180s) does not reflect SIT vs. MICT for human exercise or other stimulation protocols used. Please clarify the figure and explain the timing and choice of stimulation procedures.

Page 12: I think the proteomic analysis revealed decreased capacity for carbohydrate metabolism rather than a change in metabolism per se.

Page 13: I also do not understand the authors’ point about the second increase in mitochondrial calcium following stimulation (paragraph 2, discussion): is proteomic and respirometry data from the 72 h time point relevant to this acute increase in mitochondrial calcium content? Presumably it would have returned to baseline during the long recovery. In other words, was it necessary to demonstrate longer lasting responses related to PDH/complex I in order to confirm calcium uptake by the mitochondria following an acute stimulation of the myotubes? I apologize if this is my ignorance showing.

Page 13: Although I don’t disagree, I think some references for the health-promoting benefits of exercise and for “today’s recommendations” are needed.

Page 13: The S107 compound was not used in conjunction with a “maximum intensity variant of interval training.” That this happened in myotubes isn’t revealed for 3 more lines.

Conclusion: I think this section provides a clear summary of the general results.

Statistics: The authors should clarify which statistical test was used for which variable and when post hoc tests were performed. For ANOVA results, the p-values for main and interaction effects should be reported in supplementary information.

Reviewer #3 (Remarks to the Author):

The present paper by Zanou, Place and colleagues examines the interaction between RyR mediated calcium leak and mitochondrial respiratory capacity. The authors working hypothesis is that calcium leak is important for exercise adaptations, and furthermore that sprint interval training more robustly induces these changes. The present model is an extension of the authors previous work in PNAS showing SIT induced RyR fragmentation, and therefore the novelty is limited to the comparison between endurance and SIT exercise, and is somewhat limited in scope. In addition, previous work has shown that endurance and SIT equally induce mitochondrial biogenesis (see seminal work by Burgomaster and Gibala) and signals related to mitochondrial biogenesis (Bartlett et al JAP 2012), challenging the fundamental premise and importance of the data reported here. The overall discussion surrounding classical exercise data is limited in scope, and the authors are encouraged to take a more balanced approach to placing their data in context.

Major comments:

1- The working model is not novel as there are many papers examining the mobilization of calcium from the SR to mitochondria during contraction (PMID: 29988564; PMID: 21106237), which decreases the novelty of the study. The authors themselves have previously supported the model that exercise induces RyR fragmentation. Moreover, albeit in a high fat model, others have also previously linked redox changes in RyR to calcium leak and mitochondrial biogenesis (Jain et al Diabetes 2014) further limiting the impact of the present data.

2- A major methodological limitation is directly supporting a link between RyR calcium leak and mitochondrial function. Attenuating calcium leak will also decrease SERCA-mediated calcium uptake,

and therefore ATP utilization. It is impossible with the current methodology to divorce the relationship between ATP utilization and the observed changes.

3- At the very least the authors should consider if Ca uptake into the SR is altered, ideally SERCA activity and associated regulators (i.e. total and phosphorylated phospholamban, SLN content and association with SERCA) don't change.

4- The authors have only reported PDH phosphorylation, and have concluded that moderate intensity exercise does not affect PDHp. However, this is in direct conflict with historical data which has shown PDH activity is increased rapidly during moderate intensity exercise, including low intensity exercise (eg. 35% VO₂ peak: Howlett et al AJP 1998). Moreover, the authors have not determined the contribution of PDK to this response, as ADP is supposed to inactivate PDK. The authors have also consider the JO₂ data in the context of PDH activation, however the authors do not have chemicals in their buffers to prevent changes in PDH phosphorylation. As a result, the addition of saturating ADP in the present in vitro assay would be expected to fully activate PDH, removing any regulation exerted by calcium. This makes it impossible to directly relate the JO₂ data to any possible in vivo changes in calcium.

5- Activation of PDH does not cause any metabolic change without providing an increase in substrate (why DCA activation of PDH in humans does not affect basal metabolism). Therefore, the authors need to consider their findings in a broader context, as mitochondrial metabolism relies on cytosolic metabolism, which is tightly regulated by free ADP, as opposed to calcium, again raising concerns that SERCA ATP utilization is the key regulatory point.

6- The authors have reported calstabin1 ratio to RyR instead of the physical interaction between them. The authors need to show some IP blots to affirm that SIT can reduce the physical interaction between calstabin1 with RyR (including all necessary positive and negative controls).

7- As stated above in the first paragraph, the authors need to revise their discussion to place the present findings in the context of historical data. For instance, sprint interval training requires several days/bouts of exercise to induce mitochondrial biogenesis (Perry et al JPHYS 2010), but the present data suggests this can happen much faster?

8- It is unclear if the electrical stimulation protocols replicated moderate and high intensity 'exercise'. The authors did not develop these protocols, but rather established them in their lab, as many laboratories have utilized electrical simulation protocols with cell culture preparations. The authors should provide sufficient references to justify their model, and provide ATP, PCr and Cr concentrations following the electrical stimulation protocols. The authors should also refrain from using the MICT and SIT acronyms when referring to C2C12 experiments.

9- Why did the authors determine 100 Hz before 10 and 1 Hz stimulations, and can the authors confirm the absence of twitch potentiation using this protocol? The authors need to report the absolute data for the 10 and 100 Hz stimulations.

10- In figure 1 is the OXPHOS protein data analysed with a 2 way ANOVA? If so the representative blots should be on the same membrane (not cut) and both pre values cannot be set to 100 %.

11- Why is the confocal signal brighter after SIT compared to MIT or control? Does calcium affect the fluorescence? What is the implication of that in the mitochondrial mass analysis?

Reviewers' comments:

Reviewer #1 (Remarks to the Author):

High intensity interval training (HIIT) including sprint interval training (SIT) is known to be an efficient type of exercise that can lead to even larger increases in mitochondrial biogenesis and VO₂max than classical moderate intensity exercise. Moreover, HIIT has gained interest both for athletes as well as a promising alternative for patient care. Here the authors show that one single bout of SIT leads to increased OXPHOS proteins and improved respiratory capacity. These findings are explained to be mediated by to RyR1 Ca²⁺ leak, a well described mechanism that some of the co-authors previously have shown to contribute to muscle weakness as well as normal ageing. As the authors claim, there is strong support that Ca²⁺ is involved in HIIT-induced musculoskeletal beneficial effects, however, to solely link it to a short-term and small RyR1 Ca²⁺ leak (since effects impeded by S107) appears too simplified.

We thank Reviewer 1 for their appreciation of our present and earlier work on the role of RyR1 Ca²⁺ leak in the beneficial effects of SIT. As described below, we have now performed relevant experiments using MCU inhibitor / genetic modifications, which brought more clarity to the role of RyR1 Ca²⁺ leak and mitochondrial Ca²⁺ uptake in the process of muscle adaptations to SIT. We now have submitted a revised version of our manuscript, which addresses all the comments from the Reviewers expecting that it now reaches the standards of the journal.

Comments:

SIT is a shorter and more intensive muscle stimulation, ie. the Ca²⁺ transients during this period, as well as the intracellular Ca²⁺ and stress on the OXPHOS system will also be higher during the stimulation period than for MICT. Thus, SIT will trigger reactive oxygen species (ROS) formation. The authors have detected ROS modifications on RyR1, which raises the question if an antioxidant would treatment have the same effect as S107?

We thank Reviewer 1 for this excellent point and question. We and several other authors have already reported that antioxidants can blunt the effects of exercise training on muscle aerobic phenotype and on exercise performance (see e.g.^{1,2}). Our aim here was to identify any causal link between an acute leaky RyR1 and the metabolic adaptations to SIT and identifying the signaling pathways behind the process. We refrained from using antioxidants in the present study since inhibiting ROS activity would have had other effects than only blunting RyR oxidation while S107 is more specific for the elucidation of the consequence of RyR1/FKBP12 dissociation.

We report consistent RyR1 oxidation and nitrosylation in response to SIT in human muscle as well as in cells in response to simulated SIT (S-SIT). We cannot just assume that RyR1 oxidation is the main mechanism leading to calstabin1 dissociation and leaky RyR1 in our study and had to look for a means to demonstrate the presumed link. This was possible with S107, which blocks the RyR1 Ca²⁺ leak and thus allowed us to document the potential causal link between RyR1 Ca²⁺ leak and mitochondrial adaptations.

Is S107 treatment causing decreased amount of post-translational modifications (DNP, SNO) of RyR1 after SIT?

We thank the Reviewer for this important question. S107 treatment does not alter the RyR modifications but improves FKBP12 association to the RyR^{3,4}. We now have investigated FKBP12 association to the RyR1 in response to S107 treatment. In line with previous reports, we observed that S107 treatment reverses FKBP12 dissociation from the RyR1 and confirmed that S107 does not alter the other SIT-induced RyR1 modifications (Figure 3A-D).

The authors show that one of the effects of the SIT-induced RyR1-mediated Ca²⁺ leak is enhanced mitochondrial Ca²⁺ uptake via MCU and PDH activation, which is contributing to enhanced OXPHOS. A concern is that the MCU complex is thought to have a very low affinity for Ca²⁺ (KD of 20–30 μM under physiological conditions). Thus, the intracellular Ca²⁺ concentration should be approximately 5–10 μM for considerable mitochondrial Ca²⁺ influx, which is high and unlikely unless that mitochondria are juxtaposed with the SR (e.g Rizzuto et al. 1998, Patergnani et al. 2011). I.e. microdomains with high Ca²⁺ concentrations ([Ca²⁺]_i>10 μM) is thought to form transiently in regions of close apposition between the mitochondria and the RyR1, ensuring a prompt accumulation of Ca²⁺ inside the mitochondria. Thus, the SIT stimulation could possible lead to the required magnitude of intracellular [Ca²⁺]_i, however, how can a small and transient RyR1-mediated Ca²⁺ leak lead to the required magnitude of intracellular [Ca²⁺]_i?

We thank Reviewer 1 for raising several relevant points. In the revised version of this manuscript, we have included new data supporting the contention that an acute SIT-induced Ca²⁺ leak from the RyR1 is sufficient to modify mitochondrial Ca²⁺ content. Rapamycin triggers FKBP dissociation from the RyR⁵ and in a previous study, rapamycin was successfully used to investigate whether skeletal muscle RyR1 Ca²⁺ leak could trigger mitochondrial Ca²⁺ uptake⁶. Therefore, we used rapamycin-induced RyR1 Ca²⁺ leak and measured mitochondrial Ca²⁺ uptake in our myotubes as a proof of concept of our hypothesis that acute Ca²⁺ leak from RyR1 is sufficient to alter mitochondrial Ca²⁺ levels. As shown in Figure S4K, rapamycin induced a quick and modest mitochondrial Ca²⁺ uptake that progressively reached a plateau. Caffeine, which opens the RyR1 channel, gave a rapid and stronger increase of mitochondrial Ca²⁺ (Figure S4J). The mitochondrial Ca²⁺ levels attained after rapamycin application were similar to those obtained with caffeine, with no additional response with caffeine stimulation.

We then used a MCU inhibitor (mitoxantrone or MTX) to investigate whether the mitochondrial Ca²⁺ accumulation in response to S-SIT was MCU-dependent. Our results (Figure 5E and 5G) showed a complete inhibition of the RyR1 Ca²⁺ leak-induced mitochondrial Ca²⁺ accumulation in response to S-SIT when cells were treated with MTX, as observed with the S107 treatment. Together these results support the hypothesis of RyR1 Ca²⁺ leak leading to mitochondrial Ca²⁺ uptake through the MCU.

Furthermore, the C2C12 cells are appropriate for genetical manipulation and thus would SIT-induced mitochondrial adaptation be blunted in MCU knockdown cells?

To address this question, we first used the MCU inhibitor mitoxantrone to block Ca²⁺ entry into the mitochondria after a S-SIT session. Unfortunately, a long-term use (72h) of mitoxantrone (initially used as an immunosuppressor acting as DNA intercalating agent, its

MCU-inhibitory effect was discovered later, and short-term treatment is recommended) was toxic for the cells. We therefore decided to adopt a si-RNA based approach, which was successful (Figure 6A-C). To avoid alteration of mitochondrial Ca^{2+} uptake during electrical stimulation and to specifically target the RyR1 Ca^{2+} leak effects, we performed MCU knockdown immediately after S-SIT stimulation. 72h post-stimulation and transfection, OXPHOS CI protein increase triggered in response to S-SIT stimulation was blunted in SIT si-MCU myotubes (Figure 6C and 6D). No significant effects were observed for the proteins from other mitochondrial complexes (see Figure 6C and 6E-H). These results are in agreement with our hypothesis of a specific role of RyR Ca^{2+} leak and mitochondrial Ca^{2+} uptake in the modulation of NADH-linked mitochondrial metabolism.

Page 3, reference 15 is a study on rodents not “healthy people”.

This was rectified.

Calstabin1 and calstabin 2 are for a large majority of scientist more known as FKBP12 and FKBP12.6, respectively, thus for an unfamiliar reader it must be clear that they are the same proteins.

We thank the Reviewer for this suggestion, we have now adopted throughout the text FKBP12 to indicate calstabin 1.

Many antibodies used in the manuscript are listed, but the antibodies used to detect DNP, CySNO and P-RyR1 are not listed in the table, please add.

These antibodies are now included in the chemical, reagents and antibodies table.

Serca2a is a slow-twitch isoform and the muscle biopsy was taken from vastus lateralis which is a mixed muscle type. Is the expression of fast-twitch type Serca1 also altered by SIT (or MICT)?

We have now completed our results with additional analyses. SERCA1 is also expressed in the human muscle samples even though the antibody gives a milder signal as compared to SERCA2 (Figure S1F and S1H). By expressing all the human data as percent of Pre MICT (taken as the reference), no significant differences were observed for SERCA2a and SERCA 1 between MICT and SIT (Figure S1F and S1I).

The immunoblots for OXPHOS protein levels are a cocktail of antibodies so it would be more suitable to present it all on one gel instead of cropped into several blots.

Although the OXPHOS protein levels are assessed using a cocktail of antibodies, each complex protein (i.e. band) gives a different intensity signal and requires a different intensity of exposure to obtain an optimal band with proper detection in each sample and no saturation. For this reason, we chose to crop the blots and present the images for each protein complex separately. However, we want to make clear that all cropped western blots results are part of the same gel. We have now clarified this in the legends and for clarity have now also provided images of the whole gels presented in the results section of the paper in the supplementary data section.

Moreover, to avoid any confusion, the protein bands are now shown following the original order on the gels.

Page 5, third paragraph “Both were unchanged…” but the authors refer to three different proteins.

We apologize for this typo. This sentence has been corrected.

The authors could not see any SIT-induced differences in PGC1a expression from human biopsies (Fig S1I) but in the C2C12 cells (Fig. S2M). The molecular weight marker is different between the two blots, please specify the molecular weight for the band analyzed. The predicted molecular weight for PGC1a1 is 90.4 kDa but it's a protein know to have manu post-translational modifications and apparent weight is commonly around 120 kDa. PGC1a antibodies are notoriously unreliable and thus running it together with a negative control is recommended. Moreover, PGc1a has four different isoforms a1-a4. Which isoforms were detected with the antibody? Calbiochem/Merck (ST1202) has an antibody that binds to the N-terminus that has been tested against KO tissue and able to detect several of the PGC1 isoforms.

Reviewer 1 raised several important points and suggested to use another PGC-1 antibody. We followed this advice and now have used the Calbiochem PGC-1 antibody together with a positive control (cold exposed brown adipose tissue) to clearly recognize the appropriate band. We observed a more consistent PGC-1 increase in human MICT and SIT muscles (Figure 1I and 1K), and PGC-1 was significantly increased in S-SIT myotubes and while it tended to increase in S-MICT myotubes 72h post stimulation (Figure S2J and S2L).

Moreover, with an antibody which is unreliable and given that PGC1a is an unstable protein with a short half-life, the mRNA levels are commonly used and a tested readout to examine effects of exercise. Were any SIT or MICT-induced alterations in mRNA levels observed?

Following the advice of the Reviewer concerning the anti-body we were able to show consistent induction of PGC1 protein in response to (S)-MICT and (S)-SIT with the new PGC-1 antibody and therefore refrained from determining mRNA levels.

Page 6, first paragraph appears very speculative. Intermitochondrial communication is present in skeletal muscle but unlike mitochondria involved in fusion events in cultured cells, muscle mitochondria do not appear to move out of position during presumed fusions (see review Lavorato et al 2020). With this in mind, what is the suggested role of increased OPA1 (long-isoform) after SIT?

OPA1 function in human muscle remains much debated. Using electron microscopy, it has been reported that sprint interval exercise induces rapid intramyofibrillar mitochondria fusion and elongation⁷. By expressing all the human data as percentage of the Pre values of MICT, OPA1 long forms are no more significantly increased but only tend to increase (Figure S1M-O). We have now modified this section of our manuscript.

Page 7, Fig 2B, RyR1-phosphorylation is not mentioned but was listed in Fig 1E. Were there no increase in P-RyR1 detected? If so, are only ROS mediated post-translational modifications involved in the downstream signaling observed after SIT?

We apologize for the lack of clarity in the original text. We did not observe a consistent RyR1 phosphorylation in our *in vitro* SIT model. As stated above, the RyR1 modifications shared by our human SIT muscles and *in vitro* S-SIT myotubes are RyR1 oxidation and nitrosylation. The text has been modified accordingly.

Page 7, caffeine stimulation in the absence of extracellular Ca²⁺ was used to show that SIT influences the SR Ca²⁺ release. However, this does not necessarily need to exclude a compensatory SIT-induced SOCE mechanism. E.g. Ivarsson et al. 2019 showed that voluntary running influences SOCE and STIM expression. Were there any SIT or MICT-induced SOCE observed?

We thank Reviewer 1 for this comment. We now investigated STIM1 expression after S-MICT and S-SIT stimulation in myotubes and did not observe any significant changes. We also investigated SOCE dynamics at the same time point as the measurements of SR Ca²⁺ release after S-MICT and S-SIT (Figure 2 M-0). CTRL, S-MICT and S-SIT all gave a similar SR Ca²⁺ release in response to 1 μM thapsigargin stimulation in the absence of extracellular Ca²⁺, even if the levels were slightly lower in S-SIT myotubes (Figure 2N). Addition of extracellular Ca²⁺ to the medium also induced similar SOCE in all conditions (Figure 2O). These results point towards an absence of SOCE participation in the SR Ca²⁺ handling investigated immediately post-stimulation with caffeine treatment. However, we cannot exclude that a long-term S-SIT or S-MICT exposure could alter the SOCE mechanism as observed in ⁸.

Page 8, Fig 2J-O, difficult that by eye detect any changes in expression.

The data originally presented in Figures 2J-O have been replaced by a more representative gel (now in Figure 3F-K).

Page 8, Fig 2P-R, “observed increased mitochondrial perimeter...” Please clarify what this observation reflects or what the physiological relevance is.

We thank the Reviewer for this question. Increased mitochondrial perimeter may suggest mitochondrial expansion. In support of the latter, mitochondria form factor and aspect ratio were not modified. Alternatively, it may be that the increase in mitochondria perimeter is related to mitochondria physiological swelling, a process described for mitochondrial Ca²⁺ accumulation, that has been linked to increased energy demand that stimulates ETC activity and OXPHOS ⁹.

We have now completed our experiments using another mitotracker, since the mitotracker red signal can be altered by the mitochondrial membrane potential. With the mitotracker green probe, that is not affected by mitochondrial membrane potential, we could confirm the results obtained with the mitotracker red (Figure S2N-P). We have adapted the text accordingly.

General for all immunoblots, it is troubling that the different groups Ctrl, MICT, SIT etc are cropped and separated and hence not compared on the same blot. Especially for proteins where there are small alterations in expression levels (e.g 1D, 1L, 2B, 2J, 4I, S1F, S1I, S1M, S2M) the comparisons between groups should be made from the same membrane/gel.

All the cropped western blots are part of the same gel and were cropped only to harmonize the order of the groups with the order shown on the graphs of the quantifications. We now provide the full gels from all the western blots images as supplemental material to show that samples were analyzed from the same gels. See also answer above.

Page 9, second paragraph “S107 acts by reinforcing the physical interaction between RyR1 and calstabin1, preventing or stopping Ca²⁺ leak”. Please add a blot that verifies that S107 treatment prevents SIT-induced calstabin (FKBP12) dissociation.

We thank the Reviewer for this suggestion. As requested, we now provide a blot showing that S107 prevents FKBP12 dissociation from the RyR1 (Figure 3 A-D).

Reviewer #2 (Remarks to the Author):

The present manuscript reports a thorough investigation of molecular and cellular responses to intense exercise (human muscle) or simulated exercise (myotubes, mouse muscle fibres) that aimed to elucidate the role of RyR1 calcium leak in exercise-induced skeletal muscle remodeling. In a series of experiments, the authors demonstrated that intense, but not “moderate,” exercise caused acute modifications to the RyR1 protein that reduce SR calcium content (i.e., elicit leak), increase indices of mitochondrial content, increase mitochondrial calcium uptake, and activate PDH. The authors demonstrated that an RyR1 stabilizer blunted responses to SIT in vitro. Overall, the authors suggest that their results provide evidence for a beneficial role of acute RyR1 calcium leak in response to exercise and that this response is exercise-intensity dependent.

Clearly, the authors have done a tremendous amount of work to support their conclusions. The choice of experiments and use of various models was appropriate to test the authors’ hypothesis, and I think the conclusions are supported by the results presented in their manuscript; however, I think the authors can improve their manuscript by clarifying aspects of the methods, particularly some details regarding the Western blotting, the electrical stimulation protocols, and how they determined their sample sizes. Adding some missing information will be helpful, but I recommend that the authors also be more careful when referring to exercise, indicating whether real exercise or electrical stimulation elicited specific results. Furthermore—and I apologize for making this general statement—I do not think the manuscript is particularly well written. I think a thorough check for small grammar mistakes (e.g., missing commas, odd phrasing, etc.) will help the authors convey their results better to the audience. Examples from the introduction include (i) the final sentence of paragraph two; (ii) the first sentence of paragraph three; (iii) the sentence about S107 in the third paragraph. Some additional commas to set off certain phrases and shorter sentences may help. Similar issues are present throughout the manuscript. Personally, I had to re-read multiple sentences to discern the point the authors were trying to make, and I could not always figure it out.

After re-reading my own comments, I wish to clarify that I see a lot of value in this manuscript, and I think the results are novel and likely to be of interest to several fields of research. The authors have done a commendable job. Yet, I still have major concerns about particular aspects of the manuscript that require clarification.

We are glad that the Reviewer 2 finds merit in our work and we were delighted to be offered the possibility to improve our manuscript. All the comments raised by the Reviewer have been addressed, as detailed below.

General

Throughout the results, I am unclear of how the authors decided on their reported sample sizes. For human muscle, the sample size is obvious, as it is the number of human participants. For other experiments, my impression is that replicates are counted as independent samples. For example, in Figure 2 (panel S), the authors indicated the sample size as “4-5 wells per group from 3 independent stimulated dishes.” Here, it reads as if the sample size is 3 (i.e., 1 per group), not 12-15 (i.e., 4-5 per group). Similarly, when 6 myotubes are examined from each of 3 independent stimulated dishes (Figure 2, panel I), I interpret this as 6 replicates of 3 samples (so an n of 3). My interpretation here is further supported by the authors language later in the manuscript: In Figure 4, the authors specify that 5-9 independent dishes were used per group and 3-6 myotubes were examined per dish. In this example, it seems that the independent dishes were the sample size and that myotubes were averaged to produce one value per dish. Thus, the authors seem to determine sample size differently depending on the experiment. Please clarify the sample size throughout, distinguishing between samples and replicates.

We thank the Reviewer 2 for their appreciation of our work and the time taken to carefully read our manuscript, and to raise constructive criticism and stimulating questions. We now clarified the sample sizes in a harmonious way throughout the manuscript. Independent experiments refer to different rounds of cultures (coming from different cell vials). The statistics have been adapted throughout the paper for clarity.

I appreciate that the authors have included a lot of data and many figures, but there seem to be some important mistakes. Specifically, some Western blot images appear to be duplicated within the manuscript. This is apparent in Figure 4, panel I (Ctrl vs. MICT are identical for PDH E1alpha) and Figure S1 panel F (MICT and SIT are identical for DHPR). Please correct these images.

We thank the Reviewer 2 for their comments and apologize for the lack of clarity about our representative blots. All the cropped parts of each blot were always from the same gel. Figure 4 panel I in the previous version indeed contained a mistake that we have corrected (now Figure 5 M-O). Also, in response to Reviewer 1 comments, we now present the protein bands in the same order of the original blots to avoid cropping. The whole blots are available in the Supplemental data.

Why were only human males included in this experiment? I don't expect the authors to re-do their study, but was there any reason to expect females not to respond to SIT?

The response to SIT might indeed vary between males and females, but this would then have required twice more participants to investigate the potential influence of sex. Then, for practical reasons (especially recruiting participants consenting to repeated muscle biopsy sampling), we only focused on males as we did in our previous investigations^{2,10,11}. These experiments should be repeated in the future with female participants to determine if their response to SIT is the same as that of males. It is worth noting that a recent study already

reported that the acute skeletal muscle response to SIT is largely similar in young women and men¹².

The need to clarify actual exercise from electrical stimulation *in vitro* may reflect my own bias for human research; however, I think it's important that the authors use specific language throughout the manuscript to indicate what results occurred in response to exercise in humans and what results occurred in response to electrical stimulation in cells derived from mice. The use of multiple models is a strength of the experiment; however, others may misinterpret the findings without greater attention to detail on the part of the authors. As an example, in the first sentence of the discussion, the authors choice of wording implies that mice performed SIT, which they did not. Given that this paragraph does not mention electrical stimulation, the reader may expect that these results were the result of actual exercise. The model to which specific results belong could be made more explicit, even if the overall conclusions remain the same.

We thank the Reviewer for making this important point. We now clarify the results related to every specific model in the text and systematically use the term “simulated SIT or simulated MICT stimulation” and the acronyms S-SIT and S-MICT when referring to the *in vitro* models, and have adapted the figures and text accordingly.

The other reason to make this distinction is that the electrical stimulation protocols seem to be somewhat arbitrary. Other than representing low and high intensity stimulations, I don't think it's possible to determine how well either represents SIT and MICT performed by humans. How did the authors choose their “MICT” protocol? At 1/25th the frequency, the difference between the “SIT” and “MICT” protocols seems much larger than would be apparent for human exercise, where power outputs are probably 2-3 times greater for SIT relative to MICT. This is important—as it is with human exercise trials—because if the authors choose an intensity that is too low, the “MICT” protocol would not be expected to elicit a response. Similarly, how was the “SIT” protocol chosen? While I see less issues here, it's possible that a high-intensity electrical stimulus that was lower than what the authors used (e.g., 40 Hz) may not have elicited the results presented herein.

Please explain how the protocols were chosen. Unless the authors have a strong reason for disagreeing, I think it should be made explicit that these two protocols do not necessarily represent MICT vs. SIT in humans and that the results are not generalizable to all comparisons of continuous and interval exercise. As an example, if 4 Hz was used for the “MICT” protocol, could similar results have been obtained for both intensities? Without more information from the authors, I do not think this possibility can be ruled out. Expressing some brief limitations to the generalizability of these results would be helpful.

The Reviewer raises an important point. Our approach was to find two protocols leading to comparable effects as those we observed after SIT and MICT in humans, while we agree that exercise mode may not be identical. Based on published stimulation data¹³ we used S-SIT and S-MICT while keeping the total duration of stimulation identical to what the human participants performed, with adapted stimulation patterns. The rationale behind this choice was to specifically investigate the role of a potent exercise-induced stress on the RyR1 modifications and the resulting muscle adaptations. In the end what counted for us was to find similar adaptations in the human and cell models regarding RyR1 PTMs and FKBP12 association, since the biological question we asked was linked to RyR1 Ca²⁺ leak. We have addressed these points in the revised version of the manuscript.

The authors do not comment on their previous finding of RyR1 fragmentation. Given the prominence of this finding in the previous PNAS paper, it seems odd not to mention that result here. Indeed, on page 3, the authors only refer to their mouse data from this manuscript. Similarly, the authors do not refer to another of their studies in human muscle from Schlittler et al. Both findings in human muscle seem relevant. Please explain why full-length RyR1 abundance was unchanged in the present human/cell experiments. What is the relevance of “fragmentation” as a post-translational modification?

We have previously reported that RyR1 fragmentation seems to depend on subject susceptibility (responders and non-responders). Here, we thus focus on FKBP12 dissociation from the RyR1 as it is a strong and consistent signature of a leaky RyR1 status. We now report the studies in the text with more details and discuss RyR fragmentation.

I do not fully understand the importance of the PDH dataset. The final sentence of the results section seems to implicate PDH as being necessary in the process of adaptation rather than as a biomarker of calcium uptake into the mitochondria (and perhaps an elevated capacity to oxidize carbohydrate *in vivo*). Are the authors making the point that the PDH response is somehow necessary for the mitochondrial adaptations, or am I confused? An entire paragraph of the discussion discusses PDH data, and the finding is included in the title; but the panel presented in figure 4 doesn't seem to extend beyond PDH being dephosphorylated: what happens next that is relevant for stimulating an increase in the aerobic capacity of muscle cells? I understand that the PDH data allows the authors to suggest increased mitochondrial calcium uptake in human muscle samples, but the myotube data seems to be a more direct measure of mitochondrial calcium uptake than PDH dephosphorylation. Please clarify.

We apologize for the lack of clarity regarding this aspect. We used PDH dephosphorylation mainly as a biomarker of the increased mitochondrial Ca^{2+} and especially to link the *in vitro* data to the human model. However, it has also been reported that PDH dephosphorylation (and thus activation) supports mitochondrial bioenergetics by increasing the capacity to metabolize NADH-linked substrates¹⁴. As our results mainly showed increased NADH-linked respiration, we thought it was important to discuss our PDH results in detail making this link explicit. Our new dataset reinforces our conclusions as S-SIT myotubes treated with S107 led to less NADH levels compared to S-SIT myotubes (Figure 3X). Moreover, mitochondrial Ca^{2+} uptake in response to S-SIT seems to play a role in mitochondrial adaptations as repressing MCU blunted the increase in OXPHOS complex I expression. Interestingly, a recent paper¹⁵ has reported a specific OXPHOS complex I subunit (NDUFB8) decrease in samples from patients showing MCU downregulation. We now have adapted the text accordingly to clarify this point in the manuscript.

Introduction

Page 3: The juxtaposition of HIIT being used in patient care and SIT being as effective as MICT gives the impression that SIT is being used in patient care. I do not think that SIT is commonly used in patient care. Please correct or provide evidence for this point.

The Reviewer is right, and the text has been adapted accordingly in the revised manuscript.

Page 3: Are SIT and MICT exercise “models”? This phrasing is a bit odd, particularly in a sentence that uses the word “models” in other contexts. Furthermore, how are human muscle biopsies a “model”?

We agree with the Reviewer. The text has been adapted accordingly in the revised manuscript

Page 3: Here, the authors state “Ser393;” however, this seems to be a typo (i.e., it should be 293).

Thank you for this observation, this has been corrected in the revised manuscript.

Results/Methods

Page 4: Reporting the total work performed as kJ/kg is uncommon for exercise physiology. I suggest reporting these numbers as kJ and also including the average power output (here or in the supplementary information).

The data are now reported as suggested in the revised manuscript.

Page 4 (and methods): The MICT program for human participants needs to be clarified. At what intensity was this exercise performed? 65% of VO₂peak or the mechanical associated with 65% of VO₂peak during the ramp incremental test? The mechanical power associated with 65% VO₂peak during the ramp incremental test would elicit a much higher VO₂ during constant load exercise (due to transit time delay and kinetics of VO₂). Did the authors measure VO₂ during the MICT effort to ensure that the chosen power output elicited 65% of VO₂peak? If so, please report this value. If 65% of the peak power output was used, as indicated in the methods, please correct the results section to reflect this point. Similarly, the methods section indicates 4 min of rest; however, the results reports a workload. Please clarify that the “rests” were active recovery (and provide power output). If I’ve misunderstood the authors here, please let me know.

The Reviewer is correct about these points. The intensity for MICT corresponded to 65% of the maximum aerobic power reached in the incremental VO₂max test. The intensity was fixed throughout the MICT session and likely was accompanied by some increase in VO₂ over time, but this was not monitored. The rest periods for SIT are recovery periods. The text has been adapted accordingly in the revised manuscript to clarify these points.

Page 4 (and methods): Please indicate the timing of post-exercise neuromuscular assessment: What was the delay from the cessation of exercise to the MVC? If exact values are not available, an approximation would still be helpful to the reader, as the timing is relevant to the central fatigue measurement.

The timing of the neuromuscular assessments post-exercise is now clarified in the revised manuscript.

Methods: What is the effect of centrifuging muscle samples at 10,000 rpm on the proteins of interest? Previously, Murphy and Lamb (2013) commented that many proteins, specifically the calcium-handling protein, calsequestrin, are lost in the cellular debris that is discarded

with this procedure. Would the use of centrifugation result in fractionation of the samples in the present manuscript?

The process of cell lysate is known to not always be optimal and the probability to have non-lysed cells in the mixture is real. Without centrifugation, it was thus difficult to get stable protein loading into the gels according to the protein quantification. This is the main reason why we have centrifuged our samples to harmonize our methods for the western blots. We have quantified the amount of protein lost in the pellet after our centrifugation protocol and were reassured that only little amounts of protein are lost in the pellet. (Please see figure below)

Figures: The authors report ponceau stains in their figures and indicate in the methods that these stains were quantified, but it is unclear how these images were used for normalization, if at all. Please clarify whether these images were visually inspected or quantified to correct for differences in loading. In general, the normalization of western blots is unclear. In the figure caption for Figure 1, it states that GAPDH was used to normalize OXPHOS proteins, but this isn't described in the methods. In Figure 2, OXPHOS western blots were not normalized to GAPDH? Why not, and how were they normalized?

When we started this study with the human muscle samples, we identified GAPDH as a stable protein at the investigated time points. We then used GAPDH as a loading control for the human data throughout. When we then continued with the *in vitro* models, we observed that GAPDH was not as stable as in the human tissue. We therefore decided to use quantified total protein (whole ponceau staining) in order to normalize the *in vitro* data in a robust way. We have now adapted the text accordingly.

Figures: For all Ponceau images, it's unclear what part of the membrane is being presented.

Can the authors indicate molecular weights or name the proteins if obvious (e.g., myosin or actin)? Is the same region of each membrane shown in all Ponceau images? If so, this could be stated in the methods.

We now specified the molecular weight range for the ponceau stainings (almost the whole gel from 15 to 250 kDa), shown for all blots in the revised manuscript.

Page 4: Were there positive/negative controls for the Co-IP procedure?

Yes, and these data are now shown in Figure 1D, Figure 2C and Figure 3A in the revised manuscript.

Methods: For Western blots, were replicates performed?

Yes, we performed at least 2 replicates per condition for our western blots. N now represents biological replicates from independent cultures for the cells.

Methods: What procedure was used to probe multiple proteins? Were proteins measured on separate membranes? Were membranes cut to allow for multiple proteins to be probed simultaneously? Were membranes stripped and probed sequentially? Does the LICOR system allow for multiple primary/secondary antibodies to be used simultaneously? A brief explanation in the methods would be helpful, given the reliance on western blotting results.

We thank the Reviewer for these questions. Our LiCor system allows the use of different primary and secondary antibodies. When an antibody is used for the first time, we always use it alone on a gel to verify that the band obtained is at the correct molecular weight before combining it to other characterized antibodies. At the same time, we can combine a rabbit and a mouse primary antibody detecting proteins at the same size in order to quantify the related signals in two different channels (700 and 800 nm infrared). The use of this system helps avoid stripping the membranes which is known to alter protein quantity. The methods section has now been clarified for this approach.

Figure 1: The use of bars and asterisks in the figures needs some clarification. For example, Panels I and J of Figure 1 seems to show 6 asterisks above the SIT bars, which I think should be two groups of three asterisks (as is shown for panel K in this figure). Regardless, the meaning of three asterisks has been omitted from the figure 1 caption (I found it elsewhere).

The asterisks on those figures were actually a group of 3 to compare two by two three different groups. We apologize for the confusion. We now clearly distinguish asterisks in the figures.

Methods: I apologize if I missed it, but it doesn't seem that electrical stimulation methods for "SIT" and "MICT" in mouse muscle fibres is reported in the text.

The electrical stimulation methods for "S-SIT" and "S-MICT" in mouse muscle cells and fibers are now reported in the revised methods.

Page 5: "NRF1" should be in parentheses

This has been corrected.

Page 5: I do not understand this sentence as written: “PGC-1a total protein, nuclear localization or post-translational modifications, including acetylation, regulate PGC-1a activity 30.” Specifically, how does the total abundance “regulate” activity of this protein? Perhaps combining this sentence with the sentence that follows would help clarify the point here.

The text has been adapted accordingly in the revised manuscript.

Page 7: I do not think the term “tended” is appropriate for a p-value that was greater than 0.05.

This has been addressed in the revised manuscript.

Figure 1 (and page 12): The authors point to the rapid increases in mitochondrial content from a single session of SIT, which from Figure 1 appear to be ~150-175% relative to pre. Such changes are comparable to (or greater) than what would be expected with months of exercise training. Do these changes in complex I, II, and IV protein content actually reflect changes in mitochondrial density when used in this context? Given that 2/5 OXPHOS proteins did not respond to the SIT stimulus, how did the authors decide that the overall result was “rapid beneficial mitochondrial adaptations”? Was any attempt made to measure respiration in these samples? I see that similar (and maybe larger) responses were observed in cell models, particularly for mitochondrial respiration. While the statistical difference between groups seems relevant, do the authors think that SIT is actually capable of increasing mitochondrial content in human muscle this dramatically?

We thank the Reviewer for making this important point. It has been reported that HIIT-induced mitochondrial adaptations would require several days/bouts of exercise¹⁶. However, like us, Trewin *et al.* (2018) reported changes in mitochondrial protein levels and respiration in response to single sessions of HIIT and MICT, despite a lower workload in HIIT¹⁷. Another recent study reported a rapid PGC1 protein nuclear translocation in response to SIT compared to MICT, also supporting rapid mitochondrial adaptations in response to SIT¹⁸, as observed in our study. The term rapid is here used according to the kinetics of the observed changes and was not related to the global changes. However, whether the observed rapid changes last and would be translated into continuing adaptations is unknown. Indeed, it has been reported that the first signal triggered by acute exercise cannot necessarily be extrapolated to repeated exercise benefits¹⁹. The human muscle biopsies were frozen immediately after collection and our attempt to measure the respiration states from those tissues failed. This kind of measurements would need a new human study with fresh tissue preparation.

Page 8: Some references to figures are a little confusing. This sentence refers to multiple figures with different types of data and proteins/genes other than PGC1a: “GC-1a protein was increased 72 h post SIT, but not MICT (Figures S2K-O).”

This sentence has been adapted accordingly in the revised manuscript.

Page 9: In this sentence, it reads to me as if S107 was not effective in blocking the effects of MICT, rather than MICT not being effective, compared to SIT. Please clarify: “S107 treatment inhibited the increase in mitochondrial OXPHOS proteins (Figures 3 B-G), mitochondrial fusion protein Mfn2 (Figures 3B and 3H) and mitochondrial perimeter and area (Figures 3 I-K) observed in response to SIT – but not MICT (Figures 3 B-H).”

The text has been adapted accordingly in the revised manuscript.

Figure 4: Images and graphs are missing for SIT S107. Furthermore, if measurements were taken where arrows indicate (i.e., 300s), it seems inappropriate to compare conditions after different amounts of time have passed since stimulation (e.g., MICT vs. SIT). In Figure 4C (SIT), the trace appears likely to decrease sharply if more time were provided. Here, the difference in stimulation duration (60s vs. 180s) does not reflect SIT vs. MICT for human exercise or other stimulation protocols used. Please clarify the figure and explain the timing and choice of stimulation procedures.

We thank the Reviewer for raising this relevant point. We have now adapted the mitochondrial Ca^{2+} curves to have quantification times similar for both S-MICT and S-SIT conditions after the end of the stimulations.

We would like to clarify that the goal of this set of experiments was to apply the S-MICT and S-SIT stimulation patterns in a defined timeframe in order to follow the kinetics of mitochondrial Ca^{2+} uptake *during* electrical stimulation and *after* it was stopped.

To this end, we used 1s cycle continuous time-lapse live imaging to cover the complete kinetics of the mitochondrial Ca^{2+} uptake and adapted the stimulation protocols to avoid photobleaching. As shown by our results, the S-MICT stimulation pattern showed progressively decreased mitochondrial Ca^{2+} levels during the stimulation in myotubes, a pattern that was more stable in FDB muscle fibers. But both models showed rapid decrease of the signal towards the baseline when the stimulation was stopped. When applying the S-SIT stimulation pattern, we observed a different response: after the first bout of stimulation, mitochondrial Ca^{2+} signal decreased but did not return to baseline and was actually sustained until the end of the protocol (Figure 5 C and 5J).

These results suggest greater mitochondrial Ca^{2+} accumulation with the S-SIT stimulation pattern, even upon the cessation of electrical stimulation. We agree this is not the full stimulation protocol. We therefore tested the hypothesis of a higher mitochondrial Ca^{2+} accumulation in myotubes submitted to the full stimulation protocol by quantifying a mitochondrial Ca^{2+} uptake marker, phospho PDH S²⁹³. Our results showed a decreased PDH phosphorylation in S-SIT myotubes compared to controls 1h after completing the stimulations. Interestingly, this was reversed by S107 and mitoxantrone (MCU blocker). These results were confirmed in human muscle samples post SIT, which showed a higher PDH dephosphorylation (Figure 5O and 5P).

The combined evidence of mitochondrial Ca^{2+} fluxes and PDH phosphorylation levels in human muscle and myotubes (with S107 and MCU manipulations) demonstrates that RyR1- Ca^{2+} leak induced mitochondrial Ca^{2+} uptake in response to (S)-SIT stimulation in myotubes.

Page 12: I think the proteomic analysis revealed decreased capacity for carbohydrate metabolism rather than a change in metabolism per se.

The text has been adapted accordingly in the revised manuscript.

Page 13: I also do not understand the authors' point about the second increase in mitochondrial calcium following stimulation (paragraph 2, discussion): is proteomic and respirometry data from the 72 h time point relevant to this acute increase in mitochondrial calcium content? Presumably it would have returned to baseline during the long recovery. In other words, was it necessary to demonstrate longer lasting responses related to PDH/complex I in order to confirm calcium uptake by the mitochondria following an acute stimulation of the myotubes? I apologize if this is my ignorance showing.

We thank the Reviewer for raising this important point. We performed investigations of the time course in mitochondrial adaptations while setting experimental conditions for this study. We identified that after a single stimulation protocol the positive adaptations (increased mitochondrial OXPHOS and respiration) are seen from 48h post stimulation and are maximal at 72h post; this is the reason why we focused on the 72h post stimulation time-point for all the mitochondrial adaptation responses. Our hypothesis was that PDH activation might trigger more NADH-linked substrates metabolism later on, a contention supported by our supercomplexes and respiration data. We agree with the Reviewer that it is difficult to directly link the PDH modifications to the adapted responses 72h post and now have adapted and clarified this in the text accordingly.

Page 13: Although I don't disagree, I think some references for the health-promoting benefits of exercise and for "today's recommendations" are needed.

We have now added more recent WHO references for the health- promoting benefits of exercise ^{20,21}.

Page 13: The S107 compound was not used in conjunction with a "maximum intensity variant of interval training." That this happened in myotubes isn't revealed for 3 more lines.

The text has been adapted accordingly in the revised manuscript.

Conclusion: I think this section provides a clear summary of the general results.

We thank the Reviewer for their appreciation of the summary.

Statistics: The authors should clarify which statistical test was used for which variable and when post hoc tests were performed. For ANOVA results, the p-values for main and interaction effects should be reported in supplementary information.

We now clarified in the legends all the statistical tests used to analyze our data. Details on main and interaction effects can be provided as a supplementary file in the final version of our manuscript.

The p-values for main and interaction effects are now reported in a table in supplementary information.

Reviewer #3 (Remarks to the Author):

The present paper by Zanou, Place and colleagues examines the interaction between RyR mediated calcium leak and mitochondrial respiratory capacity. The authors working hypothesis is that calcium leak is important for exercise adaptations, and furthermore that sprint interval training more robustly induces these changes. The present model is an extension of the authors previous work in PNAS showing SIT induced RyR fragmentation, and therefore the novelty is limited to the comparison between endurance and SIT exercise, and is somewhat limited in scope. In addition, previous work has shown that endurance and SIT equally induce mitochondrial biogenesis (see seminal work by Burgomaster and Gibala) and signals related to mitochondrial biogenesis (Bartlett et al JAP 2012), challenging the fundamental premise and importance of the data reported here. The overall discussion surrounding classical exercise data is limited in scope, and the authors are encouraged to take a more balanced approach to placing their data in context.

We thank the Reviewer for the time taken to carefully read our paper and to raise these relevant points. We apologize for our lack of clarity. In the PNAS paper we reported RyR1 fragmentation in response to SIT exercise, but no link was made between the RyR1 fragmentation, potential RyR1 Ca²⁺ leak and mitochondrial adaptations (no OXPHOS proteins, mitochondrial mass and function measurements). Moreover, (chronic) RyR1 Ca²⁺ leak is generally reported to be detrimental^{3,6}. The present study thus aimed at establishing a possible causality between (acute) RyR1 Ca²⁺ leak and positive/beneficial mitochondrial adaptations in physiological conditions. We demonstrate here for the first time a causal link between RyR1 Ca²⁺ leak and increased muscle mitochondrial complex I-driven adaptations in response to SIT, an original result that has not been reported before.

Major comments:

1- The working model is not novel as there are many papers examining the mobilization of calcium from the SR to mitochondria during contraction (PMID: 29988564; PMID: 21106237), which decreases the novelty of the study. The authors themselves have previously supported the model that exercise induces RyR fragmentation. Moreover, albeit in a high fat model, others have also previously linked redox changes in RyR to calcium leak and mitochondrial biogenesis (Jain et al Diabetes 2014) further limiting the impact of the present data.

We thank the Reviewer for this comment and apologize again for the lack of clarify about the focus and novelty of the present work. Some papers indeed investigated Ca²⁺ mobilization from the SR to mitochondria *during contractions*. Our study rather focused on RyR1 Ca²⁺ leak (*a process that remains after the contraction has ended*) induced beneficial effects in physiological conditions. This role of RyR1 Ca²⁺ leak on SIT mitochondrial adaptations that we report is novel. The Jain et al. (2014) paper focused on the role of high fat diet-induced mitochondrial ROS production on mitochondrial biogenesis rather than on elucidating any causal link between RyR1 Ca²⁺ leak and mitochondrial adaptations (as is the case in our report).

2- A major methodological limitation is directly supporting a link between RyR calcium leak and mitochondrial function. Attenuating calcium leak will also decrease SERCA-mediated

calcium uptake, and therefore ATP utilization. It is impossible with the current methodology to divorce the relationship between ATP utilization and the observed changes.

Please see our answer below.

3- At the very least the authors should consider if Ca uptake into the SR is altered, ideally SERCA activity and associated regulators (i.e. total and phosphorylated phospholamban, SLN content and association with SERCA) don't change.

We thank the Reviewer for the important points raised in comments 2 and 3. We have investigated phospholamban and phospho-phospholamban Ser¹⁶ protein levels in our samples but obtained very weak non-quantifiable signals. We therefore performed a co-immunoprecipitation assay of SERCA-Sarcoplipin in CTRL, S-SIT and S-SIT S107 myotubes at 72h post stimulation and found no differences between groups (Figures S4A and S4D). We also isolated microsomes (with enriched SR fraction) to directly investigate SERCA activity, using an ATPase assay. We observed that SERCA activity was higher in S-SIT myotubes compared to CTRL and S-SIT S107 72h post stimulation (Figure S4E and S4H). At the same time point, we also measured ATP production in the whole cell lysates and observed increased ATP levels in both S-SIT and S-SIT S107 myotubes compared to controls (Figure S4I). These results indicate that despite a higher ATP utilization at the SR in S-SIT conditions, cellular ATP levels were still high, which suggests ample capacity of the S-SIT myotubes to cover all their ATP needs. We are confident that these new results adequately address the concern of the Reviewer.

Also, pharmacological inhibition of the mitochondrial Ca²⁺ uniporter (MCU) blunted the mitochondrial Ca²⁺ uptake observed in response to S-SIT (Figure 5E and 5G) and prevented the PDH dephosphorylation induced by S-SIT (Figure 5M and 5N), as observed with S107 treatment (Figure 5D, 5G, 5M and 5N). In addition, knockdown of the MCU protein after a S-SIT session specifically blunted the Complex I protein increase triggered by S-SIT (Figure 6C and 6E).

Together, these results point to a specific role of RyR1 Ca²⁺ leak in mitochondrial adaptations in response to S-SIT.

4- The authors have only reported PDH phosphorylation, and have concluded that moderate intensity exercise does not affect PDH_p. However, this is in direct conflict with historical data which has shown PDH activity is increased rapidly during moderate intensity exercise, including low intensity exercise (eg. 35% VO₂ peak: Howlett et al AJP 1998). Moreover, the authors have not determined the contribution of PDK to this response, as ADP is supposed to inactivate PDK.

We thank the Reviewer for this pertinent point. We apologize that our reasoning was not clear enough in the manuscript. We completely agree about PDH dephosphorylation *during* moderate intensity exercise, a well-described phenomenon. We are here referring to the dephosphorylation observed *after* the exercise that we suspected to be linked to the RyR1 Ca²⁺ leak and that we only observed in our S-SIT conditions. *Again, we would like to clarify the dissociation between the phenomena that happen during exercise per se from those that occur after exercise.*

The authors have also consider the JO₂ data in the context of PDH activation, however the authors do not have chemicals in their buffers to prevent changes in PDH phosphorylation. As

a result, the addition of saturating ADP in the present in vitro assay would be expected to fully activate PDH, removing any regulation exerted by calcium. This makes it impossible to directly relate the JO₂ data to any possible in vivo changes in calcium.

We fully agree with the Reviewer's comment. We performed the respiration measurements 72h post stimulation in a saturating substrate environment and could therefore not directly link our O₂-consumption data to the changes observed with the PDH investigated 1h post-stimulation. To complement our findings, we now report NADH/NAD levels in S-SIT and S-SIT S107 myotubes at the time-point of the respiration measurements (72h post stimulation) and observed a significant decrease in NADH levels in S-SIT S107 myotubes compared to S-SIT (Figure 3X). These new results are in support of the possible contribution of PDH dephosphorylation (and thus activation) to altered mitochondrial bioenergetics.

5- Activation of PDH does not cause any metabolic change without providing an increase in substrate (why DCA activation of PDH in humans does not affect basal metabolism). Therefore, the authors need to consider their findings in a broader context, as mitochondrial metabolism relies on cytosolic metabolism, which is tightly regulated by free ADP, as opposed to calcium, again raising concerns that SERCA ATP utilization is the key regulatory point.

We thank the Reviewer for this comment. Our new data reported above now shows increased NADH substrates in S-SIT compared to S-SIT S107 myotubes that we believe support the metabolic change induced by PDH dephosphorylation in S-SIT conditions.

We do agree with the Reviewer about the needed caution for data interpretation in a broader context and have now adapted the manuscript accordingly.

6- The authors have reported calstabin1 ratio to RyR instead of the physical interaction between them. The authors need to show some IP blots to affirm that SIT can reduce the physical interaction between calstabin1 with RyR (including all necessary positive and negative controls).

We apologize about the misunderstanding about the RyR1/ FKBP12 dataset. It was indeed co-immunoprecipitation data that we reported in the manuscript, indicating the physical interaction between RyR1 and FKBP12. We now have put an explicit mention on all co-IP assays to avoid further confusion. Moreover, all the negative and positive controls have now been added to the co-IPs (Figure 1D; Figure 2C; Figure 3A).

7- As stated above in the first paragraph, the authors need to revise their discussion to place the present findings in the context of historical data. For instance, sprint interval training requires several days/bouts of exercise to induce mitochondrial biogenesis (Perry et al JPHYS 2010), but the present data suggests this can happen much faster?

We thank the Reviewer for their comment. After using a PGC-1 α antibody (targeting the N terminal of the protein thus recognizing more PGC1 α isoforms) as proposed by Reviewer 1, we observed that both MICT and SIT increased PGC1-1 α protein after just one bout of exercise, which strongly suggests that adaptations to exercise can occur very quickly. However, only SIT led to a significant increase of mitochondrial OXPHOS proteins. Whether these modifications are long-lasting was not examined in our study. We here considered these

signals as a proof for initiation of mitochondrial adaptations in response to a single bout of exercise. This beneficial adaptation may indeed be sustained and/or amplified by repeating the exercise pattern.

8- It is unclear if the electrical stimulation protocols replicated moderate and high intensity 'exercise'. The authors did not develop these protocols, but rather established them in their lab, as many laboratories have utilized electrical stimulation protocols with cell culture preparations. The authors should provide sufficient references to justify their model, and provide ATP, PCr and Cr concentrations following the electrical stimulation protocols. The authors should also refrain from using the MICT and SIT acronyms when referring to C2C12 experiments.

We thank the Reviewer for their comment. Indeed, different *in vitro* electrical stimulation protocols have been reported in the literature. However, we have seen no S-SIT protocol as described in our study in the literature. We started from the human SIT paradigm to develop a S-SIT protocol in myotubes, aiming to find *a model that reproduces the changes that occurred in our human volunteers*. We were therefore quite satisfied that our S-SIT protocol in myotubes could show RyR1 Ca²⁺ leak status and similar mitochondrial adaptations as seen in the human samples, which greatly helped investigating the causal link between RyR1 Ca²⁺ leak and the mitochondrial adaptations in response to (S)-SIT. There indeed are many different MICT protocols in the literature that helped designing our MICT model.

To further provide arguments in favor of our models we have now performed a metabolomic analysis on our *in vitro* S-MICT and S-SIT models immediately after stimulation. The AMP/ATP and ADP/ATP ratios were increased in S-SIT myotubes compared to S-MICT (Figure 2B), which supports the increased P AMPK levels in S-SIT compared to S-MICT in our study, as also reported in the literature¹⁷ (Figure S2C and S2D).

We did not find any relevant changes in PCr/Cr concentrations (Figure 2B), whose dynamics are known to be tightly regulated during exercise and recovery. We could also investigate PCr levels after the first 30s of stimulation²² but this condition would not help characterizing the full phenotype of our S-MICT and S-SIT protocols. Our metabolomics data for PCr and Creatine are thus difficult to interpret, and we would appreciate any advice from the Reviewer to help further better characterizing our cellular models.

It is worth noting that our *in vitro* model nicely recapitulates our main observations on human muscle biopsies (RyR1 / FKBP12 dissociation and OXPHOS increase, as well as PDH dephosphorylation), and thus constitutes a suitable model for the mechanistic investigations.

9- Why did the authors determine 100 Hz before 10 and 1 Hz stimulations, and can the authors confirm the absence of twitch potentiation using this protocol? The authors need to report the absolute data for the 10 and 100 Hz stimulations.

We thank the Reviewer for this comment. The reported 100 Hz and 10 Hz forces are absolute data (see Figure S1 D and E). The sequence we used (100 Hz, 10 Hz, single twitch) was the same as we previously adopted (e.g.^{10,11}), which was validated to study prolonged low frequency force depression in resting and fatigued conditions. As in our previous investigations, all stimulations were delivered after maximal voluntary contractions to obtain fully potentiated responses and thus avoided the potential confounding effect of potentiation.

10- In figure 1 is the OXPHOS protein data analysed with a 2 way ANOVA? If so the representative blots should be on the same membrane (not cut) and both pre values cannot be set to 100 %.

The OXPHOS data are from the same membrane and were cut only to align the signal to the quantification bars. We now clarified this point. For technical reasons, the OXPHOS proteins are not all scanned at the same intensity because they require different exposure intensities, reason why we show separately each protein. However, for transparency the full representative blots are now shown in the supplementary information. We have also now addressed the Reviewer's comment by expressing all the human data relative to the Pre values of MICT, taken as the reference (100%) in order to properly perform the ANOVA test.

11- Why is the confocal signal brighter after SIT compared to MIT or control? Does calcium affect the fluorescence? What is the implication of that in the mitochondrial mass analysis?

The brighter mitochondrial Rhod-2 signal indicates that there were some Ca²⁺ fluxes during the stimulations as the probe is Ca²⁺ sensitive. We have now adapted this set of data for clarity (Figure 5A-J).

- 1 Gomez-Cabrera, M. C., Ristow, M. & Vina, J. Antioxidant supplements in exercise: worse than useless? *Am J Physiol Endocrinol Metab* **302**, E476-477; author reply E478-479, doi:10.1152/ajpendo.00567.2011 (2012).
- 2 Wyckelsma, V. L. *et al.* Vitamin C and E Treatment Blunts Sprint Interval Training-Induced Changes in Inflammatory Mediator-, Calcium-, and Mitochondria-Related Signaling in Recreationally Active Elderly Humans. *Antioxidants (Basel)* **9**, doi:10.3390/antiox9090879 (2020).
- 3 Bellingr, A. M. *et al.* Hypernitrosylated ryanodine receptor calcium release channels are leaky in dystrophic muscle. *Nat Med* **15**, 325-330, doi:10.1038/nm.1916 (2009).
- 4 Matecki, S. *et al.* Leaky ryanodine receptors contribute to diaphragmatic weakness during mechanical ventilation. *Proc Natl Acad Sci U S A* **113**, 9069-9074, doi:10.1073/pnas.1609707113 (2016).
- 5 Kaftan, E., Marks, A. R. & Ehrlich, B. E. Effects of rapamycin on ryanodine receptor/Ca(2+)-release channels from cardiac muscle. *Circ Res* **78**, 990-997, doi:10.1161/01.res.78.6.990 (1996).
- 6 Andersson, D. C. *et al.* Ryanodine receptor oxidation causes intracellular calcium leak and muscle weakness in aging. *Cell Metab* **14**, 196-207, doi:10.1016/j.cmet.2011.05.014 (2011).
- 7 Huertas, J. R., Casuso, R. A., Agustin, P. H. & Cogliati, S. Stay Fit, Stay Young: Mitochondria in Movement: The Role of Exercise in the New Mitochondrial Paradigm. *Oxid Med Cell Longev* **2019**, 7058350, doi:10.1155/2019/7058350 (2019).
- 8 Ivarsson, N. *et al.* SR Ca(2+) leak in skeletal muscle fibers acts as an intracellular signal to increase fatigue resistance. *J Gen Physiol* **151**, 567-577, doi:10.1085/jgp.201812152 (2019).
- 9 Javadov, S., Chapa-Dubocq, X. & Makarov, V. Different approaches to modeling analysis of mitochondrial swelling. *Mitochondrion* **38**, 58-70, doi:10.1016/j.mito.2017.08.004 (2018).

- 10 Place, N. *et al.* Ryanodine receptor fragmentation and sarcoplasmic reticulum Ca²⁺ leak after one session of high-intensity interval exercise. *Proc Natl Acad Sci U S A* **112**, 15492-15497, doi:10.1073/pnas.1507176112 (2015).
- 11 Schlittler, M. *et al.* Three weeks of sprint interval training improved high-intensity cycling performance and limited ryanodine receptor modifications in recreationally active human subjects. *Eur J Appl Physiol* **119**, 1951-1958, doi:10.1007/s00421-019-04183-w (2019).
- 12 Skelly, L. E. *et al.* Effect of sex on the acute skeletal muscle response to sprint interval exercise. *Exp Physiol* **102**, 354-365, doi:10.1113/EP086118 (2017).
- 13 Nikolic, N. *et al.* Electrical pulse stimulation of cultured skeletal muscle cells as a model for in vitro exercise - possibilities and limitations. *Acta Physiol (Oxf)* **220**, 310-331, doi:10.1111/apha.12830 (2017).
- 14 Glancy, B. & Balaban, R. S. Role of mitochondrial Ca²⁺ in the regulation of cellular energetics. *Biochemistry* **51**, 2959-2973, doi:10.1021/bi2018909 (2012).
- 15 Ghosh, S. *et al.* An essential role for cardiolipin in the stability and function of the mitochondrial calcium uniporter. *Proc Natl Acad Sci U S A* **117**, 16383-16390, doi:10.1073/pnas.2000640117 (2020).
- 16 Perry, C. G. *et al.* Repeated transient mRNA bursts precede increases in transcriptional and mitochondrial proteins during training in human skeletal muscle. *J Physiol* **588**, 4795-4810, doi:10.1113/jphysiol.2010.199448 (2010).
- 17 Trewin, A. J. *et al.* Acute HIIE elicits similar changes in human skeletal muscle mitochondrial H₂O₂ release, respiration, and cell signaling as endurance exercise even with less work. *Am J Physiol Regul Integr Comp Physiol* **315**, R1003-R1016, doi:10.1152/ajpregu.00096.2018 (2018).
- 18 Granata, C., Oliveira, R. S., Little, J. P., Renner, K. & Bishop, D. J. Sprint-interval but not continuous exercise increases PGC-1 α protein content and p53 phosphorylation in nuclear fractions of human skeletal muscle. *Sci Rep* **7**, 44227, doi:10.1038/srep44227 (2017).
- 19 Granata, C., Oliveira, R. S. F., Little, J. P. & Bishop, D. J. Forty high-intensity interval training sessions blunt exercise-induced changes in the nuclear protein content of PGC-1 α and p53 in human skeletal muscle. *Am J Physiol Endocrinol Metab* **318**, E224-E236, doi:10.1152/ajpendo.00233.2019 (2020).
- 20 WHO. *WHO guidelines on physical activity and sedentary behaviour*, <<https://apps.who.int/iris/rest/bitstreams/1315866/retrieve> internal-pdf://3829176424/9789240015128-eng.pdf> (2020).
- 21 Pedersen, B. K. & Saltin, B. Exercise as medicine - evidence for prescribing exercise as therapy in 26 different chronic diseases. *Scand J Med Sci Sports* **25 Suppl 3**, 1-72, doi:10.1111/sms.12581 (2015).
- 22 Hargreaves, M. & Spriet, L. L. Skeletal muscle energy metabolism during exercise. *Nat Metab* **2**, 817-828, doi:10.1038/s42255-020-0251-4 (2020).

REVIEWER COMMENTS

Reviewer #1 (Remarks to the Author):

The authors have added an impressive number of new experiments in the revised version of the manuscript and many of the experiments support their initial study. No further comments but want to mention that it reads well with changing into the FKBP nomenclature.

Reviewer #2 (Remarks to the Author):

Thank you to the authors for addressing my comments and providing additional details. The added experiments, more detailed methods, and general revisions to the manuscript have improved the clarity of the authors' work. The data support the conclusions reached by the authors, and I have relatively few comments/suggestions at this point.

1. The authors should add references to their broad statements about exercise. For example, I disagree with this statement: "...today's recommendations suggest that any episode of physical activity conveys benefits..." Low intensities of exercise in particular are not effective, particularly for those who are already fit. Similarly, the authors should cite their statement about physical inactivity being one of the top health risks worldwide (it's true, but a source is needed).

2. In terms of statistics, the authors should indicate when post hoc testing was applied, as it is inappropriate to perform unless certain criteria are met with the ANOVA (e.g., main effects or interaction effects, depending on the design and the post hoc analyses).

3. There is a typo in the introduction related to the term "reason why" (some words are missing for this to read clearly).

4. The abbreviation "(PDH)" is put after "phosphorylation" but it should be after the enzyme name.

5. The lanes in Figure 1D do not line up, suggesting the PTMs and the RyR1 shown are not from the same samples (i.e., the post-SIT bands for P, DNP, and CYS NO are not aligned with the RYR1 bands). Please clarify.

6. On page 9, I suggest the authors not use the term "tended" for a non-significant difference.

7. Sarcolipin is defined after the abbreviation is used. A small revision would make this clearer.

8. For the PDH blots with human muscle, the change shown in the representative blot doesn't seem to appear in the bar graph (Fig. 5P), as the largest reduction in PDH phosphorylation is ~50%, but the SIT response in the representative image appears much larger. It would be helpful for my interpretation of the blot to know the fraction of phosphorylated to total protein that was derived from this image.

Reviewer #3 (Remarks to the Author):

The novelty remains debatable. The Jain et al paper showed RyR nitrosylation occurred with CaMK phosphorylation, as well used muscle incubations to show that caffeine and H₂O₂ caused CaMK phosphorylation, processes prevented by dantrolene, an RyR inhibitor. These data provide evidence that redox changes of RyR cause CaMK phosphorylation, presumably through the induction of calcium leak. This paper also showed this 'positively' resulted in the accumulation of mitochondrial proteins. While this paper relied on high fat diet and obese models, the fundamental notion that post-translational modifications of RyR cause 'beneficial' responses has been previously observed. In addition, the previous report in muscle specific MCU^{-/-} shows the effect on fuel utilization and the effect of mitochondrial calcium. While the authors have generated a lot of data in present manuscript, arguably the authors are simply combining previous reports into a single paper examining the impact of exercise on these mechanisms. However, it is not clear if this is sufficient to warrant publication in Nature, especially since the seminal papers on PGC1 activation involved caffeine-mediated gene-transcription and nuclear PGC1a translocation (Wright et al JBC). Incidentally, it is not clear why the authors have not measured CaMK or p38MAPK phosphorylation.

At the very least the authors need to acknowledge this previous work and provide a balanced discussion.

I previously made mention that the discussion was not balanced, as endurance exercise has classically been shown to similarly induce mitochondrial adaptations, however the discussion has not been revised. The authors state 'SIT leads to similar or even larger increases in maximum aerobic capacity (VO_{2peak}) in healthy populations as compared to classical MICT due to more pronounced skeletal muscle mitochondrial remodeling', however the 'classical' MICT involves 5-days/week of training and this volume of training results in a different interpretation (again, see seminal work from Dr Gibala's laboratory).

The authors were asked about positive and negative controls following IP experiments, but these remain missing. If the authors IP RyR they should blot for 'non-RyR' proteins (e.g. CSQ, DHPR, SERCA).

The authors provide cellular NADH/NAD⁺, however it is not clear what these data tell the reader about mitochondrial metabolism, as cytosolic NAD⁺ levels are estimated to be 1000-fold higher than mitochondrial.

REVIEWER COMMENTS

Reviewer #1 (Remarks to the Author):

The authors have added an impressive number of new experiments in the revised version of the manuscript and many of the experiments support their initial study. No further comments but want to mention that it reads well with changing into the FKBP nomenclature.

We thank the reviewer for their appreciation on the revised version of our manuscript. With regard to FKBP nomenclature, in the present revision of the manuscript we have reverted to the use of the term calstabin1 to clearly indicate that we refer to the specific pool of FKBP12. We extensively discussed this point among the authors before taking this decision and apologize for the to and fro.

The reasons behind using the term calstabin1 instead of FKBP12 are the following. FK506 binding proteins (FKBPs) are a large family of proteins that possess peptidyl prolyl *cis/trans* isomerase domains. Interestingly, a specific pool of FKBP12 is located in the triad junction between the sarco/endoplasmic reticulum and the T-tubule. These FKBP12s bind to ryanodine receptors (RyRs) calcium channels to stabilize it in its closed state. Depletion of FKBP12 from RyRs causes leaky RyR channels by failing to close properly. Thus, the role of this pool of FKBP12 is quite different from that of the cytosolic pool of the protein, which mediates the immunosuppressive action of the FK506 drug. Since the discovery of this location and the associated functional differences, the FKBP12 in this reticular pool was named calstabin for Calcium Channel Stabilizing Binding protein with respect to its role in stabilizing the RyR calcium channels. Of note, the skeletal isoform is known as calstabin1 and the cardiac isoform as calstabin2. When reporting on its implication in RyR regulation the term calstabin has become the term used by the scientific community, as it describes this specific role of the protein, and prevents confusion with its implication in a immunosuppressive pathway. Making this distinction by using the term calstabin has now become an accepted practice in the field, as the term calstabin, used in this way, has been reported in at least 48 publications.

Reviewer #2 (Remarks to the Author):

Thank you to the authors for addressing my comments and providing additional details. The added experiments, more detailed methods, and general revisions to the manuscript have improved the clarity of the authors' work. The data support the conclusions reached by the authors, and I have relatively few comments/suggestions at this point.

1. The authors should add references to their broad statements about exercise. For example, I disagree with this statement: "...today's recommendations suggest that any episode of physical activity conveys benefits..." Low intensities of exercise in particular are not effective, particularly for those who are already fit. Similarly, the authors should cite their statement about physical inactivity being one of the top health risks worldwide (it's true, but a source is needed).

We thank the reviewer for their remark. We now provided several references (Erikssen et al, Lancet, 1998; Blair et al, JAMA, 1995) supporting our statements on the modalities and benefits of physical activity.

2. In terms of statistics, the authors should indicate when post hoc testing was applied, as it is inappropriate to perform unless certain criteria are met with the ANOVA (e.g., main effects or interaction effects, depending on the design and the post hoc analyses).

We thank the reviewer for raising this point. As requested, we now indicated when post-hoc testing was used and report the criteria met.

3. There is a typo in the introduction related to the term "reason why" (some words are missing for this to read clearly).

We corrected the typo.

4. The abbreviation "(PDH)" is put after "phosphorylation" but it should be after the enzyme name.

This was corrected.

5. The lanes in Figure 1D do not line up, suggesting the PTMs and the RyR1 shown are not from the same samples (i.e., the post-SIT bands for P, DNP, and CYS NO are not aligned with the RYR1 bands). Please clarify.

We thank the reviewer for this relevant observation. The tiff images from the different western blot images (RyR1-calstabin1 co-IP and RyR1 PTMs) were just not well aligned according to the samples. This is now corrected and a whole gel is provided in Figure S1G.

6. On page 9, I suggest the authors not use the term "tended" for a non-significant difference.

This was corrected.

7. Sarcolipin is defined after the abbreviation is used. A small revision would make this clearer.

This was corrected.

8. For the PDH blots with human muscle, the change shown in the representative blot doesn't seem to appear in the bar graph (Fig. 5P), as the largest reduction in PDH phosphorylation is ~50%, but the SIT response in the representative image appears much larger. It would be helpful for my interpretation of the blot to know the fraction of phosphorylated to total protein that was derived from this image.

We thank the reviewer for this observation. The change shown in the representative blot was from the subject with the lowest PDH phosphorylation levels (47 %). We now provided another gel that fits better with the mean values of each group.

Reviewer #3 (Remarks to the Author):

The novelty remains debatable. The Jain et al paper showed RyR nitrosylation occurred with CaMK phosphorylation, as well used muscle incubations to show that caffeine and H₂O₂

caused CaMK phosphorylation, processes prevented by dantrolene, an RyR inhibitor. These data provide evidence that redox changes of RyR cause CaMK phosphorylation, presumably through the induction of calcium leak. This paper also showed this 'positively' resulted in the accumulation of mitochondrial proteins. While this paper relied on high fat diet and obese models, the fundamental notion that post-translational modifications of RyR cause 'beneficial' responses has been previously observed. In addition, the previous report in muscle specific MCU^{-/-} shows the effect on fuel utilization and the effect of mitochondrial calcium. While the authors have generated a lot of data in present manuscript, arguably the authors are simply combining previous reports into a single paper examining the impact of exercise on these mechanisms. However, it is not clear if this is sufficient to warrant publication in Nature, especially since the seminal papers on PGC1 activation involved caffeine-mediated gene-transcription and nuclear PGC1 α translocation (Wright et al JBC). Incidentally, it is not clear why the authors have not measured CaMK or p38MAPK phosphorylation.

At the very least the authors need to acknowledge this previous work and provide a balanced discussion.

We appreciate the reviewer's comments and agree that novelty, in general, is a debatable notion. We all start from what was known and build upon that base. We regret that our previous discussion on this dataset in the rebuttal letter was not sufficient to convince the reviewer that our results represent a quite significant step forward, pointing towards promising follow-up science of potential pharmacological relevance. We now duly acknowledge and discuss the Jain et al. (2014) paper in our manuscript. Also, even though it was not the focus of our work, we have now measured the phosphorylation levels of CaMKII in the human muscle biopsies and our cellular models of exercise. We observed that CaMKII phosphorylation is increased both after SIT and MICT, in human and cells as well (see the new supplemental Figures S1K, S1L, S2L and S2N).

These results are in line with the increased PGC1 α levels in our study in both SIT and MICT groups. Indeed, Wright et al. (2007) reported that increased cytosolic Ca²⁺ activates CaMKII phosphorylation, which in turn phosphorylates p38 to increase PGC1 expression for mitochondrial biogenesis.

Since we observed a similar increase of CaMKII phosphorylation and PGC1 expression in both SIT and MICT, this argues for a CaMKII-independent mechanism supporting the RyR1 Ca²⁺ leak effects specifically observed in response to SIT.

We therefore interpret our results as showing a specific effect of RyR1 Ca²⁺ leak (and not resulting from a global cytosolic Ca²⁺ increase occurring during exercise, and the subsequent CaMKII activation) on mitochondrial Ca²⁺ uptake and PDH activation in response to SIT.

I previously made mention that the discussion was not balanced, as endurance exercise has classically been shown to similarly induce mitochondrial adaptations, however the discussion has not been revised. The authors state 'SIT leads to similar or even larger increases in maximum aerobic capacity (VO₂peak) in healthy populations as compared to classical MICT due to more pronounced skeletal muscle mitochondrial remodeling', however the 'classical' MICT involves 5-days/week of training and this volume of training results in a different interpretation (again, see seminal work from Dr Gibala's laboratory).

We understand and actually agree with the reviewer's point of view. Apparently, the modifications made to the discussion were not clear enough to overcome the reviewer's critique on this point. We agree and do not question that MICT is an effective means to improve aerobic capacity and therefore has its place to improve and maintain physical fitness. Previous studies from Gibala's laboratory (e.g. Burgomaster et al. 2008; Gibala et al. 2006) indeed showed similar muscle adaptations after several weeks of MICT vs. SIT. Here the point we are trying to make is that SIT (and HIIT for that matter) are particularly time-efficient means to increase aerobic capacity, and therefore a mechanistic understanding of their efficacy is of great interest. We have made further adjustments to the discussion and hope that the present version satisfies the reviewer.

The authors were asked about positive and negative controls following IP experiments, but these remain missing. If the authors IP RyR they should blot for 'non-RyR' proteins (e.g. CSQ, DHPR, SERCA).

We thank the reviewer for this clarification. We had misinterpreted the previous request of the reviewer: in the previous revision, we added positive (IP on samples treated with H₂O₂, NOC and PKA) and negative (no antibody in IP) controls for calstabin1 dissociation from the RyR1 channels. We apologize for this.

To fully address the reviewer's request, we have now checked SERCA protein presence after RyR1 immunoprecipitation. Importantly, our results showed absence of SERCA detection when the RyR1 is immunoprecipitated whereas SERCA was detectable in the non-IP sample (Figures S1F and S2G), providing additional validation for the specificity of our IPs. We hope that these results now convince the reviewer on the specificity of our RyR1 immunoprecipitation assays. This IP technique has previously been used in many publications (Marx et al, Cell. 2000; Reiken et al, JCB. 2003, Lehnart et al, Cell. 2005; Santulli et al, PNAS. 2015; Andersson et al, 2011, Matecki et al, PNAS. 2015).

The authors provide cellular NADH/NAD⁺, however it is not clear what these data tell the reader about mitochondrial metabolism, as cytosolic NAD⁺ levels are estimated to be 1000-fold higher than mitochondrial.

We agree with the reviewer on the different subcellular localizations of NAD⁺ and NADH. We performed this set of experiments to investigate at a global cellular level the NADH production in our conditions, as different observations in our study pointed to a specific NADH-linked pathways triggered by the RyR1 Ca²⁺ leak in response to SIT. We now clarified this point in the manuscript.

REVIEWERS' COMMENTS

Reviewer #3 (Remarks to the Author):

I have no further comments- congratulations on a nice manuscript.